# Evaluating NO$_x$ Fate and Organic Nitrate Chemistry from *α*-Pinene Oxidation Using Stable Oxygen and Nitrogen Isotopes

Wendell W. Walters[1,2], Masayuki Takeuchi[3,a], Danielle E. Blum[4], Gamze Eris[5], David Tanner[6], Weiqi Xu[5,7], Jean Rivera-Rios[5,b], Fobang Liu[5,8], Tianchang Xu[5], Greg Huey[6], Justin B. Min[6], Rodney Weber[6], Nga L. Ng [3,5,6], and Meredith G. Hastings[2,9]

[1]Department of Chemistry and Biochemistry, University of South Carolina, 631 Sumter St, Columbia, SC 29208, USA
[2]Institute at Brown for Environment and Society, Brown University, 85 Waterman St, Providence, RI 02912, USA
[3]School of Civil and Environmental Engineering, Georgia Institute of Technology, 311 Ferst Drive NW, Atlanta, GA 30332, USA
[4]Department of Chemistry, Brown University, 324 Brook Street, Providence, RI 02912, USA
[5]School of Chemical and Biomolecular Engineering, Georgia Institute of Technology, 311 Ferst Drive NW, Atlanta, GA 30332, USA
[6]School of Earth and Atmospheric Sciences, Georgia Institute of Technology, 311 Ferst Drive NW, Atlanta, GA 30332, USA
[7]State Key Laboratory of Atmospheric Boundary Layer Physics and Atmospheric Chemistry, Institute of Atmospheric Physics, Chinese Academy of Sciences, Beijing, 100029, China
[8]Department of Environmental Science and Engineering, Xi'an Jiaotong University, Xi'an, Shaanxi, 710049, China
[9]Department of Earth, Environmental, and Planetary Sciences, Brown University, 324 Brook Street, Box 1846, Providence, RI 02912, USA

*Correspondence to*: Wendell W. Walters (wendellw@mailbox.sc.edu)

[a]Now at Department of Mechanical Engineering, University of Colorado Boulder, 1111 Engineering Dr, Boulder, CO, 80309, USA
[b]Now at Department of Chemistry and Chemical Biology, Rutgers University, 123 Bevier Rd, Piscataway, NJ, 08854, USA

**Abstract.** The oxidation of biogenic volatile organic compounds (BVOCs) such as *α*-pinene in the presence of nitrogen oxides (NO$_x$ = NO + NO$_2$) initiates complex photochemical processes that produce organic nitrates (RONO$_2$) and influence atmospheric oxidation capacity, air quality, and the fate of reactive nitrogen. However, tracking the chemical fate of RONO$_2$ remains challenging as it includes pathways such as renoxification, aerosol partitioning, deposition, and/or hydrolysis to nitric acid (HNO$_3$). Stable oxygen ($\Delta^{17}O$, $\delta^{18}O$) and nitrogen ($\delta^{15}N$) isotope measurements can provide a unique tool to probe these processes, as NO$_y$ species can exhibit distinct isotopic signatures due to characteristic oxygen-transfer dynamics and isotope fractionation. Here, we present chamber experiments of *α*-pinene oxidation in the presence of NO$_x$ under a range of oxidant and photochemical conditions, reporting the $\Delta^{17}O$, $\delta^{18}O$, and $\delta^{15}N$ values of simultaneously collected NO$_2$, HNO$_3$, and particulate nitrate (pNO$_3$), the latter of which derived predominantly from RONO$_2$ in the conducted experiments. A strong linear relationship between $\delta^{18}O$ and $\Delta^{17}O$ across all NO$_y$ species ($r = 0.992$; $p < 0.01$) supports a two-endmember mixing model, in which oxygen atoms are transferred from isotopically distinct sources that include ozone (O$_3$) with high $\delta^{18}O/\Delta^{17}O$ and peroxy/hydroxyl radicals (RO$_2$/HO$_2$/OH) with lower values. Nitrogen isotope fractionation, quantified as the difference in

$\delta^{15}$N values ($\Delta\delta^{15}$N), revealed consistently positive $\Delta\delta^{15}$N(HNO$_3$ – NO$_2$) values (+28.9 ± 13.4 ‰ in daytime experiments; +22.2 ± 1.4 ‰ at night) and negative $\Delta\delta^{15}$N(pNO$_3$-NO$_2$) values (–13.6 ± 5.8 ‰ in daytime experiments). This reflected distinct formation pathways and isotope effects including NO$_x$ photochemical cycling, thermal dinitrogen pentoxide (N$_2$O$_5$)–nitrate radical (NO$_3$)–NO$_2$ equilibrium, and HNO$_3$ production mechanisms. Box model simulations based on $\Delta^{17}$O values as a constraint were conducted using a newly developed gas-phase mechanism, which reproduced $\Delta^{17}$O(NO$_2$) and $\Delta^{17}$O(pNO$_3$) (compared to simulated $\Delta^{17}$O(RONO$_2$)) accurately, with an average model bias of 0.9 ± 2.4 ‰ ($R^2$ = 0.98) and -1.4 ±2.4 ‰ ($R^2$ = 0.55 and $R^2$ = 0.97 when excluding one outlier), respectively. We further empirically derived important isotopic parameters such as the $\Delta^{17}$O value transferred from O$_3$ through comparison of model-simulated oxygen atom source contributions with observed $\Delta^{17}$O values for NO$_2$ and pNO$_3$ across experiments. This yielded best-fit slopes of 39.4 ± 0.6 ‰ for NO$_x$ photochemical cycling and 41.7 ± 1.2 ‰ for RONO$_2$ formation, consistent with near-surface observations of $\Delta^{17}$O in the terminal oxygen atom of O$_3$. Despite the agreement with NO$_2$ and RONO$_2$, accurately simulating $\Delta^{17}$O(HNO$_3$) proved challenging. Sensitivity tests revealed that model biases likely stemmed from a combination of factors including background HNO$_3$ chamber blanks affecting low-NO$_x$ experiments, missing N$_2$O$_5$ heterogeneous hydrolysis under nighttime conditions, and an overestimation in the $\Delta^{17}$O(HNO$_3$) mass balance resulting from the NO$_2$ + OH reaction, which was improved by adjusting the contribution from (2/3)$\Delta^{17}$O(NO$_2$) to (1/2)$\Delta^{17}$O(NO$_2$). These adjustments reduced the average model bias in $\Delta^{17}$O(HNO$_3$) from 6.7 ± 3.3 ‰ ($R^2$ = 0.39) in the base mechanism to 1.6 ± 1.3 ‰ ($R^2$ = 0.48) in the modified mechanism. These findings demonstrate the utility of $\Delta^{17}$O and $\delta^{15}$N for disentangling nitrate formation mechanisms, while also highlighting critical gaps in our understanding of the isotope dynamics involving HNO$_3$ formation. Future experimental work targeting isolated HNO$_3$ pathways is essential to refine isotopic mass-balance assumptions and nitrogen isotope fractionation.

## 1 Introduction

The oxidation of biogenic volatile organic compounds (BVOCs) in the presence of nitrogen oxides (NO$_x$ = NO + NO$_2$) plays a central role in atmospheric chemistry, linking anthropogenic emissions to the formation of ozone (O$_3$), secondary organic aerosols (SOA), and the cycling of reactive nitrogen (Hoyle et al., 2011; Ng et al., 2017; Romer et al., 2016; Sato et al., 2022; Takeuchi and Ng, 2019; Xu et al., 2015b, 2020; Zare et al., 2018). A key product of this interaction is organic nitrate (RONO$_2$), which can be produced both during daytime and nighttime oxidation reactions involving monoterpenes (Ng et al., 2017). The RONO$_2$ product can act either as a temporary NO$_x$ reservoir or a permanent sink, depending on its atmospheric fate (Fisher et al., 2016). Once formed, gas-phase RONO$_2$ can either photolyze or oxidize, leading to the release of NO$_x$ ("renoxification"), partition into the particle phase, resulting in particulate RONO$_2$ (pRONO$_2$) (Beaver et al., 2012; Browne et al., 2014; Browne and Cohen, 2012; Fisher et al., 2016; Wang et al., 2023), and/or undergo dry and wet deposition, leading to the removal of reactive nitrogen from the atmosphere. Hydrolysis has emerged as an important pathway that directly converts RONO$_2$ into

HNO₃, which acts as a permanent sink of NO$_x$, but the efficiency of this process remains challenging to constrain. Model investigations have suggested a lifetime of RONO$_2$ with respect to hydrolysis on the order of a few hours (Fisher et al., 2016; Zare et al., 2018). Experimental measurements have indicated a complex picture, in which not all monoterpene-derived RONO$_2$ hydrolyze, and that the lifetime of $\alpha$-pinene derived RONO$_2$ due to hydrolysis ranges on the order of minutes to hours and even days (Rindelaub et al., 2015; Takeuchi and Ng, 2019; Wang et al., 2021). Due to these complexities, the contributions of RONO$_2$ to HNO$_3$ and particulate nitrate (pNO$_3$) budgets remain poorly constrained. These uncertainties hinder our ability to predict NO$_x$ lifetime and oxidant budgets, particularly in regions where high BVOC emissions coincide with anthropogenic NO$_x$ such as forested areas near urban locations. Here, we develop new tools aimed at tracking RONO$_2$ contributions to HNO$_3$ and pNO$_3$ budgets using stable isotope measurements ($\Delta^{17}O$, $\delta^{18}O$, $\delta^{15}N$) in controlled $\alpha$-pinene + NO$_x$ chamber experiments, providing mechanistic insights into RONO$_2$ formation and loss pathways.

The natural variations of stable oxygen and nitrogen isotopes in various reactive nitrogen (NO$_y$ = NO$_x$ + HNO$_3$ + RONO$_2$ + peroxy nitrate (RO$_2$NO$_2$) + etc.) molecules offer a powerful diagnostic tool to investigate NO$_x$ and oxidation chemistry (Michalski et al., 2012; Walters et al., 2018). Stable isotope constraints may also serve as quantitative tracers to distinguish daytime (e.g., RO$_2$ + NO) versus nighttime (e.g., NO$_3$ + BVOC) RONO$_2$ production, quantify the extent of RONO$_2$ hydrolysis to HNO$_3$, and constrain the relative contributions of organic versus inorganic pathways to the pNO$_3$ budget. Variations in oxygen (O) isotope ratios (i.e., $^{18}O/^{16}O$ and $^{17}O/^{16}O$), commonly quantified using isotope delta notation ($\Delta^{17}O$ and $\delta^{18}O$), offer a powerful proxy for assessing oxidation pathways involving NO$_x$ photochemical cycling and nitrate formation (Albertin et al., 2021; Alexander et al., 2020; Michalski et al., 2003; Morin et al., 2011; Walters et al., 2024b). This is owing to distinct $\Delta^{17}O$ and $\delta^{18}O$ values exhibited by key atmospheric oxidants, which are proportionally transferred to NO$_x$ during oxidation in the atmosphere (Hastings et al., 2003; Michalski et al., 2003). For instance, tropospheric O$_3$ has an elevated $\Delta^{17}O$ with a mean value near 26 ‰, and the transferable terminal oxygen atom of O$_3$ (O$_3^{term}$) exhibiting a $\Delta^{17}O$ of 39 ± 2 ‰ and elevated $\delta^{18}O$ near 126 ± 12 ‰ based on recent near-surface observations (Ishino et al., 2017; Vicars and Savarino, 2014). In contrast, other atmospheric oxidants such as RO$_2$/HO$_2$ and OH have $\Delta^{17}O$ values assumed to be near 0 ‰ (Lyons, 2001; Walters et al., 2024a). The $\delta^{18}O$ values of RO$_2$/HO$_2$ and OH have not been directly determined but are anticipated to be lower than the $\delta^{18}O(O_3^{term})$ (Michalski et al., 2012).

The $\Delta^{17}O$ isotopic composition provides a quantitative framework to evaluate NO$_x$ photochemical cycling and the formation pathways of nitrate-containing species. This tracer has been widely used to assess NO$_x$ oxidation and secondary product formation, as different atmospheric reactions impart distinct $\Delta^{17}O$ signatures based on mass-balance relationships (Alexander et al., 2020; Michalski et al., 2003; Morin et al., 2011; Walters et al., 2024b) (Table 1). For instance, mass-balance calculations predict that $\Delta^{17}O$ should differ between HNO$_3$ produced via the daytime NO$_2$ + OH reaction and that derived from the hydrolysis of daytime formed RONO$_2$, where the $\Delta^{17}O$ of the nitrooxy (-NO$_3$) functional group reflects a combination of NO + RO$_2$ oxidation. These differences make $\Delta^{17}O$ a potentially powerful diagnostic tool for quantifying RONO$_2$ contributions to

HNO₃ formation. Moreover, substantial $\Delta^{17}O$ differences are expected between $RONO_2$ formed through daytime $RO_2$ + NO reactions and those formed via nighttime $NO_3$ + BVOC reactions. Thus, $\Delta^{17}O$ could also be a useful tool for distinguishing between daytime and nighttime formation pathways of $RONO_2$, potentially aiding in our ability to accurately predict the

105 atmospheric burden of $RONO_2$. Despite major advances in applying $\Delta^{17}O$ to constrain atmospheric nitrogen oxidation chemistry, a significant limitation remains: most $\Delta^{17}O$ pathway estimates rely on theoretical mass-balance assumptions that have only been empirically validated for a limited number of reactions (e.g., NO + O₃ and NO₂ + O₃) (Berhanu et al., 2012; Savarino et al., 2008). Further, we have yet to have measured the $\Delta^{17}O$ of $RONO_2$ directly. Expanding direct $\Delta^{17}O$ measurements of key $NO_y$ compounds under various oxidant conditions is critical for testing these assumptions.

Table 1. Expected $\Delta^{17}O$ values for reactive nitrogen species formed via different oxidation pathways based on oxygen isotope mass-balance. Each pathway is expressed as a weighted average of the $\Delta^{17}O$ values of precursor oxidants, based on the proposed reaction mechanism. Species include $NO_2$, $HNO_3$, and $RONO_2$.

| Formation Pathway | Expected $\Delta^{17}O$ |
|---|---|
| NO₂ | |
| NO + O₃[a] | $1/2\,(\Delta^{17}O(NO)) + 1/2\,(\Delta^{17}O(O_3^{term}))$ |
| NO + RO₂[b] | $1/2(\Delta^{17}O(NO)) + 1/2\,(\Delta^{17}O(RO_2))$ |
| NO + HO₂[c] | $1/2\,(\Delta^{17}O(NO)) + 1/2(\Delta^{17}O(HO_2))$ |
| HNO₃ | |
| NO₂ + OH | $2/3(\Delta^{17}O(NO_2)) + 1/3(\Delta^{17}O(OH))$ |
| NO₃ + HC[d] | $\Delta^{17}O(NO_3)$ |
| NO + HO₂[b] | $1/3(\Delta^{17}O(NO))+2/3(\Delta^{17}O(HO_2))$ |
| N₂O₅ + H₂O (aq) | $5/6(\Delta^{17}O(N_2O_5))+1/6(\Delta^{17}O(H_2O))$ |
| N₂O₅ + Cl⁻ (aq) | $5/6(\Delta^{17}O(N_2O_5))+1/6(\Delta^{17}O(O_3^{term}))$ |
| NO₂ + H₂O (aq) | $2/3(\Delta^{17}O(NO_2)) + 1/3(\Delta^{17}O(H_2O))$ |
| NO₃ + H₂O (aq) | $\Delta^{17}O(NO_3)$ |
| RONO₂ + H₂O (aq) | $\Delta^{17}O(RONO_2)$ |
| RONO₂[*] | |
| NO + RO₂[b] | $1/3(\Delta^{17}O(NO))+2/3(\Delta^{17}O(RO_2))$ |
| NO₃ + BVOC | $\Delta^{17}O(NO_3)$ |

[a]$O_3^{term}$ = terminal oxygen in ozone

[b]$RO_2$ = organic peroxy radical

[c]HO$_2$ = hydroperoxyl radical

[d]HC = hydrocarbon

*$\Delta^{17}$O calculated from the nitrooxy (-NO$_3$) functional group.

The stable nitrogen (N) isotope ratio variations ($\delta^{15}$N) of NO$_x$ and atmospheric nitrate have long served as a valuable proxy for evaluating precursor emission sources, because of the preserved N mass between the precursor and oxidized end-products (Elliott et al., 2019; Hastings et al., 2013). However, it is essential to consider that NO$_x$ photochemical cycling and atmospheric

nitrate formation processes can also induce significant mass-dependent fractionation effects (Freyer et al., 1993; Li et al., 2020; Walters et al., 2016; Walters and Michalski, 2015, 2016a). Field $\delta^{15}$N observations of NO$_2$ and nitrate have demonstrated the potential of these fractionation effects to offer additional valuable information concerning NO$_x$ photochemical cycling and atmospheric nitrate formation (Albertin et al., 2021; Bekker et al., 2023; Li et al., 2021; Walters et al., 2018). Recently, a novel chemical mechanism was devised to model the nitrogen isotope fractionation associated with NO$_x$ chemistry, called

incorporating $^{15}$N into the Regional Atmospheric Chemistry Mechanism (i$_N$RACM) (Fang et al., 2021). Leveraging these advancements, we may utilize $\delta^{15}$N to gather supplementary quantitative insights into BVOC/NO$_x$ interactions and their impact on RONO$_2$ and contributions to HNO$_3$ formation. Importantly, nitrogen isotope fractionation may lead to distinct $\delta^{15}$N values in HNO$_3$ and RONO$_2$ due to differences in their formation pathways. If characterized, these $\delta^{15}$N signatures could offer an additional tracer for quantifying RONO$_2$ contributions to the overall HNO$_3$ budget. While strides have been made in

understanding $\delta^{15}$N fractionation during NO$_x$ oxidation and equilibrium partitioning (Freyer et al., 1993; Li et al., 2020; Walters et al., 2016; Walters and Michalski, 2015), the specific nitrogen fractionation factors associated with HNO$_3$ and RONO$_2$ formation remain poorly constrained. Targeted laboratory and field studies are needed to directly measure these values and validate their use as diagnostic tools in atmospheric reactive nitrogen chemistry.

This study presents the first simultaneous measurements of $\Delta^{17}$O, $\delta^{18}$O, and $\delta^{15}$N in key NO$_y$ species that included NO$_2$, HNO$_3$, and RONO$_2$ that were generated under controlled laboratory conditions involving $\alpha$-pinene oxidation. By varying oxidant regimes to probe distinct RO$_2$ fates, we aimed to (1) determine the $\Delta^{17}$O and $\delta^{15}$N values of simultaneously collected HNO$_3$, NO$_2$, and RONO$_2$ under a range of atmospheric oxidation conditions; (2) evaluate the validity of oxygen isotope mass-balance assumptions used in the formation of NO$_2$, HNO$_3$, and RONO$_2$; (3) characterize nitrogen isotope fractionation patterns across

NO$_y$ species; and (4) assess whether stable isotope measurements can provide meaningful constraints on the fate of RONO$_2$ in the atmosphere. Further, by incorporating recent isotope modeling frameworks (Walters et al., 2024a) we simulated the $\Delta^{17}$O values of various NO$_y$ components and compared them against observations, yielding insights into the oxidative formation and $\Delta^{17}$O transfer dynamics during NO$_x$ oxidation.

## 2 Methods

### 2.1 Chamber Experiments

Photochemical and nighttime oxidation chamber experiments were conducted involving $\alpha$-pinene, $NO_x$, and oxidant precursors at the Georgia Institute of Technology Environmental Chamber Facility that houses two 12 $m^3$ Teflon reactors (Boyd et al., 2015). A total of six different initial experimental conditions were targeted, including five photochemical and one nighttime condition as previously reported (Blum et al., 2023) (Table 2). The experiments varied in their precursor concentrations and oxidant types, which were utilized to probe different $\alpha$-pinene oxidation reactions involving OH, $O_3$, and $NO_3$ and $RO_2$ fates. Replicates were conducted in two of the targeted experimental conditions. The conducted chamber experiments follow previous laboratory protocols (Boyd et al., 2015; Nah et al., 2016; Takeuchi and Ng, 2019; Tuet et al., 2017). Briefly, photochemical experiments were conducted by injecting dry ammonium sulfate seed aerosol and precursor (i.e., $\alpha$-pinene (99 % Sigma-Aldrich)), NO (Matheson), hydrogen peroxide ($H_2O_2$), or nitrous acid (HONO)) into the chamber, where either $H_2O_2$ or HONO was used as an OH precursor to simulate different extents of $RO_2+NO$ pathway. Once the levels of precursors stabilized, the chamber lights were turned on, signifying the start of the photochemical experiments. The procedure used to generate HONO (e.g., the reaction of sodium nitrite with sulfuric acid) also leads to the generation of significant NO and $NO_2$ as a reaction by-product (Kroll et al., 2005; Tuet et al., 2017). For simulated nighttime conditions, dry ammonium sulfate seed aerosol and $\alpha$-pinene were injected into the chamber, followed by flowing dinitrogen pentoxide ($N_2O_5$) for fifteen minutes (Boyd et al., 2015; Takeuchi and Ng, 2019). The $N_2O_5$ injection corresponded to the start of the nighttime experiments. The $N_2O_5$ was generated by reacting $NO_2$ from a gas cylinder (Matheson) with $O_3$ in a flow tube prior to the introduction to the chamber at a ratio of 2:1 to minimalize $O_3$ concentrations in the chamber to avoid ozonolysis. All experiments were conducted at a relative humidity (RH) and temperature of 30 % and 22 °C, respectively. Before each experiment, the chamber was flushed with zero air and irradiated for at least 24 hours.

**Table 2. Summary of measured $NO_y$ precursor concentrations and targeted $H_2O_2$ concentrations for the chamber experiments. All experiments were conducted using dry ammonium sulfate seed at a fixed temperature (22 °C) and relative humidity (30 %).**

| Experiment | $\alpha$-pinene (ppb) | $NO_y$ (ppb) | Oxidant (ppb) |
|---|---|---|---|
| 1 | 298 | NO = 55.3 | $H_2O_2$ = 9,000 |
| 1R | 297 | NO = 49.5 | $H_2O_2$ = 9,000 |
| 2 | 290 | NO = 112 | $H_2O_2$ = 6,000 |
| 3 | 286 | NO = 338 | $H_2O_2$ = 6,000 |
| 4 | 293 | NO = 615 | $H_2O_2$ = 4,500 |
| 4R | 316 | NO = 655 | $H_2O_2$ = 4,500 |

| 5 | 306 | HONO = 210<br>NO = 320<br>$NO_2$ = 460 | N/A |
|---|---|---|---|
| 6 | 100 | $NO_2^a$, $NO_3^a$, $N_2O_5^a$,<br>$HNO_3^a$, $O_3^a$ | N/A |

[a]The emission rate of $NO_2$, $NO_3$, $N_2O_5$, $HNO_3$, and $O_3$ into the chamber for a 20 minute injection period were modeled based on a flow tube simulation of the reaction of $NO_2$ with $O_3$ with a residence time of 70 s.

Continuous online measurements of NO, $NO_2$, and $O_3$ were conducted using chemiluminescence (Teledyne 200EU), cavity-attenuated phase shift (CAPS), and an $O_3$ monitor (Teledyne T400). A chemical ionization mass spectrometer (CIMS) was used for various $NO_y$ measurements including HONO and $HNO_3$ (Huey et al., 1998). The $\alpha$-pinene decay was monitored using gas-chromatography flame ionization detector (GC-FID; Agilent 7890A). Gaseous organic nitrate were monitored using a filter inlet for gases and AEROsols (FIGAERO) coupled to a high-resolution time-of-flight iodide chemical ionization mass spectrometer (HR-ToF-I-CIMS) with particles collected on a Teflon filter (Lopez-Hilfiker et al., 2014; Nah et al., 2016; Takeuchi et al., 2022; Takeuchi and Ng, 2019; Wang et al., 2023). Aerosol composition was measured using a high-resolution time-of-flight aerosol mass spectrometer (HR-ToF-AMS) that included measurement of non-refractory organics (Org), sulfate ($SO_4$), nitrate ($NO_3$), and ammonium ($NH_4$) (DeCarlo et al., 2006; Farmer et al., 2010). Water-soluble aerosol components were also measured using a particle-into-liquid sampler (PILS) coupled to ion chromatography (IC) (Orsini et al., 2003). This method differs from the HR-ToF-AMS measurements, which quantify total aerosol composition, including both water-soluble and water-insoluble components. For example, nitrate measured by the AMS represents total $pNO_3$, whereas nitrate measured by the PILS-IC system corresponds only to the water-soluble fraction of $pNO_3$.

Collections of various $NO_y$ gaseous and aerosol components, including $HNO_3$, $NO_2$, and $pNO_3$ were conducted using a modified version of the ChemComb Speciation Cartridge (CCSC) with an extended denuder body for offline concentration and isotope composition analysis (Blum et al., 2020, 2023). Briefly, the CCSC collections began when the aerosol mass spectrometer data indicated the nitrate and secondary organic aerosol mass concentrations had peaked. The CCSC samples were collected at 8 L min$^{-1}$ for up to 4 h. To maintain the chamber integrity, zero-air was used to dilute at 25 L min$^{-1}$ once aerosol peak was reached and CCSC sample collection initiated. The CCSC denuder bodies were replaced one to four times depending on the concentration of $NO_x$ in the chamber. For each experiment, a single filter was used in the CCSC. In addition to the chamber experiments, samples were collected directly from the $NO_2$ tank (Matheson), which was used in the generation of $N_2O_5$ for the nighttime oxidation experiments.

Honeycomb denuders were coated for the selective collection of $HNO_3$ (captured as nitrate ($NO_3^-$)) and $NO_2$ (captured as nitrite ($NO_2^-$)). A detailed description of the coating solutions, denuder preparation, and denuder extraction was previously described, and the pooled isotope reproducibility for both $HNO_3$ and $NO_2$ was ±1.7 ‰, ±1.8 ‰, and ±0.7 ‰, for $\delta^{15}N$, $\delta^{18}O$, and $\Delta^{17}O$

for these chamber experiments (Blum et al., 2023). The collection of $PM_{2.5}$ was conducted using a quartz filter (Cytiva Whatman, Grade QM-A; 47 mm diameter) that was housed in the ChemComb filter cartridge positioned downstream of the denuders. Prior to use, filters were pre-combusted at 550 °C overnight and stored in an airtight container. Quartz filters were selected because they facilitate efficient water-based extraction of collected material and tolerate high-temperature pre-
210 cleaning to remove organic contaminants. The filter samples were extracted in 20 mL of Milli-Q water (>18.2 MΩ) and allowed to leach for at least one week at room temperature. This method was conducted to enable hydrolysis of collected organic nitrate particles as previous studies have shown organic nitrate derived from $\alpha$-pinene oxidation to hydrolyze to $NO_3^-$ in water with a lifetime of 8.8 h at pH = 6.9 (Rindelaub et al., 2016) and 2.5 h at pH = 7.44 (Wang et al., 2021). Other types of organic nitrate, such as secondary nitrates, have been reported to be stable in water, especially at a neutral pH (Wang et al., 2021). The
215 efficiency of our filter extraction technique for facilitating organic nitrate hydrolysis was evaluated using the online AMS and PILS data. After the filters were leached, the filters were removed, and the samples were shipped to Brown University where they were placed in a freezer until subsequent concentration and isotope analysis. For all sample media types, including denuders and filters, lab blanks were frequently taken. These blanks were prepared, handled, and analyzed the same way as all samples.

## 2.2 Concentration and Isotope Analysis

The denuder and filter extracts were analyzed for their $NO_2^-$ and $NO_3^-$ content using a standardized colorimetric technique (e.g., EPA Methods 353.2) or ion chromatography, as previously described (Blum et al., 2020, 2023). The limits of detection (LOD) were approximately 0.1 $\mu$mol $L^{-1}$ and 0.3 $\mu$mol $L^{-1}$ for $NO_2^-$ and $NO_3^-$ via colorimetric analysis and 3.0 $\mu$mol $L^{-1}$ for
$NO_2^-$ via ion chromatography. For all analyses, the average percent relative standard deviation was below 5 %. The $NO_2^-$ and $NO_3^-$ concentrations from denuder blank extractions used for $NO_2$ ($n = 5$) and $HNO_3$ ($n = 5$) collection were below detection limits. Significant blanks were observed in the quartz filter extracts (1.5 ± 0.2 $\mu$mol $L^{-1}$; $n = 5$), which we refer to hereinafter as a method blank.

The $\delta^{15}N$, $\delta^{18}O$, and $\Delta^{17}O$ isotope compositions were analyzed using the denitrifier method for $NO_3^-$ samples (e.g., from $HNO_3$ denuder and aerosol filter extracts) following (Casciotti et al., 2002; Kaiser et al., 2007; Sigman et al., 2001) and the sodium azide/acetic acid buffer method for $NO_2^-$ samples (from $NO_2$ denuder extracts), following (McIlvin and Altabet, 2005; Walters and Hastings, 2023). Briefly, 20 nmol of $NO_3^-$ or $NO_2^-$ samples were converted to $N_2O$, which is extracted, purified, concentrated, and injected into a continuous flow isotope ratio mass spectrometer for $\delta^{15}N$ and $\delta^{18}O$ determination from $m/z$
measurement at 44, 45, and 46. In a separate batch analysis, 50 nmol of $NO_3^-$ or $NO_2^-$ were converted to $N_2O$, decomposed to $O_2$ using a gold tube heated at 770 °C, and analyzed at $m/z$ 32, 33, and 34 for $\Delta^{17}O$ determination (Kaiser et al., 2007; Walters and Hastings, 2023). To minimize potential memory effects from residual $O_2$ gold tube and headspace trapping system, samples were grouped and analyzed in separate batches based on their expected $\Delta^{17}O$ values. Specifically, $NO_2$ samples (high $\Delta^{17}O$),

HNO$_3$ (moderate $\Delta^{17}$O), and pNO$_3$ (low $\Delta^{17}$O) were each analyzed in separate batches. Analytical blanks associated with the conversion of NO$_3^-$ and/or NO$_2^-$ to N$_2$O or O$_2$ for subsequent IRMS analysis were assessed for each batch and were always below detection limit (~0.2 nmol). These blanks are not anticipated to affect the analytical precision of the reported isotope values. The samples were calibrated with respect to internationally recognized NO$_3^-$ standards (IAEA-NO-3, USGS35, USGS34) or NO$_2^-$ reference materials (RSIL-N7373 and RSIL-10219) (Böhlke et al., 2003, 2007). In line with the identical treatment principle, the reference materials are treated the same as samples, including matching concentrations, sample amounts, and reagent additions to ensure analytical consistency. The pooled standard deviations of the reference materials were ±0.1 ‰ and ±0.6 ‰ for $\delta^{15}$N and $\delta^{18}$O of the NO$_3^-$ standards ($n = 78$) and ±0.3 ‰ and ±0.3 ‰ for $\delta^{15}$N and $\delta^{18}$O of the NO$_2^-$ reference materials ($n = 15$), respectively. The $\Delta^{17}$O had a pooled standard deviation of ±0.6 ‰ ($n = 53$) .

All isotope measurements were reported relative to reference standards using delta ($\delta$) notation (Eq. 1):

$$\delta = \left(\frac{R_x}{R_{std}} - 1\right) \qquad\qquad \text{(Eq. 1)}$$

where $R$ is the ratio of the abundance of the heavy to the light isotope (i.e., $^{15}$N/$^{14}$N, $^{18}$O/$^{16}$O, or $^{17}$O/$^{16}$O), $x$ denotes the sample, and $std$ is an abbreviation for standard (Sharp, 2017). The nitrogen and oxygen reference material includes atmospheric air and Vienna Standard Mean Ocean Water (VSMOW), respectively. Oxygen isotope mass-independence ($\Delta^{17}$O) was quantified using the linear definition with a mass-dependent coefficient of 0.52, which is approximately representative of oxygen mass-dependent coefficients expected and observed in nature (Young et al., 2002) (Eq. 2):

$$\Delta^{17}O = \delta^{17}O - 0.52 \times \delta^{18}O \qquad\qquad \text{(Eq. 2)}$$

The linear $\Delta^{17}$O approximation is commonly used to describe large mass-independent fractionation such as those related to O$_3$ reactions, and this definition is commonly used in the atmospheric chemistry community to track the influence of O$_3$ oxidation and $\Delta^{17}$O propagation into reactive components (Alexander et al., 2020; Kim et al., 2022; Michalski et al., 2003; Savarino et al., 2013).

## 2.3 Data Reduction and Corrections

Due to significant NO$_3^-$ blanks found in the pNO$_3$ filter extracts (i.e., method blank), the measured nitrate isotope values ($\delta^{15}$N, $\delta^{18}$O, and $\Delta^{17}$O) were corrected using a mass-balance approach to isolate the isotopic composition of NO$_3^-$ generated within the chamber experiments (Eq. 3-4):

$$\delta(\text{corrected, pNO}_3) = \frac{\delta(\text{measure}) - (f(\text{blank}) \times \delta(\text{blank}))}{1 - f(\text{blank})} \qquad\qquad \text{(Eq. 3)}$$

$$f(\text{blank, pNO}_3) = \frac{[\text{NO}_3^-(\text{blank})]}{([\text{NO}_3^-(\text{blank})] + [\text{NO}_3^-(\text{sample})])} \qquad\qquad \text{(Eq. 4)}$$

where [NO$_3^-$], corresponds to the concentration of NO$_3^-$ in either the blank or sample, and $f$(blank) corresponds to the NO$_3^-$ method blank fraction. The quartz filter method blanks had measured $\delta^{15}$N, $\delta^{18}$O, and $\Delta^{17}$O values of 1.6 ± 1.1 ‰, 16.6 ± 1.4

‰, and 3.4 ± 0.5 ‰, respectively ($n$ = 3). Blank corrections were made for all samples when $f$(blank) was less than 30 %. Samples with an $f$(blank) that exceeded 30 % were not reported for their isotope compositions, which included 1 out of 8 quartz filter extracts. The uncertainty in the blank corrected isotope deltas was calculated using a Monte-Carlo simulation for 10,000 iterations and assuming a normal distribution using Matlab. For the quality assurance criterion of an $f$(blank) < 30 %, the uncertainties were calculated to be less than 4.1 ‰, 1.4 ‰, and 0.9 ‰ for $\delta^{15}N$, $\delta^{18}O$, and $\Delta^{17}O$, respectively. These values are small compared to the observed ranges for pNO$_3$, which spanned 67.8 ‰ for $\delta^{15}N$, 29.3 ‰ for $\delta^{18}O$, and 10.4 ‰ for $\Delta^{17}O$, and thus do not significantly affect the interpretation of isotope patterns. These uncertainties reflect the standard deviation of the 10,000 Monte Carlo iterations for each pNO$_3$ sample, which account for uncertainty in both the sample and method blank concentrations and isotope values. These reported uncertainties for chamber-derived pNO$_3$ isotope values represent total propagated error after blank correction, not the raw instrumental precision.

**2.4 Aerosol Nitrate Composition**

The relative contribution of organic aerosol nitrate (pNO$_3$(Org)) to the total pNO$_3$ was determined from two approaches. First, the relative proportion of pNO$_3$(Org), was calculated based on NO$^+$ and NO$_2^+$ HR-ToF-AMS fragmentation as previously described (Farmer et al., 2010; Fry et al., 2009; Kiendler-Scharr et al., 2016; Xu et al., 2015a) (Eq. 5):

$$f(\text{pNO}_3, \text{Org}) = \frac{(R_{\text{obs}} - R_{\text{AN}})(1 + R_{\text{ON}})}{(R_{\text{ON}} - R_{\text{AN}})(1 + R_{\text{obs}})} \qquad \text{(Eq. 5)}$$

where $f$(pNO$_3$, Org) refers to the fraction of pNO$_3$(Org) to the total pNO$_3$, $R$ refers to NO$^+$/NO$_2^+$ fragments, and obs, AN, and ON refers to the observed, ammonium nitrate, and organic nitrate, respectively. The $R_{\text{AN}}$ was obtained from routine ionization efficiency calibration of the HR-ToF-AMS using 300 nm ammonium nitrate aerosols and was 1.37. The $R_{\text{ON}}$ was calculated based on the measured $R_{\text{AN}}$ and the ratio of $R_{\text{ON}}/R_{\text{AN}}$ previously reported for similar conducted experiments (Takeuchi and Ng, 2019), resulting in an $R_{\text{ON}}$ of 2.70 ± 0.29 and 3.86 ± 0.34 for photochemical and nighttime oxidation experiments, respectively. The second method for qualitatively determining $f$(pNO$_3$, Org) involved evaluating the relative change in the molar ratio of NH$_4$/SO$_4$, as an increase in NH$_4$/SO$_4$ has been observed to be associated with inorganic pNO$_3$ formation (Takeuchi and Ng, 2019).

**2.5 Box Model Simulations**

The chamber experiments were simulated using the Framework for 0-D Atmospheric Modeling (F0AM) box model (Wolfe et al., 2016). The conducted model simulations primarily focus on accurately representing initial precursor oxidation (i.e., NO, $\alpha$-pinene) and simulation of $\Delta^{17}O$ observations. We chose not to include $\delta^{15}N$ model simulations in this study due to several key uncertainties. These include incomplete knowledge of the $\delta^{15}N$ values of all initial NO$_y$ sources, limited constraints on isotope effects during NO$_y$ oxidation and photolysis reactions, the potential for unquantified, species-specific fractionation due to chamber wall interactions, and potential for chamber blanks with unknown $\delta^{15}N$ values. Given these limitations, $\delta^{15}N$ modeling will be deferred to future work focused specifically on nitrogen isotope dynamics.

The model was initiated for each experiment using the measured precursor concentrations for NO, $NO_2$, HONO, and $\alpha$-pinene before chamber lights were turned on or $N_2O_5$ was injected and using the targeted $H_2O_2$ concentrations. The pressure, temperature, and relative humidity were fixed at 1013 mbar, 295 K, and 30 %, respectively. The measured chamber light flux data was used. The model was run in two parts for the photochemical reactions, including from lights on to peak SOA mass concentration (part 1) and from aerosol decay and chamber dilution to the end of $NO_y$ collections (part 2). For the nighttime experiments, the model simulations were conducted in three parts, including from the start of $N_2O_5$ injection to the end of $N_2O_5$ injection (part 1), from the end of $N_2O_5$ injection to peak SOA mass concentrations (part 2) and from the decay of organic aerosol and chamber dilution to the end of $NO_y$ collection (part 3). The $N_2O_5$ injection was simulated by first modeling the $NO_2$ reaction with $O_3$ in the flow tube, considering a flow tube residence time of 70 s. The nighttime experiment was then simulated by allowing the flow tube products (i.e., $NO_2$, $O_3$, $NO_3$, $HNO_3$, and $N_2O_5$) to emit into the chamber for 20 minutes (part 1). Next, the experiment was modeled without the flow tube emission to the start of aerosol decay and chamber dilution (part 2) and from aerosol decay and chamber dilution to the end of $NO_y$ collections (part 3). For both photochemical and nighttime experiments, the model simulations from the decay of organic aerosol to the end of $NO_y$ collection included a chamber dilution rate constant ($k_{dil}$) of $3.47 \times 10^{-5}$ s$^{-1}$, which was calculated based on the dilution flow rate (25 LPM) and chamber volume (12 m$^3$). Sensitivity tests were performed to assess the impact of the selected $k_{dil}$ on model results by evaluating a range of $k_{dil}$ values.

A new chemical mechanism was developed, termed $NO_x$-$\alpha$-pinene ($NO_x$-API), to accurately model the oxidation of $NO_x$ and $\alpha$-pinene (Table S1-S6). This mechanism was developed due to difficulties in simulating the initial decay of the aerosol precursors including $\alpha$-pinene, NO, $NO_2$, and HONO for the various experimental conditions using either the Regional Atmospheric Chemical Mechanism, v2 (RACM2) (Goliff et al., 2013) or the Master Chemical Mechanism v3.3.1 (Jenkin et al., 1997; Saunders et al., 2003). The $NO_x$-API mechanism focuses on simulating $\alpha$-pinene and NO decay along with $NO_x$ oxidation but does not intend to accurately simulate SOA production and later-generation chemistry. The mechanism has the inorganic reactions in RACM2, including 16 species and 45 reactions. It also incorporates 29 organic species and 61 reactions to detail organic chemistry up to one generation past pinonaldehyde formation as well as the formation of pinonaldehyde derived peroxyacetyl nitrate formation, with subsequent chemistry represented by a lumped approach. The $\alpha$-pinene oxidation pathways involving OH, $O_3$, and $NO_3$, along with specific reactions of the resulting $RO_2$ with $HO_2$, NO, $NO_3$, and other $RO_2$ radicals, are also included. The photochemical oxidation of $\alpha$-pinene largely follows the MCMv3.3.1 (Saunders et al., 2003), incorporating two hydroxyl-nitrate isomers from OH/$O_2$/NO, including one tertiary (ONITa) and one secondary (ONITb) and the formation of a tertiary pinene carbonyl nitrate (ONITc). Nighttime oxidation chemistry integrates a recent mechanism for organic nitrate formation, producing pinene nitrate hydroperoxide, including one tertiary (ONITOOHa) and one secondary (ONITOOHb) via $HO_2$ reactions and dimer/pinene dinitrate (PDN) through $RO_2$ interactions (Bates et al., 2022). The box-model simulation is a gas-phase mechanism that does not explicitly model heterogeneous reactions or aerosol chemistry, such

as organic nitrate hydrolysis. The impact of heterogenous reactions are considered in the evaluation of the $\Delta^{17}O$ model simulation results compared to observations.

The $\Delta^{17}O$ of $NO_y$ compounds were simulated using the newly developed $NO_x$-API mechanism modified using the InCorporating Oxygen Isotopes of oxidized reactive Nitrogen in the Regional Atmospheric Chemistry Mechanism, Version 2 (ICOIN-RACM2) model framework (Walters et al., 2024a). Briefly, the model framework tracks the transfer and propagation of $\Delta^{17}O$ from $O_3$ into $NO_y$ and $O_x$ species. This mechanism tags the oxygen atoms transferred from $O_3$ into $NO_y$ and $O_x$ considering mass-balance and reaction stoichiometry and enables the offline calculation of $\Delta^{17}O$ based on the output of concentrations of various $NO_y$ and $HO_x$ isotopologues (Eq. 6):

$$\Delta^{17}O(X) = f(Q) \times \Delta^{17}O(O_3^{term}) \tag{Eq. 6}$$

where $X$ refers to the various $NO_y$ and $O_x$ molecules and $f(Q)$ is the fraction of oxygen-atoms deriving from $O_3^{term}$ for a particular molecule. The $\Delta^{17}O(O_3^{term})$ represents the $\Delta^{17}O$ value of the terminal and transferrable O atom of $O_3$, which was assumed to be $39 \pm 2$ ‰ based on recent near-surface collections of $O_3$ (Ishino et al., 2017; Vicars and Savarino, 2014) and $O_3$ generated from $O_2/NO_x$ photochemical experiments conducted under normal temperature and pressure conditions (Michalski et al., 2014).

All $\Delta^{17}O$ model simulations were conducted without considering chamber wall losses. The potential impact of wall loss was evaluated through sensitivity tests that involved comparing simulated $\Delta^{17}O$ values. The sensitivity tests included a no wall loss case, wall loss involving $O_3$, NO, $NO_2$, $HNO_3$, $N_2O_5$, and organic nitrate based on previous reports from chamber experiments (Morales et al., 2021; Wang et al., 2014), and an elevated wall loss scenario, in which the wall loss rate constants were increased by ×10 scenario to account for uncertainty since wall loss rates were not determined in this work (Table S7).

## 3. Results and Discussion

### 3.1 Isotope Observations

The $\Delta^{17}O$, $\delta^{18}O$ and $\delta^{15}N$ measurements of $NO_y$ species offer insight into the oxidation pathways and sources contributing to their formation. In this study, we focus on interpreting the $\Delta^{17}O$, $\delta^{18}O$, and $\delta^{15}N$ of $NO_2$, $HNO_3$, and $pNO_3$ collected under a range of controlled experiments involving $NO_x/\alpha$-pinene oxidation. These observations provide constraints on the relative importance of different oxidants (e.g., $O_3$, OH, $RO_2$) and reaction mechanisms, and they also allow us to test our understanding of oxygen isotope mass-balance assumptions, O-isotope transfer dynamics, and nitrogen isotope fractionation associated with $NO_x$ oxidation. Below, we first discuss the observed patterns in oxygen isotopes ($\Delta^{17}O$ and $\delta^{18}O$), followed by an examination of nitrogen isotope ($\delta^{15}N$) trends, and their implications.

### 3.1.1 $\Delta^{17}O$ and $\delta^{18}O$ of $NO_y$

The observations indicate a significant relationship between $\delta^{18}O$ and $\Delta^{17}O$ across $NO_y$ species ($\delta^{18}O = 11.1(\pm1.0) + 2.42(\pm 0.04)\times\Delta^{17}O$; $r = 0.992$ ; $p < 0.01$) (Fig. 1A). The strong linear relationship between $\delta^{18}O$ and $\Delta^{17}O$ indicates that the oxygen isotopes of the various collected $NO_y$ compounds derived from two dominant pools of O-sources, with high and low $\delta^{18}O$ and $\Delta^{17}O$ values that were consistent across all experimental conditions. The observed $\Delta^{17}O$ and $\delta^{18}O$ values of $NO_y$ species are anticipated to reflect a balance between oxygen atom transfer from $O_3$, which has high $\Delta^{17}O$ and $\delta^{18}O$ values, and oxygen atom transfer from $RO_2$, $HO_2$, $OH$, and $H_2O$ which have lower $\Delta^{17}O$ and $\delta^{18}O$ values. Assuming a $\Delta^{17}O(O_3^{term})$ value of $39 \pm 2$ ‰ (Ishino et al., 2017; Vicars and Savarino, 2014), would indicate a $\delta^{18}O$ transferred from $O_3^{term}$ to the $NO_y$ products of $106 \pm 5.0$ ‰. Conversely, the low-$\Delta^{17}O$ oxidant endmember at a $\Delta^{17}O \approx 0$ ‰ implies an associated $\delta^{18}O$ value of approximately $11.1 \pm 1.0$ ‰. This value likely reflects oxygen transfer from oxidants such as $RO_2$, $HO_2$, and $OH$.

The derived low-end $\delta^{18}O$ oxidant value is consistent with a scenario in which these radicals are sourced from atmospheric $O_2$ ($\delta^{18}O \approx 23$ ‰; Craig, 1957), but undergo isotopic fractionation during their formation and subsequent oxygen atom transfer to $NO_y$. Although the exact $\delta^{18}O$ enrichment factors for $RO_2/HO_2$ or $OH$ formation and reaction are not well-constrained, a net isotope enrichment factor $\sim$ -12 ‰ is plausible, particularly for unidirectional reactions involving $^{18}O$ fractionation (Walters and Michalski, 2016). Additionally, contributions from $OH$ could further influence the low $\delta^{18}O$ endmember, especially if oxygen atom exchange with ambient water vapor occurs (Dubey et al., 1997). Altogether, the inferred $\delta^{18}O$ of 11.1 ‰ for the low-$\Delta^{17}O$ oxidant endmember likely represents a composite signal from multiple oxidants ($RO_2$, $HO_2$, $OH$) originating from $O_2$ and/or $H_2O$, modified by kinetic and equilibrium isotope effects. Despite these uncertainties, the consistent $\delta^{18}O$–$\Delta^{17}O$ trend across $NO_y$ products supports a two-endmember mixing model governed by oxidants with distinct isotopic values.

The $\Delta^{17}O$ and $\delta^{18}O$ values increased in the order $pNO_3$ ($\bar{x} \pm \sigma$; $\Delta^{17}O = 10.0 \pm 3.4$ ‰; $\delta^{18}O = 35.0 \pm 10.1$ ‰; $n = 7$) < $HNO_3$ ($\Delta^{17}O = 16.7 \pm 2.0$ ‰; $\delta^{18}O = 50.2 \pm 4.5$ ‰; $n = 20$) < $NO_2$ ($\Delta^{17}O = 29.6 \pm 12.8$ ‰; $\delta^{18}O = 84.1 \pm 29.6$ ‰; $n = 20$). The $\Delta^{17}O$ and $\delta^{18}O$ of $NO_2$, $HNO_3$, and $pNO_3$ were sensitive to the types of experiments and thus oxidant conditions (Fig. 1B). For example, $NO_2$ samples collected during the photochemical experiments (i.e., Exp. 1-5) indicated that $\delta^{18}O(NO_2)$ and $\Delta^{17}O(NO_2)$ increased with the initial $[NO_y]$, the ratio of initial $[NO_y]$:[BVOC], and with decreasing initial $[H_2O_2]$. These sensitivities of $\delta^{18}O(NO_2)$ and $\Delta^{17}O(NO_2)$ reflect the balance between NO branching ratios involving $O_3$ versus $RO_2/HO_2$ (Albertin et al., 2021; Walters et al., 2018). Thus, the relative branching ratios of $NO+O_3$ and $NO+RO_2/HO_2$ changed with experimental photochemical conditions, favoring a greater proportion of $NO+O_3$ reactions for higher initial $NO_y$ and lower $[H_2O_2]$ conditions. For the nighttime oxidation experiment (Exp. 6), the $\Delta^{17}O$ and $\delta^{18}O$ reflected the initial production of $N_2O_5$ from the oxidation of $NO_2$ (from a gas cylinder) with $O_3$. The expected $\Delta^{17}O$ and $\delta^{18}O$ values can be calculated assuming $N_2O_5$ equilibrium between $NO_3$ and $NO_2$ (i.e., $N_2O_5 \rightleftharpoons NO_2 + NO_3$) and using O isotope mass balance (Eq. 7):

$$\delta(NO_2) = \frac{1}{5}\left(\delta(O_3^{term})\right) + \frac{4}{5}\left(\delta(NO_2^{tank})\right) \qquad \text{(Eq. 7)}$$

where $\delta$ refers to either $\Delta^{17}O$ or $\delta^{18}O$, $O_3^{term}$ refers to the O-atom at the terminal end of $O_3$ and $NO_2^{tank}$ refers to the $NO_2$ from the tank source with measured $\Delta^{17}O$ and $\delta^{18}O$ values of -0.1 ± 0.1 ‰ ($n = 3$) and 13.1 ± 0.2 ‰ ($n = 3$), respectively. Using the assumed $\Delta^{17}O(O_3^{term})$ of 39 ± 2 ‰, and the calculated $\delta^{18}O(O_3^{term})$ incorporated into $NO_y$ (106 ± 5.0 ‰) we estimate the $\Delta^{17}O$ and $\delta^{18}O$ of $NO_2$ for the nighttime oxidation experiment to be 7.7 ± 0.4 ‰ and 31.7 ± 1.0 ‰, respectively, which was near their measured values from the nighttime chamber experiments of (7.2 ± 0.2 ‰) and (31.9 ± 0.7 ‰) ($n = 3$), respectively.

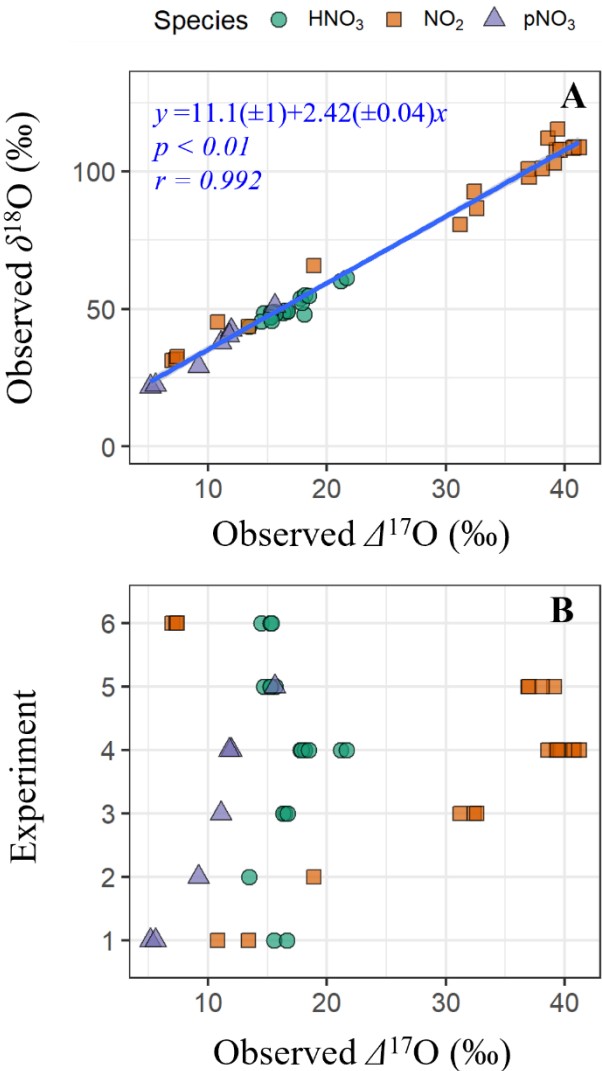

Fig. 1. The observed oxygen isotope delta values of various $NO_y$ species (i.e., $HNO_3$, $NO_2$, and $pNO_3$) from the $\alpha$-pinene/$NO_y$ oxidation experiments including, (A) linear regression between $\delta^{18}O$ and $\Delta^{17}O$ and (B) $\Delta^{17}O$ values of $NO_y$ species sorted by experiment.

### 3.1.2 $\delta^{15}N$ of $NO_y$

The observed $\delta^{15}N$ of all $NO_y$ species exhibited a large range from -90.3 to -4.0 ‰ ($n = 47$) (Fig. 2A). This large range of $\delta^{15}N$ values were significantly influenced by the $\delta^{15}N$ values of the various initial $NO_y$ sources that included tank-NO (Exp. 1-4), HONO (Exp. 5), and tank-$NO_2$ (Exp. 6). The experiments using tank-NO had the lowest $\delta^{15}N$ ($\bar{x} \pm \sigma$) of (-56.1 ± 21.3 ‰; $n = 32$), followed by tank-$NO_2$ of (-34.7 ± 12.2 ‰; $n = 6$), and the highest average was for the HONO experiments of (-7.8 ± 5.7 ‰; $n = 9$). The differences in the observed $\delta^{15}N$ values by $NO_y$ source likely reflect the isotopic composition of the initial $NO_y$ used in each experiment. The trend in observed $\delta^{15}N$ across experiments is consistent with either measured or literature-based reports of the $\delta^{15}N$ values of the initial $NO_y$ species. For example, the $\delta^{15}N$ of laboratory generated HONO, prepared using a similar methodology as in this study, has been reported to be –5.9 ± 0.5 ‰ (Chai and Hastings, 2018). This value is relatively high compared to the $\delta^{15}N(NO_2)$ from tank-$NO_2$ used in these experiments that was measured to be -40.9 ± 0.2 ‰, which is higher than the $\delta^{15}N(NO)$ of tank-NO previously reported at -70.0 ± 1.4 ‰ (Fibiger et al., 2014). In addition to the source $\delta^{15}N$ effects, the experiments also indicate large $\delta^{15}N$ fractionation between the various $NO_y$ species. Overall, $\delta^{15}N(HNO_3)$ averaged -25.9 ± 13.0 ‰ ($n = 20$), which was significantly higher than both $\delta^{15}N(NO_2)$ (-52.5 ± 25.2 ‰; $n = 20$) and $\delta^{15}N(pNO_3)$ (-72.6 ± 22.9 ‰; $n = 7$) based on a two-sample $t$-test ($p < 0.01$). While $\delta^{15}N(NO_2)$ values were higher than those of $\delta^{15}N(pNO_3)$, this difference was not statistically significant based on a two-sample $t$-test ($p > 0.05$). This trend suggests that the produced $HNO_3$ was generally associated with a positive nitrogen isotope enrichment factor ($^{15}\varepsilon$) favoring preferential incorporation of $^{15}N$ relative to $NO_2$, whereas $pNO_3$ formation involved a negative $\varepsilon$, favoring incorporation of $^{14}N$.

We quantified the $\delta^{15}N$ enrichment of $HNO_3$ and $pNO_3$ relative to $NO_2$, as $\Delta\delta^{15}N$, (defined as $\delta^{15}N(product) – \delta^{15}N(NO_2)$) (Fig. 2B). Among the photochemical experiments initiated with NO and $H_2O_2$ (Exp 1-4), the average $\Delta\delta^{15}N(HNO_3–NO_2)$ was consistently high averaging 33.7 ± 7.1 ‰, while the HONO experiment (Exp. 5) had a much smaller enrichment value of 0.4 ± 1.7 ‰. This difference is somewhat surprising, given that both systems are expected to involve $NO_2 + OH$ as a major pathway for $HNO_3$ production. The $^{15}\varepsilon$ associated with $NO_2 + OH$ has yet to be directly measured but has been predicted in the literature with large differences in the suggested value. For example, the $^{15}\varepsilon$ for the $NO_2+ OH$ has been suggested to be -3 ‰ based on the reduced masses of the transition complex (Freyer et al., 1993), while it has been predicted to be +40 ‰ in the $i_NRACM$ mechanism based on the assumption that $NO_2$ and the excited $HNO_3$ intermediate formed during the $NO_2 + OH$ reaction achieve isotopic equilibrium prior to collisional deactivation (Fang et al., 2021). The relatively low $\Delta\delta^{15}N(HNO_3–NO_2)$ observed in the HONO experiment is more consistent with the former, while the higher $\Delta\delta^{15}N(HNO_3–NO_2)$ values in the NO/$H_2O_2$ experiments support the latter interpretation. The cause of this discrepancy remains unclear but may reflect differences in reaction kinetics and environmental conditions. For example, in the HONO experiment, the aerosol formation peak (Fig. 3) and $HNO_3$ production (Fig. S1) occurred relatively rapidly compared to the NO/$H_2O_2$ experiments. This timing shift may potentially alter the influence of nitrogen isotope fractionation effects such as $NO_x$ photochemical equilibrium. Additional isotope effects may also contribute, such as fractionation during HONO photolysis, unknown mass-dependent

processes, or experimental artifacts including wall loss or residual background levels. Further investigation is needed to isolate and quantify the influence of these factors.

In the nighttime experiment (Exp. 6), $\Delta\delta^{15}N(HNO_3–NO_2)$ was also elevated with an average value of 22.2 ± 1.4 ‰. The cause of the nighttime $^{15}N$ enrichment in the generated $HNO_3$ relative to $NO_2$ is likely due to isotopic effects associated with the $NO_2$ + $NO_3$ ⇌ $N_2O_5$ equilibrium, which has been predicted to have $^{15}\varepsilon$ of 25.5 ‰ at 300 K (Walters and Michalski, 2016b) falling near the observed value for the nighttime experiments (Fig. 2B). The $\Delta\delta^{15}N(pNO_3–NO_2)$ values were consistently negative, averaging –13.6 ± 5.8 ‰ across the photochemical experiments. This value suggests that $pNO_3$ formation involved reactions

that preferentially favored $^{14}N$. Before speculating on the cause of the low observed $\Delta\delta^{15}N(pNO_3–NO_2)$, it is essential to first determine the chemical composition of the produced $pNO_3$ and whether it originates primarily from inorganic or organic nitrate.

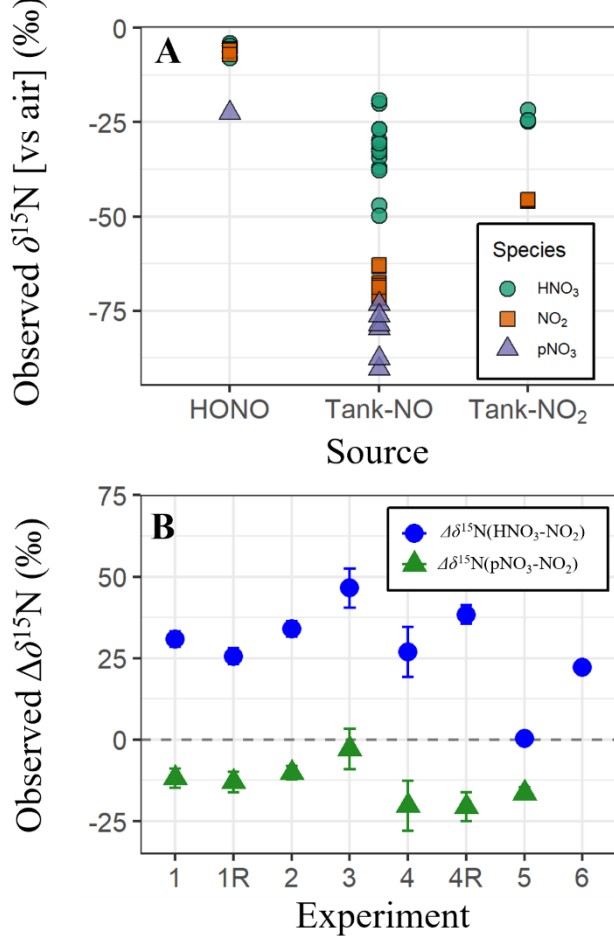

**Fig. 2. (A) The observed $\delta^{15}N$ of various $NO_y$ species (i.e., $HNO_3$, $NO_2$, and $pNO_3$) collected during the various conducted $\alpha$-pinene/$NO_y$ oxidation experiments. The measured $\delta^{15}N$ were sorted by the various starting $NO_y$ sources, including HONO, tank-NO, and tank-$NO_2$. (B) Calculated $\Delta\delta^{15}N$ values for each experiment, defined as the $\delta^{15}N$ difference between $HNO_3$ and $NO_2$, and between $pNO_3$ and $NO_2$. Data represent experiment-specific averages, with error bars reflecting the propagated standard deviation.**

## 3.2 Particle Nitrate Composition

The generated $pNO_3$ could have both inorganic (i.e., $HNO_3$ condensation) and organic (i.e., organic nitrate condensation) contributions. The $\delta^{15}N$ data qualitatively indicates that the $pNO_3$ derived from a separate source than $HNO_3$ due to their large $\delta^{15}N$ differences, which would suggest that $pNO_3$ was derived primarily from organic nitrate. For example, the observed $\delta^{15}N$ difference between $HNO_3$ and $pNO_3$ that averaged 46.7 ‰ suggests that these species may originate from distinct sources. Given that inorganic nitrate would typically equilibrate isotopically between $HNO_3$ and $NO_3^-$ with an expected offset of only ~1–3 ‰, and often slightly enriched in $pNO_3$ (Bekker et al., 2023), the substantially lower $\delta^{15}N$ values observed in $pNO_3$ imply that the collected nitrate may originate from organic nitrate species or $NO_y$ formation pathways unique from $HNO_3$ production.

We utilized the HR-ToF-AMS $NO^+$ and $NO_2^+$ data to evaluate the contributions of $pNO_3$ for the experiments. The $f(pNO_3, \text{Org})$ was calculated according to Eq. 5 for each of the conducted experiments (Table 3). Overall, $f(pNO_3, \text{Org})$ was calculated to have a mean of $(1.25 \pm 0.04; n=8)$, indicating that the generated $pNO_3$ derived from organic nitrate. The calculated $f(pNO_3, \text{Org})$ was higher than 1 even when considering uncertainty estimates. This could be due to calculating $R_{ON}$ values from reported $R_{ON}/R_{AN}$ ratios from previously conducted $\alpha$-pinene oxidation experimental conditions conducted utilizing substantially lower ($\sim \times 10$) initial precursor concentrations (Takeuchi and Ng, 2019). Thus, due to the potential uncertainty in our approach in estimating $f(pNO_3, \text{Org})$, the composition of the generated $pNO_3$ was also investigated using a qualitative approach involving evaluating the relative change in the molar ratio of $NH_4/SO_4$ from the HR-ToF-AMS (Fig. S2). For each type of experiment, we found the $NH_4/SO_4$ molar ratio to be consistently near 1.5. This type of $NH_4/SO_4$ profile is consistent with the generated $pNO_3$ deriving from organic nitrate, as the dissolution of $HNO_3$ into aqueous aerosol followed by neutralization with available $NH_3$ would be expected to lead to an abrupt increase in the molar ratio of $NH_4/SO_4$ (Takeuchi and Ng, 2019). Furthermore, the acidic nature of the particles and limited availability of $NH_4^+$ likely inhibited $HNO_3$ uptake, suppressing condensation pathways and reinforcing the interpretation that $pNO_3$ originated predominantly from organic nitrate formation. Overall, both the quantitative and qualitative analysis of $pNO_3$ composition utilizing the AMS data as well as our $\delta^{15}N$ data indicates that $pNO_3$ was mainly derived from organic nitrate. Hereafter, we assume that the $NO_3^-$ extracted from the filter collections derived from organic nitrate.

**Table 3. Summary of the HR-ToF-AMS data including $NO^+/NO_2^+$ fragmentation data ($R_{obs}$), calculated $f(pNO_3, \text{Org})$, maximum $pNO_3$ (Max($pNO_3$)). Additionally, we quantified the amount of $pNO_3$ from the PILS (PILS/AMS) and the filter collection relative to the HR-ToF-AMS (Filter/AMS). Uncertainties for $pNO_3$ quantification and intercomparison ratios are reported in parentheses.**

| Exp. | $R_{obs}$ ($\bar{x} \pm \sigma$) | $f$(pNO$_3$, Org) ($\bar{x} \pm \sigma$) | Max(pNO$_3$) ($\mu g$ m$^{-3}$) ($\pm14$ %) | PILS/AMS (%) ($\pm24$ %) | Filter/AMS (%) ($\pm24$ %) |
|------|------|------|------|------|------|
| 1 | $3.36 \pm 0.30$ | $1.26 \pm 0.10$ | 13.8 | 41.8 | 97.8 |
| 1R | $3.16 \pm 0.21$ | $1.19 \pm 0.08$ | 13.3 | 37.8 | 83.5 |
| 2 | $3.36 \pm 0.13$ | $1.27 \pm 0.05$ | 25.9 | 33.2 | 105.3 |
| 3 | $3.32 \pm 0.21$ | $1.25 \pm 0.09$ | 27.4 | NA | 80.5 |
| 4 | $3.18 \pm 0.32$ | $1.20 \pm 0.13$ | 25.5 | 42.1 | 75.8 |
| 4R | $3.40 \pm 0.33$ | $1.27 \pm 0.14$ | 15.0 | NA | 76.1 |
| 5 | $3.49 \pm 0.25$ | $1.31 \pm 0.11$ | 24.7 | NA | 59.5 |
| 6 | $5.63 \pm 0.49$ | $1.25 \pm 0.11$ | 38.6 | NA | 7.6 |

The pNO$_3$ measured by the HR-ToF-AMS indicated similar profiles for the various types of conducted experiments, in which pNO$_3$ concentrations peaked and subsequently decayed due to wall loss and chamber dilution (Fig. 3). Overall, the maximum pNO$_3$ concentrations ranged from 13.3 to 38.6 $\mu g$ m$^{-3}$, depending on the experiment (Table 3). The typical measurement uncertainty for pNO$_3$ quantification using the HR-ToF-AMS is approximately $\pm14$ % (Bahreini et al., 2009; Takeuchi et al., 2024). Given that we assume 100% of the particulate nitrate is organic nitrate (i.e., f(pNO$_3$, Org) = 1), the uncertainty in pNO$_3$ concentration is based solely on the AMS nitrate measurement error, estimated at $\pm14$ %. The lowest maximum pNO$_3$ corresponded to experimental conditions with low initial NO$_x$ relative to H$_2$O$_2$ and BVOC conditions (i.e., Exp 1). In contrast, the highest maximum pNO$_3$ occurred during the nighttime oxidation experiments (i.e., Exp 6). The pNO$_3$ concentrations determined from the HR-ToF-AMS were compared with additional measurement techniques, including the PILS and the filter collections for offline analysis (Fig. 3; Table 2). The PILS pNO$_3$ measurements were available for 4 out of the 8 conducted experiments and indicated a similar time profile as the HR-ToF-AMS; however, the PILS pNO$_3$ observations were always lower than the HR-TOF-AMS with the amount of pNO$_3$ determined from PILS relative to the HR-ToF-AMS (PILS/AMS) ranging between 33.2 % to 53.8 %. The uncertainty associated with pNO$_3$ quantification by the PILS system is approximately $\pm20$ % (Guo et al., 2016). Accordingly, the propagated uncertainty for the PILS/AMS ratio is approximately 24 %.

The pNO$_3$ quantified using filter collection and extraction technique was higher than the PILS and in closer agreement with the HR-ToF-AMS. For the photochemical experiments (Exp. 1-5), the pNO$_3$ determined using the filter collection relative to the HR-ToF-AMS (Filter/AMS) for the photochemical experiments ranged between 59.5 to 105.3 % and averaged $86.5 \pm 12.4$ % ($n=7$). The filter collection technique has an estimated uncertainty for pNO$_3$ quantification of approximately $\pm20$ %, based on the average percent difference from side-by-side ChemComb filter pack measurements of ambient air using Nylon filters

(Blum et al., 2020). Although different filters were used in this study, the same collection system and mass flow controllers were employed, and we therefore expect a comparable difference between system replicates. Accordingly, the propagated uncertainty for the Filter/AMS ratio of $\pm 24$ %. However, the filter collection resulted in nearly negligible $pNO_3$ for the nighttime oxidation experiments (i.e., Exp. 6).

The $pNO_3$ concentrations determined using the PILS were always lower than that determined by the HR-ToF-AMS and the offline filter collection technique, which would indicate that that not all collected $pNO_3$, which were shown to mainly derive from organic nitrate, were hydrolyzed to $NO_3^-_{(aq)}$ within the PILS chamber before quantification via Ion Chromatography. The filter collection and extraction method (i.e., leach in MQ water for 1 week), enabled the successful hydrolysis of the collected $pNO_3$ to $NO_3^-_{(aq)}$ from the photochemical experiments, an important pre-requisite for subsequent isotope analysis. The filter collection technique, however, resulted in near negligible $pNO_3$ for the nighttime oxidation experiments, limiting our ability to measure the isotope composition of $pNO_3$ from this experiment. This difference in the efficacy of the offline filter collection technique for $pNO_3$ characterization between the photochemical and nighttime oxidation experiments could be related to the type of organic nitrate formed during the conducted experiments. The photochemical $\alpha$-pinene oxidation experiments have been suggested to result in higher relative production of tertiary organic nitrate, while nighttime oxidation leads to a relatively lower fraction of tertiary organic nitrate with estimated values of 62 % and 15 %, respectively (Zare et al., 2018). Recent work has suggested a hydrolysis lifetime of no more than 30 minutes and a hydrolyzable portion of particulate organic nitrate from $\alpha$-pinene oxidation experiments between 23-32 % and 9-17 % for $\alpha$-pinene + OH and $\alpha$-pinene + $NO_3$ reactions, respectively (Takeuchi and Ng, 2019).

The offline filter collection and extraction technique follows the observed trend of a greater proportion of $pNO_3$ hydrolyzed to $NO_3^-_{(aq)}$ in photochemical experiments ($82.6 \pm 14$ %; $n = 7$) compared to nighttime conditions (7.6 %; $n = 1$). This is inferred based on the relative amount of quantified $NO_3^-_{(aq)}$ from the filter extraction solution to the total $pNO_3$ measured by HR-ToF-AMS. However, the filter-based method suggested a higher proportion of potentially hydrolyzable $pNO_3$ in the photochemical experiments than previously reported estimates. From these results and comparisons, we conclude that the $pNO_3$ offline filter measurements encompass the hydrolysable portion of $pNO_3$ within 1 week, while the HR-ToF-AMS measurements represent the total $pNO_3$, and the PILS measurements correspond to the readily hydrolysable portion of $pNO_3$. Further, the box model simulations (see below) of organic nitrate speciation indicate that the nighttime organic nitrate had a high fraction of dimer and pinene dinitrate (Fig. S3). If this assignment is correct, the results imply that these organic nitrate compounds are effectively non-hydrolyzable under aqueous conditions.

Our results demonstrated that organic nitrate aerosols ($pRONO_2$) can hydrolyze and contribute to the $NO_3^-$ measured in aerosol extracts. While our chamber experiments were conducted under controlled conditions with low relative humidity (~30 %) and dry aerosol seeds, they highlight the need to consider that $NO_3^-$ collected on filters may originate from both inorganic nitrate

(derived from HNO$_3$ uptake) and hydrolyzed organic nitrate. However, we caution that these findings do not imply that all pNO$_3$ observed in ambient field measurements is organic in origin. The extent to which organic nitrate contributes to pNO$_3$ in field settings will depend on regional BVOC emissions, which govern precursor availability, as well as environmental factors such as aerosol pH and relative humidity, which influence the lifetime and hydrolysis rates of pRONO$_2$ prior to filter collection. Further, while gas-phase organic nitrates (e.g., RONO$_2$, RO$_2$NO$_2$) can be present and were detected by CIMS measurements during the experiments, the strong agreement between filter-based and AMS-based pNO$_3$ measurements supports that the nitrate extracted from aerosol filters was primarily derived from particle-phase organic nitrate rather than from gas-phase organic nitrate contributions. Finally, given that the collected pNO$_3$ was predominantly derived from RONO$_2$, the negative $\Delta\delta^{15}N(pNO_3 - NO_2)$ values (Fig. 2B) suggest a preferential incorporation of $^{14}$N into the RONO$_2$ product. This is consistent with a NO$_x$ photochemical equilibrium in which $^{15}$N is enriched in NO$_2$, leaving NO relatively depleted in $^{15}$N (Li et al., 2020; Walters et al., 2016). Subsequent reaction of this $^{15}$N-depleted NO with RO$_2$ forms RONO$_2$, thereby transferring the isotopically lighter nitrogen signature into RONO$_2$ that then condenses to the particle phase and is hydrolyzed to NO$_3^-$.

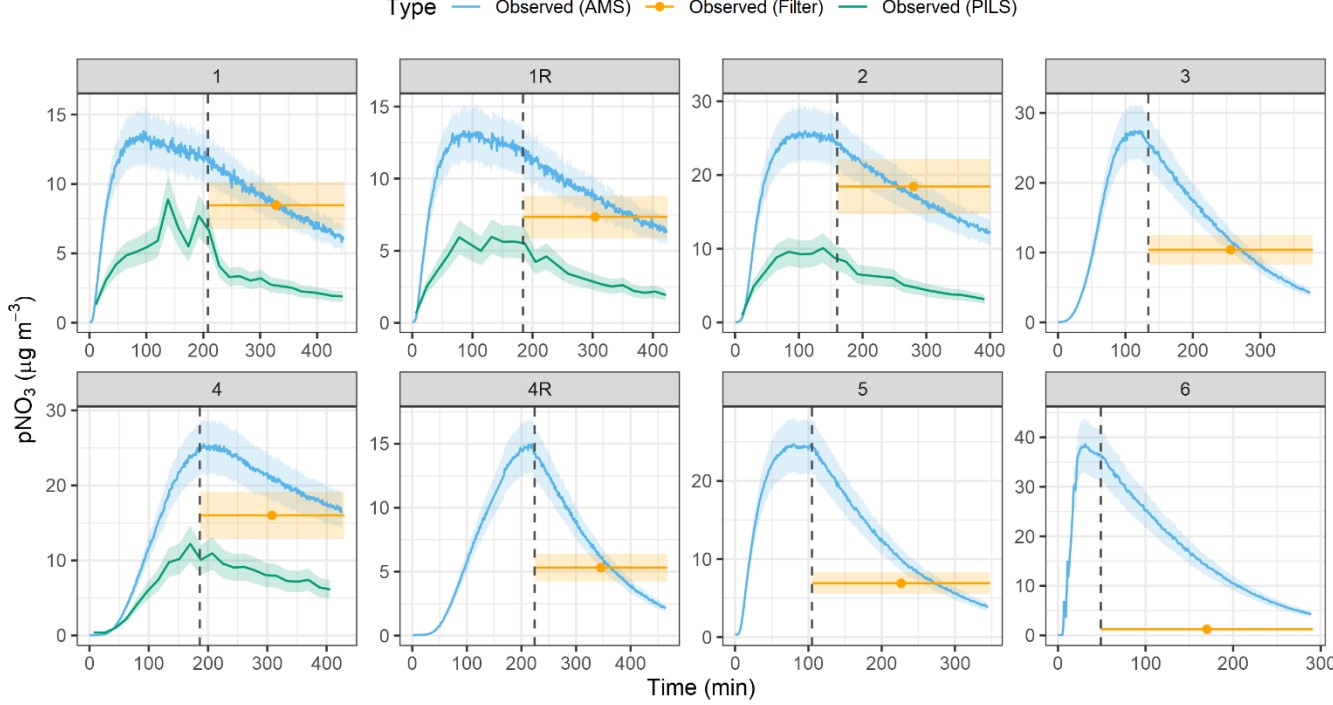

**Fig. 3.** **The observed pNO$_3$ concentration data are shown for each of the conducted experiments. Concentrations were determined using a High-Resolution Time-of-Flight Aerosol Mass Spectrometer (HR-ToF-AMS), a particle-into-liquid sampler (PILS), and filter collection (Filter). The start of chamber dilution is indicated by the dashed vertical lines, corresponding to the abrupt decrease in pNO$_3$. The lighter shaded regions correspond to the measurement uncertainty for the various analytical techniques.**

## 3.3 Model Simulations

To further interpret the experimental results, we employed a box model to simulate the formation and evolution of $NO_y$ species and their $\Delta^{17}O$ values. We begin by examining the developed gas-phase chemical mechanism ($NO_x$-API) simulation of the initial aerosol precursor decay including $\alpha$-pinene and NO. For comparison, simulations were also conducted using established

gas-phase chemical mechanisms that included RACM2 and the MCM to evaluate the treatment of $\alpha$-pinene oxidation and oxidation chemistry across different chemical frameworks for a range of experimental conditions. Model sensitivity tests were then conducted to assess the impact of key physical parameters that included chamber dilution rate and wall loss rates on simulated $\Delta^{17}O$ values. Finally, we compare the modeled $\Delta^{17}O$ values of $NO_y$ species with experimental observations to evaluate the model's ability to reproduce isotopic values under different photochemical and nighttime oxidation conditions.

### 3.3.1 Precursor Decay

Box model simulations were conducted to evaluate the oxidation and decay of precursors used in the experiments, ensuring that the correct amount of oxidant was accurately simulated. The MCM, RACM2, and $NO_x$-API chemical mechanisms were used to simulate the decay of $\alpha$-pinene and NO, with results compared against experimental observations (Fig. 4-5). Model

performance was evaluated by comparing measured and modeled concentrations of NO and $\alpha$-pinene using one-to-one plots, with corresponding $R^2$ values and quantification of model biases (Figs. S4–S5) and Table 4 summarizes these results. Overall, the $NO_x$-API mechanism provided improved model performance, evidenced consistently higher $R^2$ values (averaging 0.97 $\pm$0.03) and lower absolute residuals for both $\alpha$-pinene and NO decay compared to the other mechanisms. We attribute this enhanced model performance to the simplified treatment of higher-generation products in the $NO_x$-API mechanism. Unlike

MCM and RACM2, which allow continued gas-phase reactions of all products through extensive reaction propagation, the $NO_x$-API mechanism terminates the chemistry of these products after a limited number of steps. Given that the box-model simulations do not include an explicit aerosol-phase treatment, continued gas-phase reactivity of condensable species (as implemented in MCM and RACM2) may unrealistically disrupt the oxidant budget. Therefore, the $NO_x$-API mechanism as employed in the box model is better aligned with the experimental design and will be used for subsequent $\Delta^{17}O$ simulations of

the chamber experiments.

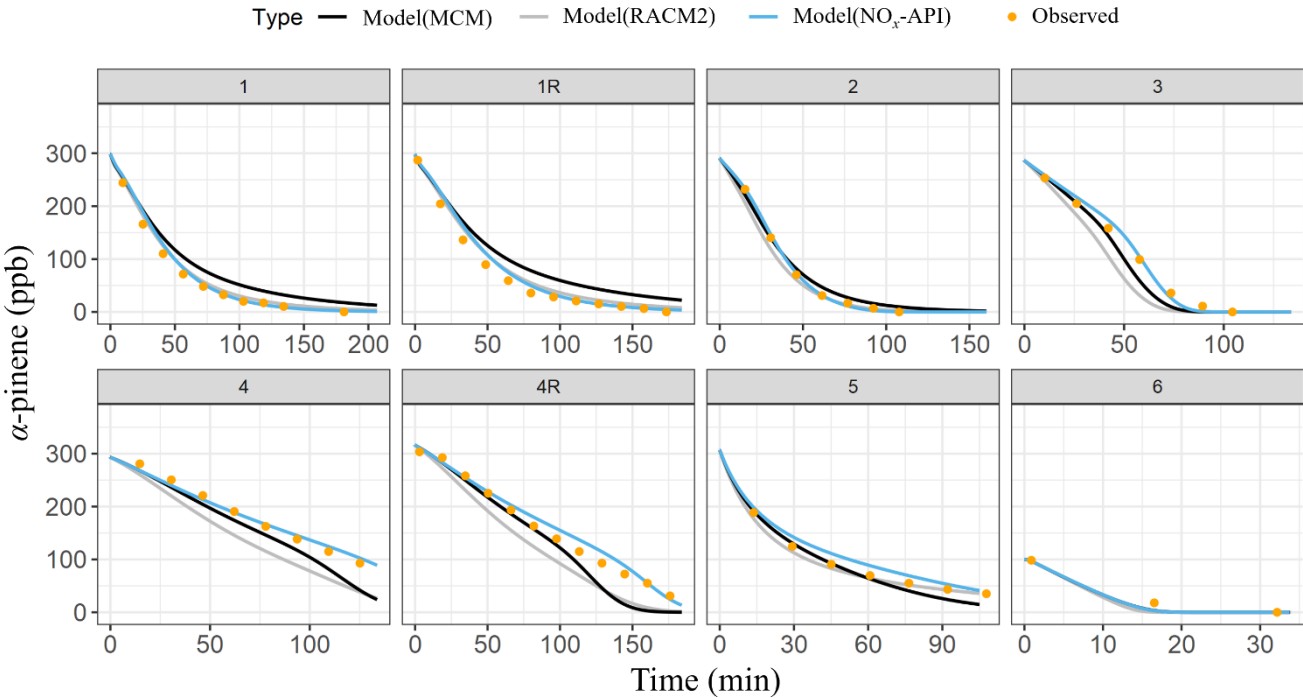

**Fig. 4. The observed (orange data points) and the modeled (lines) *α*-pinene decay for the various conducted experiments. The modeled results are based on three chemical mechanisms: MCM (black), RACM2 (grey), and NO$_x$–API (light blue).**

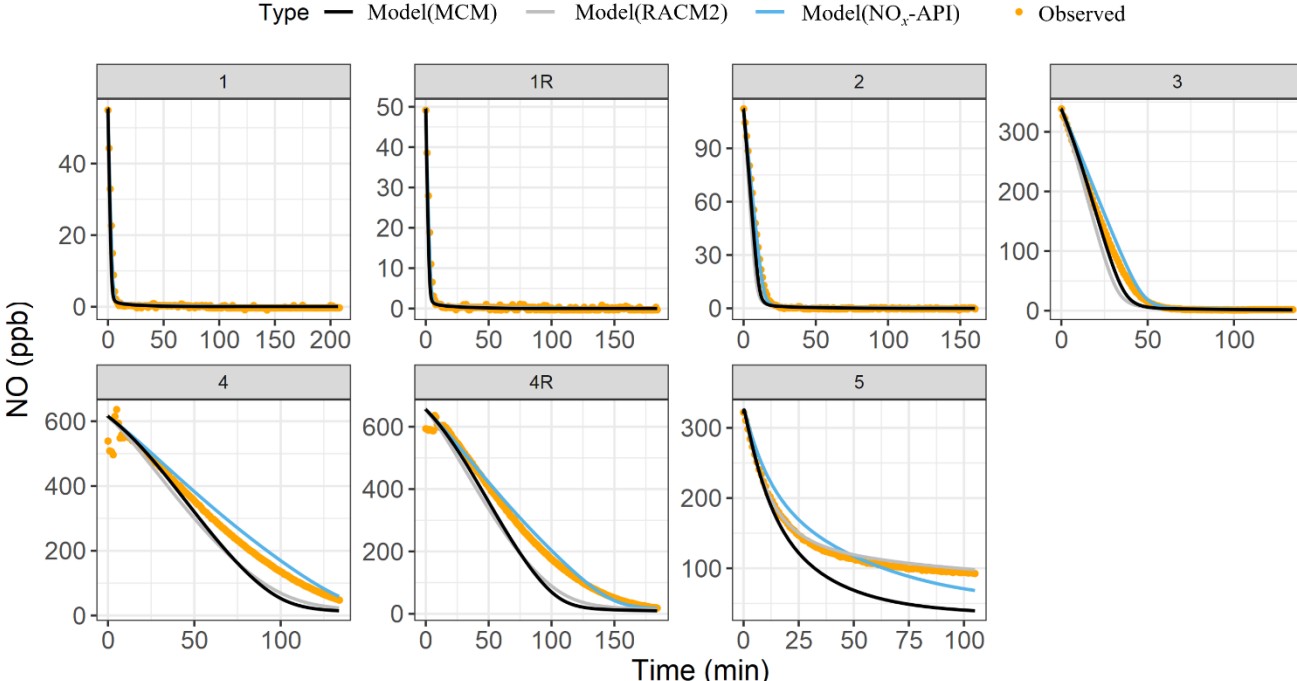

**Fig. 5. The observed (orange data points) and the modeled (lines) NO decay for the various conducted photochemical experiments. The modeled results are based on three chemical mechanisms: MCM (black), RACM2 (grey), and NO$_x$–API (light blue). Experiment 6 (nighttime oxidation) was omitted from the analysis as NO was not among the initial reactants.**

**Table 4. Summary of model performance for $\alpha$-pinene and NO using the NO$_x$-API, RACM2, and MCM mechanisms. Values represent the average coefficient of determination ($R^2$) and average mean residuals (in ppb) across all experiments, with associated standard deviations.**

| Mechanism | $\alpha$-pinene | | NO | |
|---|---|---|---|---|
| | Average $R^2$ | Average Mean Residual (ppb) | Average $R^2$ | Average Mean Residual (ppb) |
| NO$_x$-API | $0.97 \pm 0.03$ | $8.6 \pm 3.8$ | $0.97 \pm 0.03$ | $9.6 \pm 10.7$ |
| RACM2 | $0.89 \pm 0.16$ | $17.7 \pm 13.5$ | $0.92 \pm 0.03$ | $17.6 \pm 23.9$ |
| MCM | $0.84 \pm 0.19$ | $21.7 \pm 12.9$ | $0.86 \pm 0.21$ | $21.2 \pm 24.4$ |

### 3.3.2 $\Delta^{17}O$ Model Sensitivity to Dilution and Wall Loss

Before comparing modeled $\Delta^{17}O$ values to observations, we first evaluated the model's sensitivity to our treatment of key physical parameters, including chamber dilution and wall loss, and their potential impact on the simulated $\Delta^{17}O$ (Fig. 6). These sensitivity tests focused on Experiment 1, one of the longest-duration experiment, where dilution and wall loss would be expected to exert the strongest influence on gas-phase chemistry and $\Delta^{17}O$ values.

The impact of chamber dilution was assessed using four scenarios: no dilution, and first-order dilution rate constants ($k_{dil}$) of $1 \times 10^{-5}$, $5 \times 10^{-5}$, and $1 \times 10^{-4}$ s$^{-1}$ (Fig. 6A). Dilution was initiated at $t = 208$ minutes in the model to match the experimental protocol. Across all scenarios, simulated $\Delta^{17}O$ of the total organic nitrate (ONIT = ONITa + ONITb + ONITc + ONITOOHa + ONITOOHb + DIMER + PDN) values were minimally impacted by dilution. For instance, $\Delta^{17}O$(ONIT) varied by only 0.06 ‰ between the no-dilution and highest dilution rate constant scenario ($k_{dil} = 1 \times 10^{-4}$ s$^{-1}$), corresponding to a relative difference of −1.1 %. This insensitivity reflects the fact that organic nitrate chemistry was largely completed by the time dilution began. In contrast, simulated $\Delta^{17}O$ values for HNO$_3$ and NO$_2$ were more sensitive to the dilution rate. Between the no-dilution and lowest dilution rate constant scenario ($k_{dil} = 1 \times 10^{-5}$ s$^{-1}$), $\Delta^{17}O$(HNO$_3$) and $\Delta^{17}O$(NO$_2$) changed by only -0.01 ‰ (corresponding to a relative difference of −0.2 %) and −0.11 ‰ (-0.7 %), respectively. However, increasing the dilution rate constant from $1 \times 10^{-5}$ to $5 \times 10^{-5}$ s$^{-1}$ led to additional decreases of 0.32 ‰ (−5.0 %) for HNO$_3$ and 0.88 ‰ (−5.6 %) for NO$_2$. The most extreme dilution rate constant scenario ($k_{dil} = 1 \times 10^{-4}$ s$^{-1}$) reduced $\Delta^{17}O$ by a total of 0.50 ‰ (−7.9 %) for HNO$_3$ and 1.62 ‰ (−10.3 %) for NO$_2$ relative to the no-dilution case. For the main box model simulations, we adopted a dilution rate constant of $3.47 \times 10^{-5}$ s$^{-1}$, corresponding to a measured flow rate of 25 LPM in a 12 m$^3$ chamber. Assuming no more than ±20 % uncertainty in the actual flow rate, we estimate that the resulting uncertainty in simulated $\Delta^{17}O$ values due to dilution would be approximately ±2.5 % for HNO$_3$, ±3 % for NO$_2$, and less than ±1 % for ONIT. These estimates are based on the observed $\Delta^{17}O$ variation between the $1 \times 10^{-5}$ and $5 \times 10^{-5}$ s$^{-1}$ scenarios.

Next, we evaluated the potential influence of chamber wall loss on the simulated $\Delta^{17}O$ values (Fig. 6B). Three scenarios were tested: (1) no wall loss, (2) a wall loss scenario incorporating NO$_y$ and O$_3$ loss rates from previous studies (Morales et al., 2021; Wang et al., 2014), and (3) an extreme case in which wall loss rate constants were increased by a factor of 10 (Table S7). In the base comparison between the no-wall-loss and reported wall loss scenario, the effect on modeled $\Delta^{17}O$ values was minimal. Specifically, $\Delta^{17}O$ changed by -0.02 ‰ for HNO$_3$ (-0.3 %), -0.37 ‰ for NO$_2$ (-2.3 %), and +0.03 ‰ for ONIT (+0.5 %). These small differences suggest that moderate wall loss rates, consistent with literature values, would not substantially alter the $\Delta^{17}O$ simulations. However, the extreme wall loss scenario revealed a much stronger impact. In this case, $\Delta^{17}O$ decreased by 2.5 ‰ for NO$_2$ (-15.8 %) and increased by 0.50 ‰ for HNO$_3$ (+7.8 %) and 0.69 ‰ for ONIT (+11.4 %) relative to the no-wall-loss case. The drop in $\Delta^{17}O$(NO$_2$) arises from the altered oxidant concentrations and rapid photochemical cycling of NO$_x$, which can reset its $\Delta^{17}O$ values on short timescales. Conversely, the rise in $\Delta^{17}O$ for HNO$_3$ and ONIT reflects preferential removal of early-formed products with lower $\Delta^{17}O$, allowing later-formed, products with higher $\Delta^{17}O$ values to

dominate. Because we lacked experimental constraints on wall loss during the chamber experiments, the box model simulations presented in this study did not include wall loss. Based on our sensitivity analysis, omitting wall loss introduces an estimated uncertainty of $\pm0.3$ % for $HNO_3$, $\pm2.3$ % for $NO_2$, and $\pm0.5$ % for ONIT assuming moderate wall loss scenario. If actual wall loss rates in the chamber were significantly higher than those reported in the literature, the model is anticipated to overestimate $\Delta^{17}O$ in $NO_2$ and underestimate it in $HNO_3$ and ONIT.

Overall, based on the results of the model sensitivity tests, we estimate that the propagated uncertainty in the simulated $\Delta^{17}O$ values due to chamber dilution and wall loss is approximately $\pm2.5$ % for $HNO_3$, $\pm3.8$ % for $NO_2$, and $\pm1.1$ % for ONIT. These values reflect the combined effects of plausible variation in dilution rate ($\pm20$ %) and the difference in simulated $\Delta^{17}O$ values from literature-based estimates of $NO_y$ and $O_3$ wall loss compared to a case of no wall loss. Further, considering the uncertainty in the measured $\Delta^{17}O(O_3^{term})$ is approximately $\pm5$ % (Vicars and Savarino, 2014), we calculate an overall model uncertainty of $\Delta^{17}O$ of $\pm5.6$ %, $\pm6.3$ %, and $\pm5.1$ % for $HNO_3$, $NO_2$, and ONIT, respectively. Clearly, the uncertainty in the $\Delta^{17}O(O_3^{term})$ is expected to be the largest source of uncertainty in the modeled $\Delta^{17}O$ values as opposed to our treatment of dilution and chamber wall loss. Still, we conservatively apply these propagated uncertainty estimates when presenting and interpreting the model results.

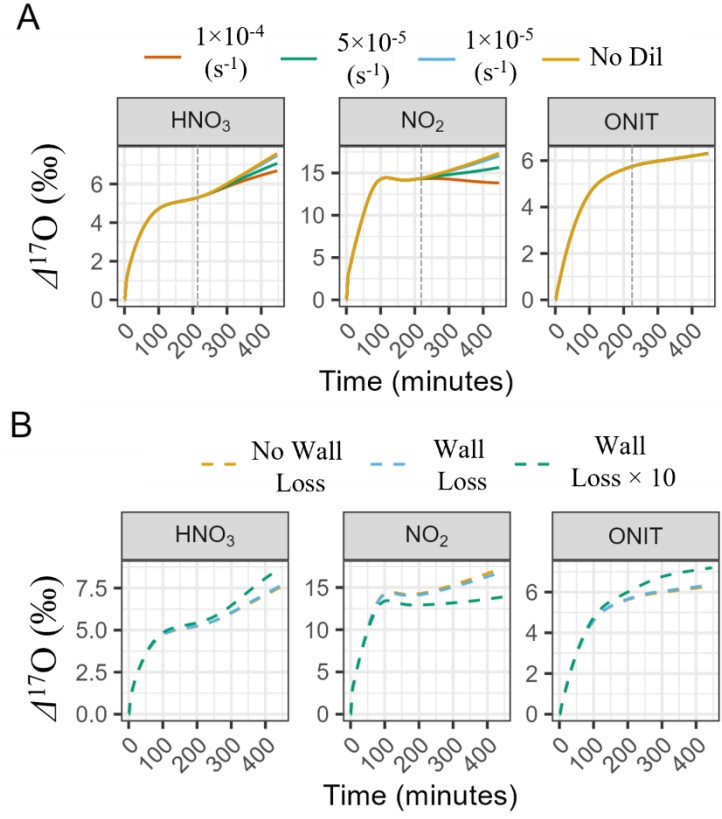

**Fig. 6.** Modeled $\Delta^{17}O$ sensitivity to physical chamber parameters. (A) Model simulations testing the sensitivity of $\Delta^{17}O$ in $NO_2$, $HNO_3$, and ONIT to chamber dilution rates, including scenarios with no dilution and with first-order dilution constants of $1 \times 10^{-5}$, $5 \times 10^{-5}$, and $1 \times 10^{-4}$ $s^{-1}$. (B) Model simulations testing the impact of wall loss, comparing a no-wall-loss case, a wall loss scenario using reaction rates from (Wang et al., 2014) and (Morales et al., 2021), and a high-loss case in which wall loss rates were increased by a factor of 10. All simulations were performed under the conditions of Experiment 1. The gray dashed line in A corresponds to the start of chamber dilution.

### 3.3.3 $\Delta^{17}O$ Base Model Simulations

Building on the results of the sensitivity analyses, we next evaluate the performance of the base box model in simulating the $\Delta^{17}O$ values of key $NO_y$ species, including $NO_2$, $HNO_3$, and ONIT under the various experimental conditions. These simulations incorporate the best-estimate physical parameters (e.g., dilution rate) and chemical mechanism inputs, including a representative $\Delta^{17}O(O_3^{term})$ value of $39 \pm 2$ ‰ (Fig. 7). Overall, the model simulations for the photochemical experiments indicate a substantial temporal change in $\Delta^{17}O$ for all considered $NO_y$ compounds. The $\Delta^{17}O(NO_2)$ initially starts at 0 ‰ and begins to increase due to the production of $O_3$ that elevate $\Delta^{17}O(NO_2)$ as NO is oxidized by $O_3$. For the nighttime experiment, the box model predicts $\Delta^{17}O(NO_2)$ to remain steady with a value near 7.6 ‰, due to $N_2O_5$ thermal equilibrium with $NO_2$ and

NO$_3$ resulting in the $\Delta^{17}O(NO_2) \approx \Delta^{17}O(N_2O_5)$. Generally, the $\Delta^{17}O$ simulation of NO$_2$ were in excellent agreement with the

690 observations, as indicated by an average model bias of 0.9 ± 2.4 ‰ ($n = 8$; Table 5) and a strong correlation indicated by a regression $R^2$ value of 0.98 . This strong agreement indicates that the box model and employed chemical mechanism accurately simulated NO$_x$ photochemical cycling and NO$_2$/NO$_3$/N$_2$O$_5$ thermal equilibrium. The simulation of $\Delta^{17}O(NO_2)$ is inherently sensitive to the assumed value of $\Delta^{17}O(O_3^{term})$, which represents the isotopic signature transferred from the terminal O atom of O$_3$ to NO$_2$ during oxidation. The relationship between measured $\Delta^{17}O(NO_2)$ and the modeled fraction of O atoms in NO$_2$

deriving from O$_3$ (denoted as $f(Q)$) indicates that $\Delta^{17}O(O_3^{term})$ is 39.4 ± 0.6 ‰ (Fig. S6), which is in excellent agreement with recent independent measurements of tropospheric O$_3$ reporting values of 39 ± 2 ‰ (Ishino et al., 2017; Vicars and Savarino, 2014).

The measured $\Delta^{17}O(pNO_3)$ was compared with the simulated $\Delta^{17}O$ of (ONIT), based on our understanding that pNO$_3$ was

700 apparently dominated by RONO$_2$ contributions. The temporal evolution of the simulated $\Delta^{17}O(ONIT)$ closely followed that of $\Delta^{17}O(NO_2)$ but remained lower due to dilution effects associated with the dominant formation pathway of organic nitrates via NO + RO$_2$ reactions during the photochemical experiments, resulting in ONIT dominated by ONITa, ONITb, and ONITc compounds (Fig. S3). During nighttime oxidation experiments, ONIT formation primarily proceeded via $\alpha$-pinene + NO$_3$ reactions, leading to ONIT with higher contribution of DIMER and PDN compounds. Due to NO$_2$/NO$_3$/N$_2$O$_5$ thermal

equilibrium that resulted in $\Delta^{17}O(NO_2) \approx \Delta^{17}O(NO_3) \approx \Delta^{17}O(N_2O_5)$, the simulated nighttime $\Delta^{17}O(ONIT)$ values were approximately equal to the simulated $\Delta^{17}O(NO_2)$. The simulated $\Delta^{17}O(ONIT)$ closely matched the $\Delta^{17}O(pNO_3)$ observations with an average bias of -1.4 ± 2.4 ‰ ($n =7$), indicating that the relative production routes of organic nitrate (RO$_2$ + NO vs BVOC + NO$_3$; Table 1) were correctly simulated for the various experimental conditions. The overall correlation was moderate, with $R^2 = 0.55$.

The bias for simulated $\Delta^{17}O(ONIT)$ compared to the $\Delta^{17}O(pNO_3)$ observations was within 1.4 ‰ for all experiments except Exp. 5, where a substantially higher bias of -7.0 ‰ was observed. Excluding this outlier increased the correlation to $R^2 = 0.97$. Based on Cook's Distance and the Studentized Residual, Experiment 5 was identified as an outlier in the linear regression analysis. The larger model-data difference for Exp. 5 may reflect different oxidation dynamics or, more likely, uncertainty in

the extraction of pNO$_3$. Specifically, Experiment 5 yielded the lowest Filter/AMS ratio among the photochemical experiments, with the ratio falling below the quantitative range when accounting for propagated uncertainty. This suggests potential under-recovery or sampling artifacts, indicating that the measured $\Delta^{17}O(pNO_3)$ in this experiment may not accurately represent the pNO$_3$ formed during the chamber experiment (Table 3). Thus, the larger model-measurement $\Delta^{17}O(ONIT)$ disagreement for Exp. 5 likely reflects uncertainty in the extracted NO$_3^-$ rather than model misrepresentation of ONIT formation pathways. The

measured $\Delta^{17}O(pNO_3)$ was also evaluated relative to the modeled $f(Q)$ for ONIT formation. A linear regression constrained through the origin, excluding Exp. 5, yielded a slope of 41.7 ± 1.2 ‰ ($R^2 = 0.996$; Fig. S7). This derived $\Delta^{17}O(Q)$ value is in

close agreement with the assumed $\Delta^{17}O(O_3{}^{term})$ value of 39 ± 2 ‰, and is consistent with the value determined from the measured $\Delta^{17}O(NO_2)$ comparison with the modeled $f(Q)$ of $NO_2$.

The simulated dynamics of $\Delta^{17}O(HNO_3)$ closely follow the temporal evolution of $\Delta^{17}O(NO_2)$, initially starting at 0 ‰ and increasing as $O_3$ production occurs in the chamber during the photochemical experiments (i.e., Exp 1-5) (Fig. 7). The simulated $\Delta^{17}O(HNO_3)$ for the photochemical experiments was always lower than $\Delta^{17}O(NO_2)$ for the photochemical experiments, as the model simulation predicts $HNO_3$ is predominantly produced through the $NO_2 + OH$ pathway (Fig. S8). Based on the conventional oxygen isotope mass-balance calculations, this pathway results in a dilution factor of 2/3 relative to $\Delta^{17}O(NO_2)$

(Table 1). For the nighttime experiment (Exp. 6), $\Delta^{17}O(HNO_3)$ is predicted to be slightly lower than $\Delta^{17}O(NO_2)$, primarily due to the contributions from two formation pathways: $NO_3 + $ pinonaldehyde and $N_2O_5 + H_2O(g)$, representing approximately 60 % and 40 % of nighttime $HNO_3$ production, respectively. Since $\Delta^{17}O(NO_3) \approx \Delta^{17}O(NO_2)$ under nighttime conditions due to rapid thermal equilibrium, the $NO_3 + $ pinonaldehyde should result in $\Delta^{17}O(HNO_3) \approx \Delta^{17}O(NO_2)$. However, the $N_2O_5$ hydrolysis pathway incorporates one oxygen atom from water, which has $\Delta^{17}O \approx 0$ ‰, resulting in an effective dilution of the product

$\Delta^{17}O(HNO_3)$ relative to $\Delta^{17}O(NO_2)$. Overall, the model exhibits poor agreement with observed $\Delta^{17}O(HNO_3)$ values with a model bias that ranged from -11.5 to 7.9 ‰ (Table 4) and a weak correlation ($R^2 = 0.39$). The model underpredicts $\Delta^{17}O(HNO_3)$ for the low-$NO_x$ photochemical experiments (Exp. 1 and 2) and the nighttime oxidation experiment (Exp. 6), but overpredicts $\Delta^{17}O(HNO_3)$ for the high-$NO_x$ photochemical experiments (Exp. 3, 4, and 5). If the isotope mass-balance assumptions are correct, this pattern suggests that a missing or underrepresented source of high-$\Delta^{17}O$ $HNO_3$ may exist under low-$NO_x$

conditions, and a missing low-$\Delta^{17}O$ $HNO_3$ source may exist under high-$NO_x$ conditions.

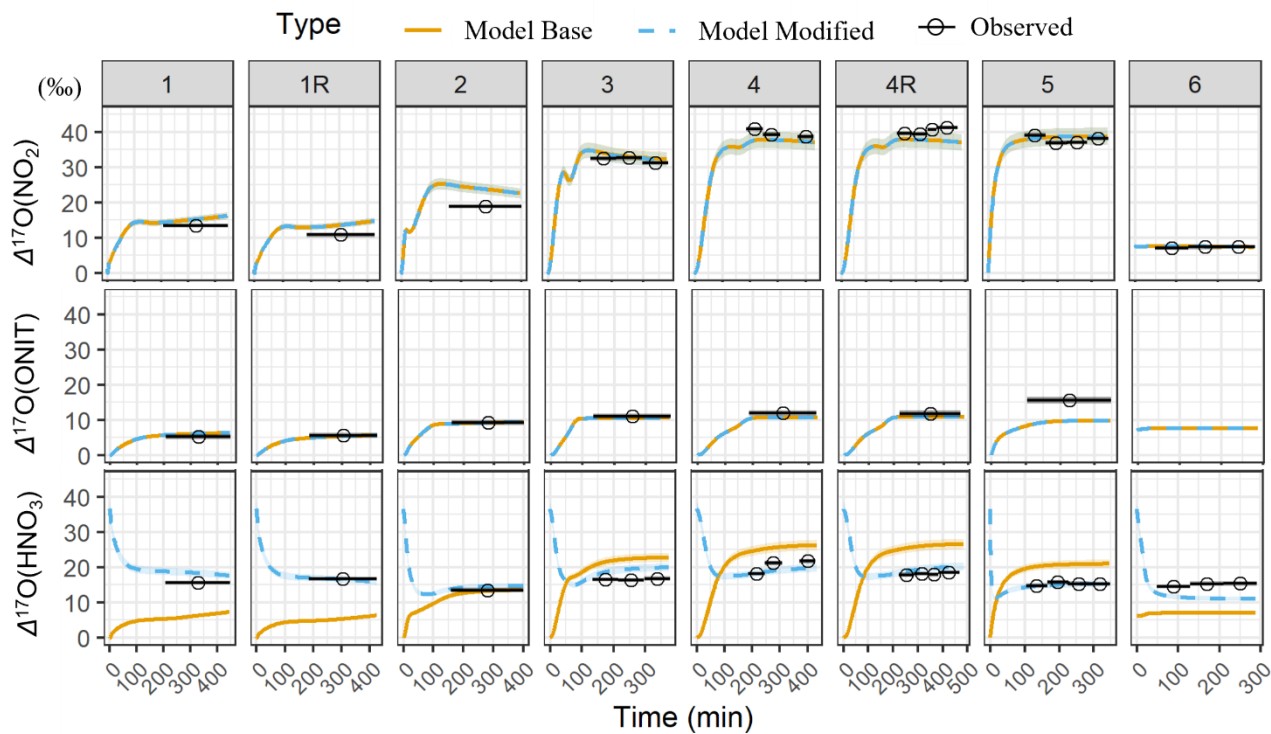

**Fig. 7.** Comparison between the modeled (base and modified) and observed $\Delta^{17}O(NO_2)$, $\Delta^{17}O(ONIT)$, and $\Delta^{17}O(HNO_3)$ values sorted by the various conducted experiments. The data points represent the average time of each sample collection, and the black line spans the duration of the collection period. The measurement uncertainty ($\pm\sigma$) is included as the shaded region. The modeled $\Delta^{17}O(ONIT)$ is compared to the observed $\Delta^{17}O(pNO_3)$.

**Table 5.** Summary of the calculated average bias ($\Delta^{17}O(Model) - \Delta^{17}O(Observed)$) for the $\Delta^{17}O$ model simulations using the $NO_x$-API base and modified (Mod) mechanisms. For experiments with multiple observations, the bias is reported as the mean ± standard deviation.

| Exp | NO₂ | | | ONIT | | | HNO₃ | | |
| --- | --- | --- | --- | --- | --- | --- | --- | --- | --- |
| | **Base** | **Mod** | ***n*** | **Base** | **Mod** | ***n*** | **Base** | **Mod** | ***n*** |
| 1 | 1.7 | 1.7 | 1 | 0.8 | 0.8 | 1 | -9.4 | 2.7 | 1 |
| 1R | 2.6 | 2.6 | 1 | -0.4 | -0.4 | 1 | -11.5 | 0.0 | 1 |
| 2 | 5.3 | 5.3 | 1 | -0.1 | -0.1 | 1 | -0.7 | 0.8 | 1 |
| 3 | $1.2 \pm 0.7$ | $1.1 \pm 0.8$ | 3 | -0.7 | -0.7 | 1 | $5.5 \pm 0.7$ | $2.7 \pm 0.7$ | 3 |

| 4 | -2.1 ± 0.9 | -2.1 ± 0.9 | 3 | -1.4 | -1.4 | 1 | 5.2 ± 1.1 | -1.4 ± 1.1 | 3 |
|---|---|---|---|---|---|---|---|---|---|
| 4R | -2.6 ± 0.7 | -2.6 ± 0.9 | 4 | -1.0 | -1.0 | 1 | 7.9 ± 0.3 | 1.4 ± 0.3 | 4 |
| 5 | 0.7 ± 1.1 | 0.7 ± 1.3 | 4 | -7.0 | -7.0 | 1 | 5.3 ± 0.3 | -0.3 ± 0.3 | 4 |
| 6 | 0.2 ± 0.2 | 0.2 ± 0.2 | 3 | NA | NA | NA | -8.0 ± 0.4 | -3.7 ± 0.7 | 3 |

### 3.3.4 $\Delta^{17}O(HNO_3)$ Model Sensitivity Tests

To investigate the causes of the observed discrepancies between modeled and measured $\Delta^{17}O(HNO_3)$ values, we conducted a series of model sensitivity tests focused on modifying $HNO_3$ formation pathways. These tests aimed to assess whether

adjustments to reaction pathways or rate constants could reconcile the overprediction of $\Delta^{17}O(HNO_3)$ during the high-$NO_x$ photochemical experiments (Exp. 3, 4, and 5) and the underprediction during low-$NO_x$ and nighttime experiments. The following sections describe these targeted evaluations and their implications for understanding the $\Delta^{17}O(HNO_3)$ produced under different chemical regimes.

We first investigated the potential causes of the model overprediction in $\Delta^{17}O(HNO_3)$ during the high-$NO_x$ photochemical experiments (Exp. 3, 4, and 5). Two alternative $HNO_3$ production pathways were identified that could yield lower $\Delta^{17}O(HNO_3)$ values than the assumed predominant daytime production pathway involving $NO_2 + OH$ reactions based on oxygen isotope mass-balance assumptions (Table 1). One such pathway is the reaction of NO with $HO_2$, which can form $HNO_3$ with a $\Delta^{17}O$ transfer factor of $(1/3)\Delta^{17}O(NO)$ (Alexander et al., 2020; Table 1). However, an underestimation of this pathway in the model

is unlikely to account for the observed model–measurement discrepancy across all experiments. For instance, $HO_2$ concentrations in Exp. 5 are expected to be low due to the absence of significant amount of $HO_2$ precursors, unlike Exp. 3 and 4, which had elevated $HO_2$ precursors from the initial injected $H_2O_2$. Moreover, Exp. 1 and 2, which had the highest initial $H_2O_2$ concentrations and thus the greatest potential for $HO_2$ formation, exhibited model $\Delta^{17}O(HNO_3)$ values that were too low, further suggesting that this pathway cannot explain the observed discrepancies. Still, as a sensitivity test, we increased the rate

constant for the $NO + HO_2 \rightarrow HNO_3$ reaction (R027 in the $NO_x$-API mechanism; Table S3) by an order of magnitude (Fig. S9). Despite this adjustment in the $NO + HO_2 \rightarrow HNO_3$ reaction rate constant, the impact on modeled $\Delta^{17}O(HNO_3)$ remained small with a change in $\Delta^{17}O(HNO_3)$ of less than 1.6 ‰, and as little as 0.3 ‰ for Exp. 5. This change in model $\Delta^{17}O(HNO_3)$ is too small to account for the observed model bias of 5.2 to 7.9 ‰ across Exp. 3 to 5 (Table 4).

Next, the hydrolysis of organic nitrates to $HNO_3$ was evaluated as a potential low-$\Delta^{17}O(HNO_3)$ source, since hydrolysis of photochemically produced $RONO_2$ ($R + OH/O_2/NO$) could produce $HNO_3$ with a $\Delta^{17}O$ transfer factor of $(1/3)(\Delta^{17}O(NO))$ (Table 1) considering that $\Delta^{17}O(RO_2)$ is presumably 0 ‰. Based on simulation results (Fig. S10), 21.8 %, 14.8 %, and 13.4 % of the total $NO_y$ was present as $HNO_3$ for Exp. 3, 4, and 5, respectively, compared to only 11.0 %, 6.8 %, and 4.0 % as

RONO₂ (defined as ONIT in our model simulations; ONIT = ONITa + ONITb + ONITc + ONITOOHa + ONITOOHb + PDN + DIMER). Nonetheless, as a sensitivity test, we added ONIT hydrolysis with an assumed lifetime of 30 minutes (Table S8), consistent with recent experimental determinations (Takeuchi and Ng, 2019); however, this study showed that only 23–32 % of organic nitrates derived from $\alpha$-pinene + OH and 9–17% from $\alpha$-pinene + NO₃ are susceptible to hydrolysis (Takeuchi and Ng, 2019), suggesting that this sensitivity test should overestimate the impact of ONIT hydrolysis as a production route for HNO₃. The inclusion of ONIT hydrolysis in the model reduced $\Delta^{17}O(HNO_3)$ by 1.4–3.4 ‰ relative to the base model, partially improving agreement with observations (Fig. S11). However, even under the unrealistic assumption of complete organic nitrate hydrolysis, the bias between the model and observed $\Delta^{17}O(HNO_3)$ values remained between 1.7 ‰ and 4.6 ‰ across experiments, indicating that hydrolysis alone cannot fully explain the model–observation discrepancy. Furthermore, while organic nitrate hydrolysis modestly improved the $\Delta^{17}O(HNO_3)$ comparison, it worsened the agreement for $\Delta^{17}O(ONIT)$ (Fig. S12). The bias for the simulated $\Delta^{17}O(ONIT)$ relative to the observed $\Delta^{17}O(pNO_3)$ increased substantially, ranging from 5.1 ‰ to 15.0 ‰. These results suggest that organic nitrate hydrolysis is unlikely to be the primary cause of the observed lower $\Delta^{17}O(HNO_3)$ values relative to model simulations.

 Overall, we were unable to identify a missing or underrepresented HNO₃ production pathway that could reconcile the observed $\Delta^{17}O(HNO_3)$ values within the bounds of the assumed $\Delta^{17}O$ mass-balance framework (Table 1). This suggests that current assumptions regarding oxygen atom transfer during HNO₃ formation may need to be revisited. For the high-NO$_x$ photochemical experiments (Exp. 3–5), plotting measured $\Delta^{17}O(HNO_3)$ against the modeled fraction of O₃-derived oxygen atoms ($f(Q)(HNO_3)$) gave a slope of 28.9 ± 0.5 ‰, representing the $\Delta^{17}O$ of the O₃$^{term}$ (Fig. S13). This slope is lower than the ~39-41 ‰ slope obtained from similar analyses of $\Delta^{17}O(NO_2)$ and $\Delta^{17}O(pNO_3)$ (Fig. S6-S7), which are more consistent with recent near-surface observations of $\Delta^{17}O(O_3^{term})$ (Ishino et al., 2017; Vicars and Savarino, 2014). The lower slope in the $f(Q)$ versus $\Delta^{17}O(HNO_3)$ relationship potentially implies that not all oxygen atoms from NO₂ are retained during HNO₃ formation in the dominant NO₂ + OH reaction under high-NO$_x$ photochemical conditions. Traditionally, it is assumed that two-thirds of the oxygen atoms in HNO₃ are inherited from NO₂ and one-third from OH (with $\Delta^{17}O \sim 0$ ‰) (Table 1), but this oxygen mass-balance may need adjustment. Adjusting the slope for $\Delta^{17}O(HNO_3)$ versus $f(Q)$ to match that for NO₂, would require to adjustment of the oxygen mass-balance in the NO₂ + OH reaction. Specifically, scaling the NO₂ contribution by ~0.74 lowers its O atom fractional contribution in HNO₃ from 66.7 % to ~49 %, resulting in an effective relationship of $\Delta^{17}O(HNO_3) \approx \frac{1}{2}\Delta^{17}O(NO_2)$. This deviation from the expected 2/3 to an effective 1/2 contribution of $\Delta^{17}O(NO_2)$ may reflect partial oxygen atom exchange or isotopic scrambling during the formation of an excited HNO₃ intermediate, prior to collisional stabilization. Such processes could result in the effective loss or redistribution of the oxygen atoms originally inherited from NO₂ in the HNO₃ product. Indeed, previous experimental studies using isotopically labeled ¹⁸OH in reactions with NO₂ have demonstrated that the O atoms in the HNO₃ reactive intermediate product can undergo rapid intramolecular scrambling (Donahue et al., 2001). While this specific mechanism alone cannot easily explain the observed loss of approximately one-sixth of the original NO₂-derived oxygen atoms in the final HNO₃ product, it does suggest that interesting

O isotope dynamics occur during the $NO_2$ + OH reaction. While the exact mechanism that could explain the redistribution of ~1/6 of O atoms derived from $NO_2$ in the $HNO_3$ product remains uncertain and warrants further investigation, we conducted a sensitivity test by updating the $NO_x$-API mechanism to reflect the potential modified mass balance for the $NO_2$ + OH reaction, assuming $\Delta^{17}O(HNO_3) = \frac{1}{2}\Delta^{17}O(NO_2)$ (Fig. S14). This adjustment substantially improved the model performance for Exp. 3-5, yielding biases that ranged from –2.6 ‰ to +1.0 ‰ relative to the observations.

Next, we examine the cause of the low model bias in $\Delta^{17}O(HNO_3)$ relative to observations during the nighttime oxidation experiment. In this experiment, the measured $\Delta^{17}O(HNO_3)$ exceeded that of $\Delta^{17}O(NO_2)$, implying greater $O_3$ incorporation into $HNO_3$ than is represented in the model. The dominant modeled pathway for $HNO_3$ formation at night was the reaction of $NO_3$ with pinaldehyde (Fig. S8). However, this route yields $\Delta^{17}O(HNO_3) \approx \Delta^{17}O(NO_3)$, because $NO_2$, $NO_3$, and $N_2O_5$ rapidly equilibrate thermally overnight leading to $\Delta^{17}O(NO_2) \approx \Delta^{17}O(NO_3) \approx \Delta^{17}O(N_2O_5)$. Therefore, this pathway alone cannot account for the elevated observed $\Delta^{17}O(HNO_3)$ relative to the model simulation. To explain this discrepancy, an additional source of high-$\Delta^{17}O$ external to the $NO_y$ reservoir must be involved. One possibility is that the heterogeneous uptake of $N_2O_5$ on aerosol surfaces could involve incorporation of an oxygen atom from $O_3$ rather than liquid water. In this case, based on our mass-balance framework (Table 1), the $\Delta^{17}O$ of $HNO_3$ formed via this pathway can be represented as:

$$\Delta^{17}O(HNO_3) = (5/6)\Delta^{17}O(N_2O_5) + (1/6)\Delta^{17}O(O_3^{term}) \qquad \text{(Eq. 8)}$$

Since $\Delta^{17}O(NO_2) \approx \Delta^{17}O(N_2O_5)$ for our nighttime oxidation experiment conditions, this expression simplifies to:

$$\Delta^{17}O(HNO_3) = (5/6)\Delta^{17}O(NO_2) + (1/6)\Delta^{17}O(O_3^{term}) \qquad \text{(Eq. 9)}$$

This mass-balance equation is similar to that assumed for $N_2O_5$ heterogenous reaction involving particulate $Cl^-$ (Table 1). While, we did not initiate our experiment with a $Cl^-$ source, we must have a source of $\Delta^{17}O(O_3^{term})$ external to $NO_x$ to explain the underestimate of simulated $\Delta^{17}O(HNO_3)$ relative to observations.

To test the potential impact of this mechanism, we incorporated $N_2O_5$ heterogeneous uptake into the model (Table S9). The uptake rate coefficient ($k_{het}$) was estimated assuming an aerosol seed number and volume concentration upon atomization of $2 \times 10^4$ cm$^{-3}$ and $2 \times 10^{10}$ nm$^3$ cm$^{-3}$, respectively, which was taken from previously reported values from similarly conducted $\alpha$-pinene + $NO_3$ nighttime experiments (Takeuchi and Ng, 2019), and an $N_2O_5$ uptake coefficient ($\gamma$) of $1.5 \times 10^{-4}$ for organic carbon with RH $\geq$ 30 % (Escorcia et al., 2010). Due to uncertainty in these parameters, we performed a sensitivity analysis across a range of $k_{het}$ values from $0.6 \times 10^{-4}$ to $6 \times 10^{-4}$ s$^{-1}$, corresponding to $N_2O_5$ lifetimes between approximately 0.46 and 4.6 hours. Model results show that increasing the $N_2O_5$ heterogeneous reaction rate systematically reduces the $\Delta^{17}O(HNO_3)$ model bias. Specifically, the bias decreased from 8.1 ‰ in the base case (no heterogeneous uptake) to 3.5 ‰ for the highest assumed $k_{het}$ (Fig. S15). This supports the hypothesis that $N_2O_5$ heterogeneous reactions incorporating $O_3$-derived oxygen

significantly influence the isotopic composition of HNO$_3$ produced under nighttime conditions. However, the highest assumed $k_{het}$ rate led to an increase in the overall model bias for $\Delta^{17}O(HNO_3)$ simulations the high-NO$_x$ photochemical experiments (Exp 3-5), shifting from $-0.8 \pm 1.5$ ‰ in the modified mass-balance simulation to $+1.8 \pm 2.5$ ‰ (Fig. S16). To avoid overcorrection, we selected a $k_{het}$ rate of $9.11 \times 10^{-5}$ s$^{-1}$, which improved agreement in the nighttime oxidation

experiments without negatively impacting the $\Delta^{17}O(HNO_3)$ predictions under high-NO$_x$ photochemical conditions.

Lastly, we sought to diagnose the potential underprediction of $\Delta^{17}O(HNO_3)$ in the model relative to observations from the low-NO$_x$ photochemical experiments (Exp. 1–2). We observed a relatively large amount of HNO$_3$ present before the experiments began, ranging from 3.2 to 5.1 ppb, which represented 20.1 %, 18.0 %, and 7.9 % of the maximum HNO$_3$

concentrations measured during Exp. 1, 1R, and 2, respectively (Table S10; Fig. S1). This suggested the possibility of a substantial chamber blank influencing the $\Delta^{17}O(HNO_3)$ measurements. Unfortunately, we did not directly quantify chamber blanks during these experiments. Thus, to evaluate the potential impact, we conducted a sensitivity analysis by introducing a new model tracer, HNO$_3^{blank}$, which was initialized using pre-experiment CIMS measurements. This tracer was treated identically to HNO$_3$ in the model, undergoing the same reactions and loss processes as HNO$_3$ (Table S11). We then used an

isotope mass-balance framework to infer the $\Delta^{17}O$ of the blank component required to reproduce the observed $\Delta^{17}O(HNO_3)$ (Eq. 10):

$$\Delta^{17}O(HNO_3^{blank}) = \frac{\Delta^{17}O(HNO_3^{obs}) - (1 - f_{blank})\Delta^{17}O(HNO_3^{prod})}{f_{blank}} \qquad \text{(Eq. 10)}$$

where $\Delta^{17}O(HNO_3^{obs})$ is the observed value, $\Delta^{17}O(HNO_3^{prod})$ is the value for photochemically produced HNO$_3$ in the model,

and $f_{blank}$ is the fractional contribution of the chamber blank from the model simulation:

$$f_{blank} = \frac{[HNO_3^{blank}]}{[HNO_3^{blank}] + [HNO_3^{prod}]} \qquad \text{(Eq. 11)}$$

Using this approach, we calculated an average $\Delta^{17}O(HNO_3^{blank})$ of $36.1 \pm 4.0$ ‰ ($n = 3$). While elevated, this value is plausible if HNO$_3$ formation occurred predominantly through nighttime N$_2$O$_5$ heterogeneous uptake under dark chamber conditions prior to the start of the experiment, and if the precursor NO$_2$ had a high $\Delta^{17}O$ value. Such conditions could arise

from residual NO$_2$ in the chamber either from previous experiments or background air that underwent oxidation while the chamber remained dark and inactive before the conducted experiments were initiated. It is also important to note that the CIMS HNO$_3$ measurements are subject to a relatively large uncertainty of approximately $\pm 20$ ‰, which may further influence the inferred blank correction.

Overall, to improve the accuracy of simulated $\Delta^{17}O(HNO_3)$, we implemented a series of modifications to the $NO_x$-API mechanism based on the conducted sensitivity tests, term $NO_x$-API (Modified). First, we revised the oxygen mass balance for $\Delta^{17}O(HNO_3)$ formation via the $NO_2 + OH$ reaction, to $(1/2)\Delta^{17}O(NO_2)$. Second, we included a chamber-derived $HNO_3$ background using a fixed $\Delta^{17}O$ value of 36.1 ‰. This background was initialized in all simulations based on CIMS-derived $HNO_3$ concentrations prior to photolysis onset (Table S10). Finally, we added a heterogeneous $N_2O_5$ hydrolysis pathway with a first-order loss rate of $9.11 \times 10^{-5}$ s$^{-1}$, using a $\Delta^{17}O(HNO_3)$ formation mass balance of $(5/6) \times \Delta^{17}O(N_2O_5) + (1/6) \times \Delta^{17}O(O_3{}^{term})$. As shown in Fig. 7, while these modifications had a minor effect on $\Delta^{17}O(NO_2)$ and $\Delta^{17}O(ONIT)$, in which the base $NO_x$-API mechanism were already in strong agreement with the observations, they substantially improved the model's performance for $\Delta^{17}O(HNO_3)$. The average absolute bias across all experiments decreased from $6.7 \pm 3.3$ ‰ in the base mechanism to $1.6 \pm 1.3$ ‰ in the modified mechanism (Table 5) and with an improved correlation ($R^2 = 0.48$). These results demonstrate the difficulty of accurately simulating $\Delta^{17}O(HNO_3)$ across diverse experimental conditions, highlighting the need for future experiments that more directly constrain oxygen isotope mass-balance assumptions in $HNO_3$ formation pathways. At the same time, careful attention to $HNO_3$ collection methods and blank corrections is essential to ensure meaningful comparisons between models and observations.

## 5. Conclusion

This study presents the first chamber experiments combining comprehensive $NO_y$ and $\alpha$-pinene oxidation chemistry with stable isotope constraints. Using a suite of observations and box model simulations, we demonstrate how multi-isotope analyses of $\Delta^{17}O$, $\delta^{18}O$, and $\delta^{15}N$ can yield novel insights into atmospheric oxidation pathways, including the formation and transformation of gas-phase and $pNO_3$ species. We observed a strong linear relationship between $\Delta^{17}O$ and $\delta^{18}O$ values across all collected $NO_y$ species. We derived a $\delta^{18}O$ value of $106 \pm 5$ ‰ for oxygen atoms transferred into $NO_y$ from $O_3{}^{term}$, and a value of $11.1 \pm 1.0$ ‰ for oxygen atoms transferred from other oxidants with an assumed $\Delta^{17}O \sim 0$ ‰, such as $RO_2$, $HO_2$, and $OH$. These results provide a new isotopic constraint for disentangling multi-oxidant systems in both chamber and ambient settings. Nitrogen isotope fractionation was evaluated from the observations, which indicated $\Delta\delta^{15}N(HNO_3 - NO_2)$ values were generally enriched, while $\Delta\delta^{15}N(pNO_3 - NO_2)$ values were negative, reflecting preferential $^{15}N$ incorporation into $HNO_3$, and $^{14}N$ incorporation into $pNO_3$. However, large differences in the observed $\Delta\delta^{15}N(HNO_3 - NO_2)$ between $H_2O_2/NO$ ($33.7 \pm 7.1$ ‰) and HONO ($0.4 \pm 1.7$ ‰) photochemical experiments remain unresolved, particularly since these experiments should have produced $HNO_3$ from a similar pathway, namely the $NO_2 + OH$ reaction. These discrepancies make it challenging for us to recommend a fractionation value associated with the $NO_2 + OH$ reaction and indicates the need for future targeted studies of the isotope effects during the $NO_2 + OH$ reaction. Importantly, isotope observations revealed stark differences between $HNO_3$

(medium-high $\Delta^{17}O$ and high $\delta^{15}N$) and pNO$_3$ (low $\Delta^{17}O$ and low $\delta^{15}N$), with the later predominately derived from organic nitrate. These isotope differences between HNO$_3$ and organic nitrate could serve as a useful qualitative constraint to evaluate inorganic and organic nitrate contributions to HNO$_3$ and pNO$_3$ budgets.

Model simulations of $\Delta^{17}O$ were conducted and indicated our model captured $\Delta^{17}O(NO_2)$ and $\Delta^{17}O(ONIT)$ (compared with observed $\Delta^{17}O(pNO_3)$ values) well. Simulating $\Delta^{17}O(HNO_3)$, however, proved more challenging. The model tended to overestimate $\Delta^{17}O$ under high-NO$_x$ conditions, underestimate it in low-NO$_x$ experiments, and underpredict nighttime values. A series of sensitivity tests suggest this mismatch likely arises from multiple contributing factors, including potential HNO$_3$ measurement biases (e.g., chamber blank), missing heterogeneous pathways (e.g., N$_2$O$_5$ hydrolysis), and the need to revisit

assumptions in the isotopic mass balance of the NO$_2$ + OH reaction. From these experiments, it is evident that we have a solid understanding of oxygen isotope transfer associated with NO$_x$ photochemical cycling, and the formation of organic nitrates. However, our understanding of the oxygen and nitrogen isotope dynamics of HNO$_3$ remains more uncertain. Our findings indicate the need for future experiments specifically designed to probe the formation pathways of HNO$_3$ and their associated isotope dynamics. This includes chamber studies that isolate individual pathways as well as targeted flow tube experiments.

Such efforts are essential to refine oxygen isotope mass-balance assumptions and are critical if $\Delta^{17}O$ and $\delta^{15}N$ is to be used quantitatively to track HNO$_3$ chemistry in both laboratory and ambient environments.

*Code and Data availability.* The box model simulations, including model mechanisms, input files, and output files have been made publicly available at: https://doi.org/10.5281/zenodo.15615851. The experimental data and figure scripts have been made

publicly available at: https://doi.org/10.5281/zenodo.15616525. The chemical mechanism and isotope data are provided in the Supplementary Material.

*Author contributions.* WWW, MT, NLN, MGH designed the conducted experiments. WWW, MT, DEB, GE, PT, WX, JR, FL, GH, JBM conducted the experiments. WWW and DEB conducted the offline data analysis. WWW conducted the chamber

simulations with input from MGH, MT and NLN. WWW wrote the manuscript with input from all authors. WWW and MGH secured funding.

*Competing interests.* The contact author has declared that none of the authors has any competing interests.

*Acknowledgements.* We thank Ruby Ho for laboratory assistance.

*Financial Support.* This research has been supported by NOAA's Climate Program Office's Atmospheric Chemistry, Carbon Cycle, and Climate Program (NOAA AC4 NA18OAR4310118).

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
