# Peer review of "Evaluating $NO_x$ Fate and Organic Nitrate Chemistry from $\alpha$ -Pinene Oxidation Using Stable Oxygen and Nitrogen Isotopes"

_EGUsphere, 2024_

## Author Response (AR1)

**Response to Reviewers**

We thank the reviewers for their thoughtful and thorough feedback on our manuscript. We have revised the manuscript accordingly to address all comments and improve clarity, organization, and scientific rigor. Major updates include: (1) Substantial text clarifications and improved flow throughout the manuscript; (2) Restructuring of the abstract and introduction to better highlight the motivation, novelty, and significance of the work; (3) Expanded $\Delta^{17}O$ model sensitivity tests, including additional simulations to assess the impact of chamber background $HNO_3$, wall loss, and revised $\Delta^{17}O(HNO_3)$ mass balance assumptions; (4) Refinement of the nitrogen isotope discussion: Since $\delta^{15}N$ values were not available for all initial $NO_y$ species in our experiments, we focused the interpretation on $\delta^{15}N$ enrichment patterns between $NO_x$ products and $NO_2$. We also removed the modeled $\delta^{15}N$ results and associated discussion, given the high degree of uncertainty and the need for a more comprehensive investigation beyond the current scope; (5) Significant strengthening of the $\Delta^{17}O$ model framework, enabling a clearer understanding of the key controls necessary to reconcile modeled and observed values. We have also modified our title to, "Evaluating $NO_x$ Fate and Organic Nitrate Chemistry from $\alpha$-Pinene Oxidation Using Stable Oxygen and Nitrogen Isotopes" to more accurately reflect the goals and scope of our work. Overall, these changes, together with point-by-point responses provided below, aim to address reviewer concerns and enhance the clarity and impact of the manuscript.

**Reviewer #1-Matthew Johnson**

**General Comments:** The goal of this paper is to use analysis of the relative abundance of stable isotopes of N and O to plumb the interactions coupled NOx and alpha pinene oxidation. It is an ambitious goal and a powerful tool, and the authors are able to derive some important and unique results. The authors use a sophisticated atmospheric chamber and isotopic measurement methods and protocols that have been developed, painstakingly, over many years. There can be a gap between the isotope and aerosol communities and one concern is that the results be presented in a way that allows both groups to understand the work and its implications. Here I am mainly thinking of researchers outside the isotope community. In addition, the style should be adjusted to present quantitative results in place of broad-brush qualitative descriptions. A final concern is that modeling is used to derive the results and additional work should be done to describe the impacts of the many necessary assumptions. This can be done by paying attention to error budgets and sensitivity analysis, and by validating the model through comparison to experimental and field results. Overall I like the paper very much and recommend publication after revision to address these points.

**Response:** We thank the reviewer for their thoughtful comments and for recognizing the painstaking development of methodologies and analysis that culminated in this manuscript. As noted, this work involved the development of new techniques for collecting $NO_y$ compounds, advancements in modeling $\Delta^{17}O$ chemistry, and careful interpretation of both stable isotope and aerosol mass spectrometry data. We appreciate the reviewer's recognition of the ambition and potential impact of this study, as well as their helpful suggestions to improve clarity and accessibility for a broader audience. In response, we have significantly revised the manuscript to address these concerns.

Specifically, we have rewritten the abstract and introduction to more clearly articulate the motivation for the study and to better bridge the gap between the isotope and aerosol chemistry communities. We have also expanded our discussion of model assumptions and uncertainties. This includes a suite of new sensitivity tests for our $\Delta^{17}O$ simulations, which examine the effects of dilution rate, wall loss, reaction rate constants, organic nitrate hydrolysis, $N_2O_5$ heterogeneous chemistry, chamber background levels, and assumptions regarding oxygen isotope mass balance. These additions have allowed us to better evaluate the robustness of our $\Delta^{17}O(HNO_3)$ results and to clarify the implications of our findings. In response to the reviewer's concern (as well as similar comments from other reviewers) regarding the $\delta^{15}N$ modeling, we have decided to remove this portion from the revised manuscript. Given the number of uncertainties associated with nitrogen isotope modeling, we believe that a more rigorous and comprehensive treatment is warranted. We plan to address $\delta^{15}N$ modeling in a future manuscript that will incorporate additional chamber experiments, including those involving inorganic chemistry, to ensure that the analysis is well-supported and focused on understanding nitrogen isotope fractionation effects. Below, we provide a detailed point-by-point response to each of the reviewer's comments.

**Specific Comments**

**Comment:** The prose flows, there are many details, and it is entirely possible for the reader to get lost in the trees and fail to appreciate the landscape. I suggest being sure to add text to frame or give context, for example by adding transitions to introduce sections and show how they fit into the bigger picture. One example is that Section 3 dives straight into Section 3.1 and then into subsection 3.1.1 without a word of text, just section headings.

**Response:** Thank you for pointing this out. We agree that the manuscript would benefit from improved transitions and contextual framing to help guide the reader through the results. In response, we have added introductory text at the beginning of each major section in the *Results and Discussion* to clearly state the purpose and relevance of the upcoming content.

Specifically, we have:
- Added an introductory paragraph to Section 3.1 ("Isotope Observations") to contextualize the measured nitrogen and oxygen isotope data on Lines 359-366
- Included a transition at the end of Section 3.1.2 to highlight the importance of distinguishing between organic and inorganic nitrate on Lines 450-452, providing motivation for the following Section 3.2.
- Introduced Section 3.3 ("Model Simulations") with an overview of the modeling framework and goals on Lines 562-569.

**Comment:** There are many passages that make strong qualitative claims and I strongly advise to instead show the evidence, 'write it in numbers', and let the data speak for itself. Examples below. The Abstract says that there are uncertainties regarding coupling between a-pinene and NOx in NOx loss, renoxification and oxidation chemistry, and says that the study 'provides insights into D17O transfer dynamics, nitrogen isotope fractionation, and the role of NOx-BVOC chemistry in air quality, highlighting the potential of D17O and d15N as tools for evaluating complex atmospheric processes.' Is it possible to convert some or all of these claims into

quantitative statements? What specifically is the new insight into D17O transfer dynamics? Is the promise of D17O and d15N analysis realized? Has the new mechanism been able to explain something that could not otherwise be explained, or make predictions?

**Response:** Thank you for this thoughtful comment. We agree that quantitative evidence is essential to support the qualitative claims made in the manuscript, and we have revised the text accordingly. We have updated the *Abstract* to present more specific quantitative findings. The revised version now includes the following: "Nitrogen isotope fractionation, quantified as the difference in $\delta^{15}N$ values ($\Delta\delta^{15}N$), revealed consistently positive $\Delta\delta^{15}N(HNO_3 - NO_2)$ values ($+28.9 \pm 13.4$ ‰ in daytime experiments; $+22.2 \pm 1.4$ ‰ at night) and negative $\Delta\delta^{15}N(pNO_3-NO_2)$ values ($-13.6 \pm 5.8$ ‰ in daytime experiments). This reflected distinct formation pathways and isotope effects including $NO_x$ photochemical cycling, thermal dinitrogen pentoxide ($N_2O_5$)–nitrate radical ($NO_3$)–$NO_2$ equilibrium, and $HNO_3$ production mechanisms. Box model simulations based on $\Delta^{17}O$ values as a constraint were conducted using a newly developed gas-phase mechanism, which reproduced $\Delta^{17}O(NO_2)$ and $\Delta^{17}O(pNO_3)$ (compared to simulated $\Delta^{17}O(RONO_2)$) accurately, with an average model bias of $0.9 \pm 2.4$ ‰ ($R^2 = 0.98$) and $-1.4 \pm 2.4$ ‰ ($R^2 = 0.55$ and $R^2 = 0.97$ when excluding one outlier), respectively. We further empirically derived important isotopic parameters such as the $\Delta^{17}O$ value transferred from $O_3$ through comparison of model-simulated oxygen atom source contributions with observed $\Delta^{17}O$ values for $NO_2$ and $pNO_3$ across experiments. This yielded best-fit slopes of $39.4 \pm 0.6$ ‰ for $NO_x$ photochemical cycling and $41.7 \pm 1.2$ ‰ for $RONO_2$ formation, consistent with near-surface observations of $\Delta^{17}O$ in the terminal oxygen atom of $O_3$." This update was made on Lines 35-46.

Additionally, to address the reviewer's important question about the predictive value and utility of our modeling approach, we now include a more detailed quantitative comparison of model performance for NO and $\alpha$-pinene decay across the different mechanisms evaluated (MCM, RACM2, and $NO_x$-API). This is presented via one-to-one plots with $R^2$ values and bias calculations (Figures S4 and S5) and summarized in a new Table 4.

**Comment:** In the Introduction line 41, reading that the 'interplay bears significant consequences' is saying, 'take my word for it'. Please rewrite to give evidence. The reader is provided with eight references that cover 'air quality, climate, global reactive nitrogen budget and secondary organic aerosols'. The authors seem to be asking us to go figure it out. Instead I would suggest taking the time to identify a significant consequence of BVOC oxidation in the presence of NOx on a few of these areas and give specific examples. This paragraph has 27 references but the reader is left uncertain regarding why exactly 'Understanding the fates of organic nitrogen and the feedback in oxidation chemistry arising from BVOC/NOx interactions is critical for accurately assessing their roles in NOx loss and recycling, O3 formation, and SOA generation.'

**Response:** We appreciate the reviewer's thoughtful suggestion. In response, we have revised the *Introduction* to replace general statements with specific examples that illustrate the atmospheric relevance of BVOC–$NO_x$ chemistry. We now explain that organic nitrates ($RONO_2$), formed through oxidation of monoterpenes, can serve either as $NO_x$ reservoirs or permanent sinks, depending on their subsequent fate. For example, $RONO_2$ can undergo hydrolysis to form $HNO_3$ with lifetimes ranging from minutes to hours and hydrolyzable yields varying from 9 % to 34 % depending on structure and aerosol conditions (Takeuchi and Ng, 2019; Rindelaub et al., 2015). Additionally, we have clarified the core motivation of our study: that the contribution of $RONO_2$

to $HNO_3$ and $pNO_3$ budgets remains poorly constrained in current atmospheric models and measurements. To address this, we aim to develop new diagnostic tools based on stable isotope measurements to track the fate of $RONO_2$ in controlled chamber experiments. This framing now appears clearly in the revised Introduction (Lines 72–77).

**Comment:** Line 59, 'The natural variations..may be a promising tool to enhance our insight into the intricate connections..and their implications for atmospheric composition.' A promise is made to the reader here - be sure to revisit this point in the discussion at the end. Does the present work provide the evidence?

**Response:** Thank you for this insightful comment. We agree that the manuscript should clearly revisit this point. In the revised *Conclusion*, we have addressed this directly by highlighting how our multi-isotope approach successfully revealed distinct oxidation pathways. Specifically:

- We demonstrated that $\Delta^{17}O$ and $\delta^{15}N$ values systematically differ between inorganic nitrate ($HNO_3$) and organic nitrate ($RONO_2$), offering a powerful means to disentangle their contributions to $NO_y$ and $pNO_3$ budgets.
- We quantified the oxygen atom transfer from $O_3$ and other oxidants using dual-isotope ($\Delta^{17}O$, $\delta^{18}O$) slopes, establishing a new constraint on oxidant identity and relative importance in nitrate formations.
- Model simulations accurately reproduced $\Delta^{17}O(NO_2)$ and $\Delta^{17}O(pNO_3)$, validating key aspects of photochemical cycling and organic nitrate formation mechanisms, and confirming that $\Delta^{17}O$ serves as a robust tracer of oxidation chemistry under a range of conditions.

While our model had greater difficulty reproducing $\Delta^{17}O(HNO_3)$ across all conditions, we used this mismatch as a diagnostic tool to identify and prioritize knowledge gaps, such as the role of chamber blanks, heterogeneous $N_2O_5$ chemistry, and mass-balance assumptions in the $NO_2 + OH$ reaction. Thus, we believe this work delivers on the initial promise and provides clear, quantitative evidence that stable isotopes are powerful tools for diagnosing atmospheric nitrate formation pathways.

**Comment:** Line 61, 'Stable isotope approaches offer novel avenues to probe and refine our understanding..unravel the dynamics..of interactions and ultimately contribute to formulating informed air quality management strategies.' Also here - this sounds nice, but would be more valuable if it is connected with specific findings. What specific, informed air quality management strategies can be made based on the work described in this paper?

**Response:** Thank you for this thoughtful comment. We agree that the original sentence was too broad and aspirational without being directly tied to the specific findings of this study. While our results suggest that stable isotope analyses have potential to inform source attribution of $pNO_3$ and distinguish between inorganic and organic nitrate pathways, their direct application to regulatory air quality management is still premature. To avoid overstating the implications, we have removed this sentence from the revised manuscript and now focus our discussion on the diagnostic potential of isotopes to advance process-level understanding. We believe that future studies, especially those linking isotope-based source attribution to ambient observations and control strategies, will be needed before drawing direct connections to air quality management applications.

**Comment:** How common are the conditions under which BVOC-NOx interactions will be important? Given the relatively short lifetimes of each, the overlap might be restricted to specific zones. However, the authors are claiming global impacts in addition to regional. Please walk the reader through how these larger phenomenon can arise.

**Response:** Thank you for pointing this out. In response, we have removed the word "global" from the original statement in Line 41 and clarified that the impacts of BVOC-$NO_x$ interactions are more regionally specific. In the revised *Introduction*, we now emphasize that these processes are especially relevant in regions with significant overlap of anthropogenic $NO_x$ emissions and biogenic volatile organic compound (BVOC) emissions, such as the southeastern United States, parts of Europe, and forested urban interface zones. We added the following to Lines 73-75, "These interactions are particularly important in regions where high biogenic monoterpene emissions coincide with elevated anthropogenic $NO_x$ levels, such as forested areas near urban regions."

**Comment:** There are many approximations in the model and even for experts it is difficult to disentangle the implications of the assumptions and uncertainties. Some examples:

**Response**: Thank you for pointing this out. We have addressed each of the cited examples of the model approximations (see below).

**Comment:** Line 33, 15alpha for NO2+OH is taken as 0.997. How well known is this value? What happens if the value is taken as 0.998, or 1.1?

**Response:** Thank you for raising this point. The $^{15}\varepsilon$ (and $^{15}\alpha$) value for the $NO_2 + OH \rightarrow HNO_3$ reaction is currently not well constrained and remains an area of active uncertainty. In the revised manuscript, we no longer perform $\delta^{15}N$ modeling and thus do not apply or rely on a specific value for this isotope effect in our simulations. However, we do discuss this reaction in the context of observed nitrogen isotope fractionation between $NO_2$ and $HNO_3$, expressed as $\Delta\delta^{15}N(HNO_3-NO_2)$, which we calculate from our measurements.

These observed values range from ~0.4 ‰ to ~45 ‰, highlighting substantial variability that cannot currently be reconciled with a single assumed $^{15}\varepsilon$ value. In the literature, estimates for the $^{15}\varepsilon$ of this reaction span a wide range from approximately –3 ‰ (based on reduced mass arguments; Freyer et al., 1993) to +40 ‰ (as assumed in the iNRACM mechanism based on equilibrium assumptions; Fang et al., 2021). Interestingly, our data offer some experimental support for both interpretations: the low $\Delta\delta^{15}N(HNO_3-NO_2)$ observed in the HONO experiment aligns with the smaller $^{15}\varepsilon$ value, while higher values in the $NO/H_2O_2$ experiments are consistent with a larger fractionation. We discuss these findings in the revised manuscript (Lines 431–443) and emphasize the need for dedicated experimental studies to better constrain the $^{15}\varepsilon$ value for this key reaction under relevant atmospheric conditions. Given the wide range of plausible values, we have avoided speculative interpretation and instead highlight this as a critical gap in our understanding.

**Comment:** Line 204 mass-dependent coefficient taken to be 0.52. What is the error, what are the implications of this choice?

**Response:** Thank you for raising this point. The use of a mass-dependent coefficient of 0.52 is standard in the atmospheric chemistry community, particularly in studies focusing on large mass-independent oxygen isotope effects such as ozone-derived $\Delta^{17}O$ transfer. The uncertainty associated with this coefficient is typically less than ±0.1 ‰, which is negligible relative to the large $\Delta^{17}O(O_3^{term})$ values (~39 ‰) considered in this study. As this convention is already described

and referenced in the manuscript (Lines 257–260), and widely accepted within this specific research community, we have not made changes to the text.

**Comment:** Line 222, uncertainties were calculated to be less than 4.1, 1.4 and 0.9. Thank you, it is good, and please add just a bit to put the information in context: are these values significant?
**Response:** Thank you for the suggestion. We have revised the sentence to provide additional context for the reported uncertainties by comparing them to the observed range of isotopic values in our dataset. Specifically, these uncertainties are small relative to the observed variability and do not meaningfully impact the interpretation of our results. The sentence was revised on Lines 274-276, as follows, "For the quality assurance criterion of $f$(blank) < 30 %, the uncertainties were calculated to be less than 4.1 ‰ for $\delta^{15}N$, 1.4 ‰ for $\delta^{18}O$, and 0.9 ‰ for $\Delta^{17}O$. These values are small compared to the observed ranges for pNO₃, which spanned 67.8 ‰ for $\delta^{15}N$, 29.3 ‰ for $\delta^{18}O$, and 10.4 ‰ for $\Delta^{17}O$, and thus do not significantly affect the interpretation of isotope patterns."

**Comment:** Line 289 'This value was not measured but assumed..'
**Response:** Thank you for pointing out this important assumption regarding the potential chamber $HNO_3$ blank and its assumed $\Delta^{17}O$ value. In the revised manuscript, we have addressed this issue more directly. The original text on Lines 289–291 has been removed. Instead, we now explicitly explore the impact of a potential $HNO_3$ blank on the model–observation mismatch in the low-$NO_x$ experiments. Specifically, we have calculated the $\Delta^{17}O$ value that the chamber blank would need to have to reconcile the observed $\Delta^{17}O(HNO_3)$ values for Exp. 1 and 2. This analysis has been added to Section 3.3.4 ("$\Delta^{17}O(HNO_3)$ Model Sensitivity Tests") in the revised manuscript.

**Comment:** Line 297 15N/14N ratio taken as 0.003677 -- is there an error on this number?
**Response:** Thank you for pointing this out. You are correct that there was a typo in the originally reported $^{15}N/^{14}N$ ratio. The correct value is 0.003676, which is the recommended atmospheric ratio published by the *Commission on Atomic Weights and Isotopic Abundances* (Coplen et al., 1992) and widely used in the isotope geochemistry community. However, in our revised manuscript, this line has been removed as we have also removed the $\delta^{15}N$ modeling component from the study due to uncertainties in the isotopic input parameters. We appreciate your attention to detail and have ensured that all instances referencing this ratio have been updated or removed accordingly.

**Comment:** Lines 295 to 305, a series of values are assumed. How sensitive is the result to these values?
**Response:** Thank you for raising this important point. We agree that the $\delta^{15}N$ model simulations are highly sensitive to the assumed initial $\delta^{15}N$ values of $NO_y$ precursors, which were not fully constrained in our experiments. Because of this sensitivity and the lack of comprehensive isotopic measurements for all initial $NO_y$ species, we chose to remove the $\delta^{15}N$ modeling from the revised manuscript. Instead, we have slightly reframed our $\delta^{15}N$ discussion to focus on the observed differences in $\delta^{15}N$ between $NO_2$ and its products, specifically $\Delta\delta^{15}N(HNO_3 - NO_2)$ and $\Delta\delta^{15}N(pNO_3 - NO_2)$. This approach avoids reliance on uncertain source values and enables us to assess nitrogen isotope fractionation trends during $NO_x$ oxidation. These updates are described in the revised manuscript on Lines 427–452.

**Comment:** Line 309, Wall loss is not considered in the model. How severe is this

approximation, detail chamber volume, leak rate/lifetime of air in chamber, diffusion time to wall, etc to put in perspective.

**Response:** Thank you for this important comment. In the revised manuscript, we now explicitly address the potential influence of chamber wall loss on the simulated $\Delta^{17}O$ values through a new set of sensitivity analyses (see Lines 618–665, Fig. 6B, and Table S7). We tested three wall-loss scenarios: (1) no wall loss, (2) a moderate wall-loss case using literature-based rate constants for $NO_y$ and $O_3$ (e.g., Morales et al., 2021; Wang et al., 2014), and (3) an extreme scenario with wall-loss rates increased by a factor of 10. The moderate wall-loss scenario resulted in minimal changes in $\Delta^{17}O$ values (≤2.3 %) relative to the no-wall loss scenario, suggesting that omitting wall loss in the base model does not substantially affect the interpretation of our results. However, the extreme case produced larger shifts in $\Delta^{17}O(NO_2)$, $HNO_3$, and ONIT, highlighting the sensitivity of the system to strong wall interactions. We also provide estimated uncertainties associated with both dilution and wall loss. Combined, they introduce propagated uncertainties of approximately ±2.5 % for $HNO_3$, ±3.8 % for $NO_2$, and ±1.1 % for ONIT. When accounting for the additional uncertainty in the $\Delta^{17}O(O_3{}^{term})$ reference value (±5 %), we estimate overall model $\Delta^{17}O$ uncertainties of ±5.6 %, ±6.3 %, and ±5.1 % for $HNO_3$, $NO_2$, and ONIT, respectively. Although we did not measure wall loss directly in these experiments, this analysis provides quantitative bounds on its potential impact and confirms that uncertainties in the $\Delta^{17}O(O_3{}^{term})$ value are a larger source of model variability. These uncertainty estimates are now consistently applied in the interpretation of our modeling results.

**Comment:** Line 326, 'The δ18O of RO2/HO2 radicals has previously been suggested to be near δ18O(O2)' How near? Suggested? Are we on solid ground?

**Response:** Thank you for pointing out that this statement was vague. We have updated the language in this section and have removed this particular line, but we have expanded on our discussion of the low-end member $\Delta^{17}O/\delta^{18}O$ oxidant on Lines 377-385: "The derived low-end $\delta^{18}O$ oxidant value is consistent with a scenario in which these radicals are sourced from atmospheric $O_2$ ($\delta^{18}O \approx$ 23 ‰; Craig, 1957), but undergo isotopic fractionation during their formation and subsequent oxygen atom transfer to $NO_y$. Although the exact $\delta^{18}O$ enrichment factors for $RO_2/HO_2$ or OH formation and reaction are not well-constrained, a net isotope enrichment factor ~-12 ‰ is plausible, particularly for unidirectional reactions involving $^{18}O$ fractionation (Walters and Michalski, 2016). Additionally, contributions from OH could further influence the low $\delta^{18}O$ endmember, especially if oxygen atom exchange with ambient water vapor occurs (Dubey et al., 1997). Altogether, the inferred $\delta^{18}O$ of 11.1 ±1.0 ‰ for the low-$\Delta^{17}O$ oxidant endmember likely represents a composite signal from multiple oxidants ($RO_2$, $HO_2$, OH) originating from $O_2$ and/or $H_2O$, modified by kinetic and equilibrium isotope effects."

**Comment:** Line 327 'the derived d18O of 11.8 +/- 1.0 ‰ is near the atmospheric d18O(O2) value of 23.2 ‰' (Thank you that here, there is a space between number and unit). These two values are not within their mutual error ranges and one is twice that of the other, please help the reader understand how this is 'near'?

**Response:** Thank you for noting that the difference between the $\delta^{18}O$ values is larger than originally conveyed. We agree that the use of the word "near" may have been misleading and have revised the sentence for clarity. While the derived $\delta^{18}O$ value of 11.1 ‰ is ~12 ‰ lower than that of atmospheric $O_2$ (~23 ‰), this difference is still relatively modest in the context of the broad

$\delta^{18}O$ range predicted in atmospheric oxidants to range from ~-50 to 120 ‰. We have updated the text on Lines 377-385, as noted in the previous response.

**Comment:** Please walk the reader through the inputs and outputs of the model, the assumptions and adjustable parameters. Conduct sensitivity analysis and validation. It would be helpful to provide an analysis of which parts of the model require additional research, to help guide future research/as a service to the community.
**Response:** Thank you for this valuable suggestion. In response, we have expanded the manuscript to include a more thorough examination of the model assumptions, inputs, and outputs, as well as a series of targeted sensitivity analyses aimed at identifying which components most influence the modeled $\Delta^{17}O$ values. First, we added a new section (Section 3.2: $\Delta^{17}O$ Model Sensitivity to Dilution and Wall Loss) that explores the influence of two key physical processes that included dilution and wall loss on $\Delta^{17}O$ model outputs. These simulations help clarify the extent to which chamber-specific conditions may bias our results. Second, we conducted an expanded set of chemical sensitivity tests, now described in Section 3.3.4: $\Delta^{17}O(HNO_3)$ Model Sensitivity Tests). These include:

- Varying the rate constant of the $NO + HO_2$ reaction,
- Evaluating the potential contribution of ONIT hydrolysis to $HNO_3$ formation,
- Including $N_2O_5$ heterogeneous uptake reactions,
- Accounting for a potential chamber $HNO_3$ blank, and
- Revising the oxygen atom mass balance assumptions for the $NO_2 + OH$ reaction.

Together, these sensitivity tests allow us to better identify which processes drive mismatches between observed and modeled $\Delta^{17}O(HNO_3)$, and which areas of the mechanism require further experimental or theoretical constraints. Finally, we note that due to the numerous uncertainties associated with nitrogen isotope effects ($\delta^{15}N$), we have chosen to remove the model-based $\delta^{15}N$ discussion from this manuscript. We believe a more focused, future study is warranted to fully explore the relevant parameters and fractionation factors, which would exceed the scope of the current work.

**Comment:** Line 310 'low relative humidity conditions', but how low, there must be a measurement?
**Response:** Thank you for pointing that out. The relative humidity was fixed at 30 % for all experiments. We added that detail to this line in the revised text on Line 304.

**Comment:** The Conclusion section is powerful, thank you. It ties the paper together. It could be revisited during the rewrite - to bring focus, brevity, specific quantified results. These could include results from the sensitivity analysis and specific research needs.
**Response:** Thank you for the positive feedback on the Conclusion section. In the revised manuscript, we have refined the *Conclusion*. We now emphasize key quantified results from the sensitivity analyses, particularly those related to $\Delta^{17}O(HNO_3)$ model performance, and summarize the main factors influencing model–observation agreement. In addition, we highlight specific research needs, including uncertainties in nitrogen isotope fractionation and areas where further mechanistic or isotopic constraints are critical.

**Comment:** Setting 'the weighted branching ratio of α-pinene+OH+NO leading to organic nitrate versus NO2 production' to 0.222 is a very specific number and deserves further comment. Do you have any thoughts how this value would change with atmospheric conditions like humidity, temperature, pressure, NO concentration? Is it as accurate as three digits or is it merely to within e.g. 10%?

**Response:** Thank you for raising this important point. The value of 0.222 represents the branching ratio used in the MCM and adopted in the $NO_x$-API mechanism. This value has been shown to simulate $\Delta^{17}O$ of $NO_2$ and $RONO_2$ reasonably in this study, suggesting it is a suitable approximation under typical chamber conditions. However, as we did not explicitly evaluate the sensitivity of our $\Delta^{17}O$ model to variations in this branching ratio, nor explore how it may shift under different environmental conditions (e.g., humidity, temperature, NO levels), we agree that its inclusion in the conclusion section without additional context may be misleading. To avoid over-interpreting the precision or broader applicability of this value, we have removed this sentence from the conclusion in the revised manuscript.

**Comment:** Line 699, 'Our findings strongly suggest that pNO3 in these experiments originated exclusively from organic nitrate, a conclusion supported by online AMS data. Furthermore, the Δ17O and δ15N evidence demonstrated that organic nitrate hydrolysis was not a major source of HNO3 under the studied conditions, which predominantly involved low relative humidity.' I just want to say that these are excellent results, very interesting! Thank you.

**Response:** Thank you for your positive feedback on our findings and the demonstration of the utility of isotope constraints in our experimental investigation.

**Technical Corrections**

**Comment:** Recommended practice by professional societies such as IUPAC and SI is that there should always be a space between a number and a unit. So for example 1.7 ‰ not 1.7‰. Check throughout, current usage is inconsistent.

**Response:** Thank you for raising this point. Throughout the revised manuscript, we have ensured that there is a space between number and units.

**Comment:** Line 217 'corresponds to the fraction of NO3- that corresponds to the blank.' Please rewrite.

**Response:** Thank you for the comment. We have revised the language of this sentence to the following, "…and $f$(blank) corresponds to the $NO_3^-$ method blank fraction." This correction was made on Lines 269-270.

**Comment:** Line 260 change experiment to experimental

**Response:** Thank you for catching this error, which was corrected in the revised manuscript on Line 321.

**Comment:** Figure 1 inset, y, p, r, x should be italicized. Also caption of Table 2, 'Robs' should be italicized e.g. '$R_{obs}$'. All variables should be italicized.

**Response:** Thank you for pointing this out. The referenced variables were italicized in the revised Figure 1. Further, all delta values in the figures were italicized in the revised manuscript. In

addition, *"$R_{obs}$"* in the Table 3 (formerly Table 2) caption was also italicized in the revised manuscript.

**Comment:** Table 2, what are the uncertainties on the values in the final three columns? Perhaps a blanket uncertainty could be used e.g. '+/- 5 %' or '10 % of the given value'.
**Response:** Thank you for pointing this out. In the revised manuscript, we have added typical uncertainties associated with $pNO_3$ quantification for each measurement method: ±14 % for AMS, ±20 % for PILS, and ±20 % for filter-based collection. These uncertainty estimates are now included in the text on Lines 491–511. Additionally, we have propagated these uncertainties through the various ratio calculations and now report them in the updated version of Table 3 (formerly Table 2) to provide a more complete and transparent assessment of measurement precision.

**Comment:** Line 437 suggest rewriting this sentence for clarity, tone, grammar, break into two etc. 'The offline filter collection and extraction technique matches the trend in which more pNO3 hydrolyzed for the photochemical experiments compared to the nighttime; however, the filter technique would indicate a higher proportion of potential hydrolysable pNO3 from photochemical experiment than these previous estimates, though with a different timescale.'
**Response:** Thank you for the suggestion. We have revised the sentence as follows, "The offline filter collection and extraction technique follows the observed trend of a greater proportion of $pNO_3$ hydrolyzed to $NO_3^-$ (aq) in photochemical experiments ($82.6 \pm 14$ %; $n = 7$) compared to nighttime conditions (7.6 %; $n = 1$). This is inferred based on the relative amount of quantified $NO_3^-$ (aq) from the filter extraction to the total $pNO_3$ measured by HR-ToF-AMS. However, the filter-based method suggests a higher proportion of potentially hydrolyzable $pNO_3$ in the photochemical experiments than previously reported estimates." This revision was made on Lines 529-533.

**Comment:** Line 370 spelling 'masse'
**Response:** Thank you for catching this error, which has been corrected on Line 433 in the revised manuscript.

**Comment:** Line 445, check word choice, consider replacing 'speciation' with 'interpretation' or 'assignment'?
**Response:** Thank you for the suggestion. We have replaced 'speciation' with 'assignment' in the revised manuscript on Line 538.

**Comment:** Figure 3 caption, 'The observed pNO3 concentrations are faceted by the various experiments conducted'. Do I understand this correctly to mean, 'This figure shows how the variety of experiments performed impact the observed pNO3 concentrations'? I found it hard to interpret and suggest rewriting for simplicity, clarity.
**Response:** Thank you for pointing out that this figure caption was confusing. This caption for Fig. 3 has been revised to, "The observed $pNO_3$ concentration data are shown for each of the conducted experiments."

**Comment:** Line 459, 'This comparison indicates that the developed mechanism well represents the oxidation of α-pinene and formation of oxidants under a wide range of experimental

conditions. The simulations using the USC-API mechanism was a vast improvement' Please quantify; 'well represents', 'wide range' and 'vast improvement' are imprecise and will mean different things to different readers. Similar line 489, 'well-suited', line 497 'well-reproduced' and line 501, 'well-calibrated'.

**Response:** Thank you for pointing out that our comparison was not quantitative. In the revised manuscript, we have quantified the performance of the model simulations using the absolute mean bias of the model output compared to the observations for $\alpha$-pinene decay as well as NO decay. These quantitative metrics of model performance were added to the revised manuscript on Lines 574-578, "Model performance was evaluated by comparing measured and modeled concentrations of NO and $\alpha$-pinene using one-to-one plots, with corresponding $R^2$ values and quantification of model biases (Figs. S4–S5) and Table 4 summarizes these results. Overall, the $NO_x$-API mechanism provided improved model performance, evidenced consistently higher $R^2$ values (averaging 0.97 ±0.03) and lower absolute residuals for both $\alpha$-pinene and NO decay compared to the other mechanisms."  Further, we have provided a direct comparison of model simulated $\alpha$-pinene and NO concentrations to the observations (Fig. S4-S5). Note: we changed the name of the mechanism to $NO_x$-API to be more representative of the chemistry included in the mechanism).

Furthermore, all words using, "well-…" were removed from the revised manuscript.

**Comment:** Line 493 'leaded' to 'leading' (I think?) and this sentence should be rewritten to simplify, clarify: 'Oxygen isotope mass-balance indicates that the $\alpha$-pinene-derived peroxy radicals + NO pathway would be the expected pathway leaded to a low $\Delta17 O(pNO3)$ value as only one oxygen atom in the nitro group of the generated RONO 2could derive from O3.'

**Response:**  Thank you for pointing out the typo and for suggesting a clearer formulation of the sentence. We have corrected the grammatical error and revised the sentence for clarity as suggested. However, we also significantly updated the surrounding text to reflect a more nuanced discussion of the formation pathways of organic nitrates (ONIT) and their associated $\Delta^{17}O$ values based on model outputs and included reference to our oxygen isotope mass balance equations (summarized in Table 1). The revised text now reads: "The temporal evolution of the simulated $\Delta^{17}O(ONIT)$ closely followed that of $\Delta^{17}O(NO_2)$ but remained lower due to dilution effects associated with the dominant formation pathway of organic nitrates via $NO + RO_2$ reactions during the photochemical experiments (Table 1), resulting in ONIT dominated by ONITa, ONITb, and ONITc compounds (Fig. S3). During nighttime oxidation experiments, ONIT formation primarily proceeded via $\alpha$-pinene + $NO_3$ reactions, leading to ONIT with higher contribution of DIMER and PDN compounds. Due to $NO_2/NO_3/N_2O_5$ thermal equilibrium that resulted in $\Delta^{17}O(NO_2) \approx \Delta^{17}O(NO_3) \approx \Delta^{17}O(N_2O_5)$, the simulated nighttime $\Delta^{17}O(ONIT)$ values were approximately equal to the simulated $\Delta^{17}O(NO_2)$."  This update appears on Lines 701–707 of the revised manuscript and provides improved clarity and mechanistic context in response to your helpful suggestion.

**Comment:** 526 suggest change 'enable' to 'allow'
**Response:** Thank you for the suggestion. The original line was removed from the manuscript, in our revised treatment of $HNO_3$ sensitivity tests.

**Comment:** 552 'partially improve', how is this different from 'improve'? Suggest edit.

**Response:** Thanks for your suggestion. In our revised sensitivity analysis, we have quantified the model simulations performance by providing means biases and $R^2$ values. These updates were made on Lines 775-884.

**Comment:** Figures 6, 7, 8, 9, 10 and 11 have largely identical captions, it doesn't seem necessary that they all repeat the same details. They also all have the gramamtical error, 'with the black line span the collection time range'.
**Response:** Thank you for pointing this out. In the revised manuscript, we combined Figures 6, 7, and 8 (now Fig. 7) and removed 9, 10, and 11 since we no longer included the $\delta^{15}N$ model results in the revised manuscript. This helps cut down on the redundance in the caption. Also, we have also corrected the grammatical errors.

**Comment:** Table S1 and S2, suggest taking a moment to edit the superscripts and subscripts in chemical formulas (subscript numbers), term symbols (superscript numbers), etc.
**Response:** Thank you for the suggestion. We have formatted the formulas and term symbols in the revised Supplement for Table S1 and Table S2.

**Comment:** Table S2, do you have the references for these rate coefficient values? Note that it is not a reaction rate but a rate coefficient. What is the difference? Consider A + B --> C with rate coefficient $k$. The rate of the process is $r = k[A][B] = d[C]/dt$. The value '$k$' is sometimes called the rate constant, but it is not constant, it changes with temperature, etc.; it is the rate coefficient. It would be useful to indicate the units for these rate coefficients, 1/s, cm^3/s etc. I might guess the activation energies are given in units of Kelvin, is that correct? Add a footnote. R022 says to multiply by 'H2O', should this be '[H2O]'? R039 and R040 are not reaction rates $r$ or rate coefficients $k$, they are equilibrium constants $K$.
**Response:** Thank you for pointing out the flaws in the presentation of the original Table S2. In the revised manuscript, we have more thoroughly presented the reactions and their rate coefficients. The original Table S2 has been split into 4 tables that includes: photolysis reactions (Table S2), summary of the thermal reactions (Table S3), Troe Reaction Parameters (Table S4), Troe Equilibrium reactions (Table S5), and reactions with complex rate expressions (Table S6). We have provided the pre-exponential factor and activation energies, units, as well as references.

**Reviewer #2: Mei-Yi Fan & Yanlin Zhang**

**Summary:** This manuscript aims to modify the NOx-BVOC chemical mechanism in the INRACM model by analyzing stable nitrogen and oxygen isotopes ($\Delta^{17}O$ and $\delta^{15}N$) of chamber experimental samples, particularly focusing on the interaction mechanism between nitrogen oxides and the oxidation of $\alpha$-pinene. However, the description of the experimental sample collection in the chamber and the testing methods for stable nitrogen and oxygen isotopes using IRMS in the manuscript is somewhat thin and requires further elaboration of experimental details. Additionally, regarding the bacterial denitrification method for testing $\delta^{18}O$, $\Delta^{17}O$, and $\delta^{15}N$, the authors report high blanks (or low sample amounts) in the tests, which may not be convincing to the community developing low-sample-amount testing methods. The last concern is in the model section, where the authors use multiple assumptions in the parameter setting part and do not provide sufficient evidence to convince the community. Therefore, a comparison

study with actual observations can make the results more solid. Overall, this is a work on modeling the mechanism of organic nitrate formation, and the mechanism proposed in this paper is expected to improve the simulation of organic nitrate concentration in box models or transport models. In all, I l recommend this manuscript to be published after revision of the following comments.

**Response:** We thank the reviewer for their thoughtful and constructive feedback. We have revised the manuscript to include additional details regarding our experimental design, particularly the IRMS methodology and sample collection procedures within the chamber experiments. Our laboratory takes great care in isotope analysis, and we have added a more thorough description of our nitrate sample preparation and measurement protocols. In response to the concern about high blanks or low sample amounts, we would like to clarify that the reported uncertainties account for *both analytical variance* and *sample collection variability*, including the influence of small but non-negligible $NO_3^-$ blanks on the filter used for aerosol collection.

We have implemented a Monte Carlo-based uncertainty analysis that propagates the effects of *blank correction* and *measurement error* (from IRMS), ensuring that the final reported $\delta^{15}N$, $\delta^{18}O$, and $\Delta^{17}O$ values reflect the full range of potential uncertainty of the collected sample, rather than only reporting the instrumental uncertainty which will not reflect the actual uncertainty of the collected sample. In each case, the Monte Carlo-based uncertainty estimates are larger than the raw IRMS analytical uncertainty. For the samples in question, the blank contribution was less than 30 % of the total collected $NO_3^-$, and while this impacts the overall uncertainty, it does not bias the raw isotope measurements.

Regarding the modeling concerns, we have now conducted additional sensitivity tests to evaluate the robustness of our assumptions, including those related to dilution rates, wall loss, background chamber values, ONIT hydrolysis rates, and oxygen isotope mass-balance assumptions. These results and discussions have been added to the modeling section in response to this and other reviewer comments. Overall, we believe these additions significantly strengthen the manuscript. We thank the reviewer again for their suggestions, which have improved the clarity and rigor of the work.

**Major comments:**

**Comment:** 1 . When gas-source IRMS is used for isotope testing, the isotope values of the corresponding gases will gradually increase and stabilize as the amount of prepared gas samples increases. Previous results on gas-source IRMS articles have shown that a sample amount of 10 nmol (0.2 nmol blank) is usually required to achieve the precision of effective testing within an error range of 0.4 per mil. The authors claim that the Monte Carlo method used in the article can achieve isotope testing of tiny amount of samples within an error of 1.1 per mil, but there is not enough solid evidence to convince the community.

Firstly, the concentration of the blank itself varies, and testing with low sample amounts using gas-source IRMS itself brings great uncertainty. The authors report that the isotope values of the three blanks are no meaning for further analysis. Because the 1 standard deviation of the three blanks is greater than 1 per mil, indicating that the difference between the blanks themselves is at least more than 2 per mil. The author cliamed that this 2 per mil (1SD) error could be combined

with Monte Carlo testing and then used to reduce the overall error to <1 per mil. At least in the current evidence, the effectiveness of this method is in doubt.

**Response:** We appreciate the reviewer's thoughtful comment and the opportunity to clarify our methodology. In our study, we distinguish between two sources of blank: (1) **analytical blanks**, which refer to potential contamination during the wet chemical conversion of $NO_2^-$ and $NO_3^-$ to $N_2O$ or $O_2$ for IRMS analysis, and (2) **method blanks**, which refer to $NO_3^-$ contamination on the sampling media (e.g., quartz filters).

**Analytical blanks** were assessed for each batch of IRMS analysis and were consistently below the detection limit (~0.2 nmol). We routinely run reagent and water blanks for each analysis, following our lab's standard QA/QC procedures. For $\delta^{15}N$ and $\delta^{18}O$ measurements, we used 20 nmol of $NO_2^-$ or $NO_3^-$ derived $N_2O$, and for $\Delta^{17}O$ analysis, we used 50 nmol of nitrate-derived $O_2$. These sample sizes are well above the typical thresholds required for high analytical precision, and our IRMS routinely achieves sub-permil reproducibility under these conditions. We added these sample amount details to the revised manuscript on Lines 233-235. For reference, the standard deviations of our isotope reference $NO_2^-$ and $NO_3^-$ materials are reported on Lines 246–248. Based on this, we are confident that analytical blanks do not affect the reported isotope values or their uncertainties.

In contrast, the **method blank** was non-negligible for samples collected on **quartz filters** (i.e., particulate nitrate samples), which contained a measurable background level of $NO_3^-$ prior to use (as indicated on Line 227-228). To quantify its impact on the isotope values of $NO_3^-$ generated during the chamber experiments, we analyzed multiple blank filters and observed variability in both $NO_3^-$ concentration and isotopic composition (see Line 269-270). To rigorously propagate the uncertainty from both concentration and isotope measurements in the mass-balance correction, we used a Monte Carlo simulation (10,000 iterations) to account for uncertainty in the measured values of both the **sample** and the **method blank**. This simulation yields a statistically robust estimate of the isotope composition attributable to **chamber-derived $NO_3^-$**, along with its associated uncertainty.

We would also like to clarify that the blank-corrected isotope values (i.e., those representing $NO_3^-$ from the experiments) carry larger uncertainties than those of our reference materials or the triplicate measurements of blank filters, due to the magnitude and variability of the method blank. For samples that met our quality control criterion (blank fraction < 30 %), the total propagated uncertainties after blank correction were as follows: $\delta^{15}N$: up to ±4.1 ‰; $\delta^{18}O$: up to ±1.4 ‰; $\Delta^{17}O$: up to ±0.9 ‰ (see Lines 273-274). These propagated uncertainties exceed our analytical IRMS uncertainty of reference materials that were ±0.1 ‰ for $\delta^{15}N$; ±0.6‰ for $\delta^{18}O$ and ±0.6‰ for $\Delta^{17}O$ for $NO_3^-$ (see Lines 246-248).

The increase in uncertainty for the sample $NO_3^-$ isotope values is, as expected, dependent on the fractional contribution of the method blank $NO_3^-$ to the total $NO_3^-$ collected. As this blank fraction increases, the propagated uncertainty also increases. To ensure data quality, we chose to report only those samples for which the blank fraction ($f_{blank}$) was less than 30 %, which resulted in the uncertainty described above. These values represent the **total uncertainty following blank correction and should not be interpreted as raw IRMS analytical precision**. We have clarified

this distinction in the revised manuscript. Importantly, we do not claim that the blank-corrected $\delta^{15}N$ values have a precision better than ±1.1 ‰. Rather, the original uncorrected standard deviation for $\delta^{15}N$ was approximately ±1.1 ‰ for the $NO_3^-$ blank on the quartz filter (method blank), and this increased the $\delta^{15}N$ uncertainty of the chamber produced $NO_3^-$ to as much as ±4.1 ‰ after full uncertainty propagation using the Monte Carlo approach. We added the following to Lines 276-279, to make it more clear that the Monte Carlo simulation is used to account for the method blank, "These reported uncertainties for chamber-derived $pNO_3$ isotope values represent total propagated error after blank correction, not the raw instrumental precision."

**Comment:** Secondly, regarding the precision of the method reported by the authors, are the authors referring to the standard deviation of the mean obtained by multiple Monte Carlo tests on the same standard to characterize the overall standard deviation of the test? If this is the case, the isotope values reported by the authors may not be much convincing.

**Response:** We appreciate the reviewer's question and the opportunity to clarify. The Monte Carlo simulation was not used to simply calculate the standard deviation of repeated measurements of the same standard. Instead, it was designed to propagate uncertainty in the blank correction process for the $pNO_3$ samples, which had a small but non-negligible method blank.

Specifically, for each of the 10,000 iterations, the simulation randomly draws values from normal distributions defined by the measured concentration and isotope values (and their respective uncertainties) of both the sample and the method blank. These values are then used in a mass-balance calculation to estimate the blank-corrected $\delta^{15}N$, $\delta^{18}O$, or $\Delta^{17}O$ of the sample. The resulting standard deviation across all iterations reflects the propagated uncertainty in the blank-corrected isotope value for that sample. The standard deviation across these 10,000 iterations is reported as uncertainty for the $pNO_3$ samples (corrected for method blank). Thus, the reported uncertainties correspond to the variability resulting from input **uncertainties in both concentration and isotope measurements** and not from replicate analysis of the same standard. This approach provides a statistically robust estimate of the true uncertainty introduced by the method blank correction. To make this point clearer, we added the following to the revised manuscript on Lines 276-278, "These uncertainties reflect the standard deviation of the 10,000 Monte Carlo iterations for each $pNO_3$ sample, which account for uncertainty in both the sample and method blank concentrations and isotope values."

**Comment:** In addition, the blank concentration accounts for 30% of the total sample amount. If assuming the entire process blank as 0.2 nmol, some sample amounts are even less than 1 nmol. This extremely low sample amount may not provide precise enough data for the discussion of the α-pinene and NOx mechanisms in this article. It is recommended that the authors use higher sample amounts or lower blanks to retest the generated chamber samples.
The blank of isotope testing is the key factor limiting the sample amount. The authors should follow the practices of other laboratories in dealing with N pollution and strive to reduce the sample blank, instead of using this 30% blank to report the test results of the samples.

**Response:** We thank the reviewer for raising this important point. We would like to clarify the distinction between **analytical blanks** and **method blanks** in our study, as this is central to interpreting our results. For the isotope ratio measurements, we used approximately 20 nmol of $NO_3^-$ and/or $NO_2^-$ derived $N_2O$ for $\delta^{15}N$ and $\delta^{18}O$ analysis (via $m/z$ 44, 45, and 46), and 50 nmol

of $NO_3^-$ and/or $NO_2^-$- derived $O_2$ for $\Delta^{17}O$ analysis (via $m/z$ 32, 33, and 34). With these sample amounts, the blanks associated with the wet chemical conversion of $NO_3^-$ to $N_2O$ or $O_2$, and their subsequent analysis by IRMS, were always below our detection limits (<0.2 nmol). As such, we do not expect these **analytical blanks** to influence the reported isotope values. We have now included these sample amounts and blank considerations in the revised methods section (Lines 233–237). Further, we added the following to Lines 239-241 to highlight that analytical blanks were assessed for each batch analysis and were always below the detection limit, "Analytical blanks associated with the conversion of $NO_3^-$ and/or $NO_2^-$ to $N_2O$ or $O_2$ for subsequent IRMS analysis were assessed for each batch and were always below detection limit (~0.2 nmol). These blanks are not anticipated to affect the analytical precision of the reported isotope values".

The 30 % blank contribution referenced in the comment pertains instead to a **method blank.** Specifically, the presence of background $NO_3^-$ on the aerosol filters used for sample collection. These filters contained a small but non-negligible amount of $NO_3^-$ prior to use ($1.5 \pm 0.2\ \mu mol\ L^{-1}$; $n = 5$) (as discussed on Line 225-229). Therefore, the total $NO_3^-$ collected in each experiment reflects both the nitrate produced in the chamber and the background $NO_3^-$ on the filter. While this affects our ability to quantitatively attribute isotope values purely to chamber-generated $NO_3^-$, it does not impact the **analytical precision** or accuracy of the isotope measurements themselves. To address this, we propagated the **method blank uncertainty** using a Monte Carlo approach as discussed in section 2.3 in the original manuscript. We have further clarified this text in particular, "Due to significant $NO_3^-$ blanks found in the $pNO_3$ filter extracts (i.e., method blank), the measured nitrate isotope values ($\delta^{15}N$, $\delta^{18}O$, and $\Delta^{17}O$) were corrected using a mass-balance approach to isolate the isotopic composition of $NO_3^-$ generated within the chamber experiments (Eq. 3-4)", which is on Lines 263-265 in the revised manuscript. We hope that this makes it clear that the Monte Carlo simulation is used to correct for the **method blank**.

We appreciate the reviewer's suggestion and acknowledge that further reductions in method blank contributions through improved filter cleaning or selection could enhance future measurements. This has proven to be a challenge, and we have tried many efforts to thoroughly clean filters below our $NO_3^-$ detection limits but has proved difficult in practice. Furthermore, we were limited in the production of $NO_3^-$ within the chamber. To ensure sufficient $NO_3^-$ for isotope analysis, we had to conduct our experiments at elevated precursor concentrations and still required approximately 4 hours of collection time. Extending the experiments beyond this duration was not feasible due to increasing wall losses and a progressive drop in bag pressure, which necessitated substantial dilution, which was already a part of our experimental strategy, but will act to lower the $pNO_3$ concentration in the process. Still, the isotope data presented here are robust with respect to analytical quality, and we believe they remain valid for interpreting the $\alpha$-pinene–$NO_x$ chemistry studied in this work.

**Comment:** 2 . Memory effect: The $\Delta17O$ results shown by the authors in Figure 1 are between 0 and 40 per mil. This magnitude of difference has a significant memory effect on the testing of small sample amounts, that is: the $O_2$ split from part of the samples in the previous test will remain in the Pt/Au tube in the conflo-IRMS, contaminating the $O_2$ split from the next sample test, thereby causing deviations in the isotope values. The authors need to carefully discuss the potential effects of these effect, and carefully revised their manuscript.

**Response:** We thank the reviewer for raising this important point regarding potential memory effects in $\Delta^{17}O$ measurements using gas-source IRMS. We have carefully considered this issue and would like to clarify our analytical procedures and observations. In our analyses, we did not observe any noticeable $\Delta^{17}O$ memory effects or carryover between samples in our IRMS setup. Specifically, we took several precautions to minimize this risk. First, samples were grouped and analyzed in separate batches based on expected $\Delta^{17}O$ magnitude: $NO_2$ samples (high $\Delta^{17}O$), $HNO_3$ (moderate $\Delta^{17}O$), and $pNO_3$ (low $\Delta^{17}O$) were each measured in their own batch. This approach minimizes the risk of cross-contamination between samples with widely different $\Delta^{17}O$ values. Finally, we note that $\delta^{18}O$ and $\Delta^{17}O$ were measured from the same $O_2$ split but in independent batch runs, and we observed strong internal correlations between $\delta^{18}O$ and $\Delta^{17}O$ across all sample types. This consistency supports the conclusion that memory effects, if present, were minimal and did not significantly bias our results. We have added a brief discussion of this point in the revised manuscript on Lines 237-239, "To minimize potential memory effects from residual $O_2$ gold tube and headspace trapping system, samples were grouped and analyzed in separate batches based on their expected $\Delta^{17}O$ values. Specifically, $NO_2$ samples (high $\Delta^{17}O$), $HNO_3$ (moderate $\Delta^{17}O$), and $pNO_3$ (low $\Delta^{17}O$) were each analyzed in dedicated batches."

**Other comments:**
**Comment:** Line 189-190: In citation part, Casciotti et al., 2002 and Sigman et al., 2001 is based on bacterial conversion of nitrate to N2O, while McIlvin and Altabet, 2005 and Walters and Hastings, 2023 using chemical reaction to convert nitrate to N2O. I don't understand the meaning of those citations. Which method are you using?
**Response:** Thank you for pointing out the confusion regarding the isotope conversion methods. We have clarified the citations and corresponding methodologies in the revised manuscript. Specifically, we used the bacterial denitrification method described by Casciotti et al. (2002) and Sigman et al. (2001) to convert $NO_3^-$ in $HNO_3$ denuder and aerosol filter extracts to $N_2O$. For $NO_2^-$ samples derived from $NO_2$ denuder extracts, we used the azide reduction method described by McIlvin and Altabet (2005) and Walters and Hastings (2023). We have clarified this distinction in the revised text on Lines 230–233, "The $\delta^{15}N$, $\delta^{18}O$, and $\Delta^{17}O$ isotope compositions were analyzed using the denitrifier method for $NO_3^-$ samples (e.g., from $HNO_3$ denuder and aerosol filter extracts) following (Casciotti et al., 2002; Kaiser et al., 2007; Sigman et al., 2001) and the sodium azide/acetic acid buffer method for $NO_2^-$ samples (from $NO_2$ denuder extracts), following (McIlvin and Altabet, 2005; Walters and Hastings, 2023)."

**Comment:** Line 195: "The pooled standard deviations of the standards were ±0.1‰ and ±0.6‰ for δ15N and δ18O of the NO3- standards (n = 78) and ±0.3‰ and ±0.3‰ for δ15N and δ18O of the NO2- reference materials (n = 15), respectively. The Δ17O had a pooled standard deviation of ±0.6 ‰ (n = 53) ."Can the author explain the sample size of the different standards tested here and the chamber experimental sample used in this work? Does they have the exactly the same molar amount, at the same level, or not comparable? Besides, the ±0.1‰ is the detection limite of d15N measurement or the 1 SD of your 78 times' measurement?
**Response:** Thank you for the comment. Yes, we strictly adhere to the principle of identical treatment as part of our laboratory quality control protocols. This includes matching sample concentrations, reagent volumes, and treatment conditions between samples and reference materials to ensure analytical consistency. We added this point to the revised manuscript on Lines

243-245. The reported standard deviations are correct and represent pooled standard deviations calculated from multiple runs of well-characterized isotope standards. While the $\delta^{15}N$ standard deviation may appear low, this reflects the result of careful and consistent QA/QC practices and regular IRMS maintenance in our laboratory. For $\delta^{15}N$, we analyzed across three batch runs: USGS34 ($n = 27$) standard deviation = 0.09 ‰;  USGS35 ($n = 24$) standard deviation = 0.16 ‰; IAEA-N3 ($n = 27$) standard deviation = 0.15 ‰. From these results, the pooled standard deviation was calculated to be 0.13 ‰, which we report as 0.1 ‰ (to one significant figure). In addition, our $\delta^{15}N$ standard deviation from IRMS $N_2O$ reference gas pulses are routinely < 0.02‰, further supporting the high analytical precision of our measurements.

**Comment:** Equation 1: usually we use R^{15}N, R^{18}O, and R^{17}O, not {15}RN, {18}RO, and {17}RO, so the xRsample should be changed to R^{x}_{sample}.
**Response:** Thank you for pointing out that our notation was bizarre. We have updated the $R$ notation to be consistent with (Sharp, 2017). This update has been made on Lines 251-252.

**Comment:** Section 2.4: How about the error of this simulation? Does particle inorgnaic nitrate only pair with NH4+?
**Response:** Thank you for the question. While inorganic nitrate can, in principle, associated with various cations, ammonium was the only significant non-volatile cation present in our experimental setup. Therefore, simulated inorganic nitrate was assumed to pair exclusively with $NH_4^+$. We have clarified this in the revised manuscript.

**Comment:** Line 261: one right parenthesis of "(RACM" is missing; "Master Chemical Mechanism v3.3.1" should be "Master Chemical Mechanism v3.3.1 (MCMv3.3.1 )"
**Response:** Thank you for catching this typo. This has been corrected on Line 322 in the revised manuscript.

**Comment:** Line 267: The format of "3" in O3 and NO3 is different from others. Please check.
**Response:** Thanks for pointing this out. The correction has been made on Line 328 in the revised manuscript.

**Comment:**  Line 286: You write $\Delta 17O(O3term) = 39\pm2$‰ here, while you use $\Delta 17O(O3term) = 39.3\pm2.0$‰ in Lines 323, 346, etc. Should be consistence.
**Response:** Thank you for pointing this out. We have used 39 ± 2 ‰ throughout the manuscript to be consistent.

**Comment:**  Line 341: night time also have OH radicals which could form NO3 by the following equation: HNO3 + OH = H2O + NO3. It's better to consider both NO3 from N2O5 as well as HNO3 or NO3- in the $\Delta 17O$ calculation.
**Response:**  Thank you for your suggestion. However, under our experimental conditions, nighttime OH concentrations are expected to be extremely low and unlikely to contribute significantly to $NO_3$ formation via the $HNO_3 + OH$ reaction. For example, based on our box model simulations, [OH] during nighttime experiments was estimated to be on the order of $10^3$ molecules $cm^{-3}$, which is several orders of magnitude lower than daytime concentrations (e.g., ~$10^6$ molecules $cm^{-3}$ n Exp. 1). Furthermore, the rate constant for the reaction $HNO_3 + OH \rightarrow H_2O + NO_3$ is

approximately $1.5 \times 10^{-13}$ cm$^3$ molecule$^{-1}$ s$^{-1}$ at 298 K (Atkinson et al., 2004). Even assuming daytime-level OH concentrations ($10^6$ molecules cm$^{-3}$), this would yield a reaction lifetime for HNO$_3$ of approximately 77 days, which is far too long to be relevant within the timescale of our experiments. Under our actual nighttime OH conditions, this pathway would be even less significant. Therefore, we conclude that this reaction does not contribute meaningfully to NO$_3$ formation in our system, and that N$_2$O$_5$ and NO$_3$ +HC (pinonaldehyde) reactions remain the dominant nighttime NO$_3$ source considered in our $\Delta^{17}$O analysis.

**Comment:** Lines 366-367: "δ15N(NO2) that averaged -52.5 ±25.2‰ (n = 20), which were higher than δ15N(pNO3) that averaged -72.6±22.9‰ (n = 7)" However, -52.5 ±25.2‰ and -72.6±22.9‰ almost at the same level by considering the 25 per mil standard deviations.
**Response:** Thank you for this helpful comment. We agree that the relatively large standard deviations for $\delta^{15}$N(NO$_2$) and $\delta^{15}$N(pNO$_3$) result in overlapping ranges. A two-sample $t$-test confirms that $\delta^{15}$N(HNO$_3$) is significantly different from both $\delta^{15}$N(NO$_2$) and $\delta^{15}$N(pNO$_3$) ($p <$ 0.01), but the difference between NO$_2$ and pNO$_3$ is not statistically significant. We have clarified this in the revised manuscript on Lines 422-425, "Overall, $\delta^{15}$N(HNO$_3$) averaged -25.9±13.0 ‰ ($n$ = 20) and was signficnatly higher than both $\delta^{15}$N(NO$_2$) (-52.5 ±25.2 ‰; $n$ = 20) and $\delta^{15}$N(pNO$_3$) (-72.6±22.9 ‰; $n$ = 7) based on a two-sample $t$-test ($p <$ 0.01). While $\delta^{15}$N(NO$_2$) values were higher than those of $\delta^{15}$N(pNO$_3$), this difference was not statistically significant based on a two-sample $t$-test ($p >$ 0.05)."

**Comment:** Lines 403-404: The author assumed that the NO3- extracted from the filter collection could represent the organic nitrate by using the d15N isotope value. What kind of d15N value should the organic nitrate should have?
**Response**: Thank you for pointing this out. To the best of our knowledge, $\delta^{15}$N values for organic nitrate aerosols have not been directly reported in the literature, and determining these values was one of the main goals of our study. The significant difference in $\delta^{15}$N between HNO$_3$ and pNO$_3$ observed in our study suggests distinct sources. Since equilibrium isotope exchange between HNO$_3$ and inorganic NO$_3^-$ would typically lead to similar $\delta^{15}$N values (offset by 1-3 ‰; (Bekker et al., 2023; Elliott et al., 2009), the large $\delta^{15}$N offset of ~47 ‰ supports the possibility that a portion of the pNO$_3$ is derived from organic nitrate. This is further supported by our AMS measurements, which indicated strong evidence that the NO$_3^-$ aerosol derived from organic nitrate contributions. We have revised the text to reflect this interpretation on Lines 462-466, "For example, the observed significant $\delta^{15}$N difference between HNO$_3$ and pNO$_3$ of ~47 ‰ suggests that these species may originate from distinct sources. Given that inorganic nitrate would typically equilibrate isotopically between HNO$_3$ and NO$_3^-$ with an expected offset of only ~1–3 ‰, and often slightly enriched in pNO$_3$ (Bekker et al., 2023), the substantially lower $\delta^{15}$N values observed in pNO$_3$ imply that the collected nitrate may originate from organic nitrate species or NO$_y$ formation pathways unique from HNO$_3$ production."

Further, we discuss the mechanism that could be at play to explain the low $\delta^{15}$N values observed in the RONO$_2$ derived pNO$_3$ on Lines 550-554, "Finally, given that the collected pNO$_3$ was predominantly derived from RONO$_2$, the negative $\Delta\delta^{15}$N(pNO$_3$ – NO$_2$) values (Fig. 2B) suggest a preferential incorporation of $^{14}$N into the RONO$_2$ product. This is consistent with a NO$_x$ photochemical equilibrium in which $^{15}$N is enriched in NO$_2$, leaving NO relatively depleted in $^{15}$N (Li et al., 2020; Walters et al., 2016). Subsequent reaction of this $^{15}$N-depleted NO with RO$_2$ forms

RONO$_2$, thereby transferring the isotopically lighter nitrogen signature into RONO$_2$ which then condenses to the particle phase and hydrolyzed to NO$_3^-$."

**Comment:** Table 2, why the Filter/AMS of experiment 2 is higher than 100%?

**Response:** Thank you for pointing this out. We agree that the Filter/AMS ratio in Experiment 2 appeared higher than 100 % and have clarified this in the revised manuscript. In the original Table 2, we did not include the propagated uncertainties associated with the AMS, filter-based, and PILS pNO$_3$ measurements. Based on our assessments, we estimate: Filter-based pNO$_3$ collection uncertainty: ±20 % (based on side-by-side collections using the ChemComb speciation cartridge for [pNO$_3$] quantification; (Blum et al., 2020); AMS pNO$_3$ measurement uncertainty: ±14 %; and PILS pNO$_3$ collection uncertainty: ±20 %. Using standard uncertainty propagation rules for division, the total uncertainty for the Filter/AMS ratio is estimated to be approximately ±24 %, and for the PILS/AMS ratio approximately ±24 %. Thus, the reported Filter/AMS ratio of 105.3 24 % falls within the expected range of quantitative collection. We have updated **Table 3** (formerly Table 2) to include these propagated uncertainties for all reported pNO$_3$ quantification techniques and clarified this point in the manuscript. Further, we have added error bars to Fig. 3 to show the uncertainty of the measured [pNO$_3$] values. Finally, we updated the text to include these uncertainty values on Lines 491-494; 501-502; 507-511.

**Comment:** Besides, The calculated f(pNO3, Org) was higher than 1, does it mean this method has some problem?

**Response:** Thank you for raising this point. We agree that the calculated values of $f$(pNO$_3$, Org) exceeding 1 indicate limitations in the estimation method, and we have clarified this issue in the revised manuscript. Specifically, while the $f$(pNO$_3$, Org) values were greater than 1 even when accounting for propagated uncertainties, we had already discussed possible reasons for this in the original manuscript. The $R_{ON}$ (organic nitrate) values used for the $f$(pNO$_3$, Org) calculation were derived from previously reported $R_{ON}/R_{AN}$ ratios from $\alpha$-pinene oxidation experiments conducted under substantially lower precursor concentrations (~10 times lower) than those used in our study (Takeuchi and Ng, 2019). This difference in experimental conditions could introduce systematic bias into the $f$(pNO$_3$, Org) estimates under our high-concentration setup.

To address this uncertainty, we employed a complementary **qualitative approach** based on evaluating the molar ratio of NH$_4$/SO$_4$ from our HR-ToF-AMS measurements (Fig. S2). Across all experiments, the NH$_4$/SO$_4$ molar ratio remained consistently near 1.5. This observation is consistent with pNO$_3$ being dominated by **organic nitrate**, as the dissolution of HNO$_3$ into aerosol followed by NH$_3$ neutralization is anticipated to cause a more abrupt increase in NH$_4$/SO$_4$ ratio than observed (Takeuchi and Ng, 2019). Furthermore, our $\delta^{15}$N isotope data provide additional support for this interpretation. A significant $\delta^{15}$N offset (~47‰) between HNO$_3$ and pNO$_3$ was observed. Given that inorganic nitrate typically equilibrates isotopically between HNO$_3$ and inorganic pNO$_3^-$ with an offset of only ~1–3‰ and is often slightly enriched in pNO$_3$ (Bekker et al., 2023), which is the opposite trend that we observe in the experiments. The substantially lower $\delta^{15}$N values in pNO$_3$ imply that the nitrate we collected likely originated from organic nitrates or from NO$_y$ formation pathways distinct from HNO$_3$ production.

We have added the following text in the revised manuscript on Lines 480–483 to elaborate on this point: ". Overall, both the quantitative and qualitative analysis of $pNO_3$ composition utilizing the AMS data as well as our $\delta^{15}N$ data indicates that $pNO_3$ was mainly derived from organic nitrate. Hereinafter, we will assume that the $NO_3^-$ extracted from the filter collections derived from organic nitrate."

**Comment:** If all pNO3- measured in this experiment all from organic nitrate, does it mean all pNO3- measured in field observation from organic nitrate?

**Response:** No, our findings do not suggest that all $pNO_3^-$ measured in field observations is derived from organic nitrate. However, our results demonstrate that organic nitrate aerosols ($pRONO_2$) can hydrolyze to form $NO_3^-$ in aerosol extracts during sample processing. Therefore, in field environments where organic nitrate concentrations are high (such as regions with elevated BVOC emissions), it is important to recognize that the $NO_3^-$ measured in aerosol extracts could include contributions from both inorganic particulate nitrate (derived from $HNO_3$ uptake) and hydrolyzed organic nitrate ($pRONO_2$). Additionally, we note that some atmospheric $pRONO_2$ species are expected to hydrolyze under ambient conditions, leading to $HNO_3$ formation (Takeuchi and Ng, 2019). However, under our experimental conditions that are characterized by relatively low relative humidity (~30 %) and dry aerosol seeds, we did not observe evidence for substantial $pRONO_2$ hydrolysis to $HNO_3$ during the experiments. Thus, while organic nitrate hydrolysis is a relevant process, its extent will depend strongly on environmental factors such as aerosol liquid water content, RH, and aerosol acidity.

We have added the following text in the revised manuscript on Lines 543–546 to elaborate on this point: "However, we caution that these findings do not imply that all $pNO_3$ observed in ambient field measurements is organic in origin. The extent to which organic nitrate contributes to $pNO_3$ in field settings will depend on regional BVOC emissions, which govern precursor availability, as well as environmental factors such as aerosol pH and relative humidity, which influence the lifetime and hydrolysis rates of $pRONO_2$ prior to filter collection."

**Comment:** Fig.8: For D17O(HNO3) value, the observation results shown different with the simulation results, why?

**Response:** Thank you for the comment. We agree that there are discrepancies between the observed and simulated $\Delta^{17}O(HNO_3)$ values and in the original manuscript we discussed some of the possibilities for this discrepancy. In the revised manuscript, we addressed this more thoroughly. Specifically, we have added several new sensitivity tests to examine the potential causes of the model-observation mismatch. These include evaluations of wall loss and dilution effects (Section 3.3.2), uncertainties in reaction rate constants, organic nitrate ($pRONO_2$) hydrolysis, $N_2O_5$ heterogeneous chemistry, chamber blank contributions, and revised oxygen isotope mass-balance assumptions (Section 3.3.4). Ultimately, we conclude that the model-observation differences in $\Delta^{17}O(HNO_3)$ likely stem from multiple interacting factors: (1) Chamber blank contributions significantly influenced low-$NO_x$ experiments (e.g., Exp. 1, 2), where $HNO_3$ production was limited. (2) $N_2O_5$ heterogeneous chemistry played a critical role in nighttime oxidation,

particularly in Exp. 6. (3) For high-$NO_x$ experiments, improved agreement was achieved by modifying the $\Delta^{17}O$ mass-balance assumptions for the $NO_2$ + OH reaction, specifically, reducing the assumed transfer of $\Delta^{17}O$ from $NO_2$, potentially reflecting isotopic scrambling or oxygen atom exchange during $HNO_3$ formation. These updates are now reflected in the revised manuscript, with expanded discussion in Section 3.3.2 and Section 3.3.4. We have added a new figure evaluating the sensitivity tests (Fig. 6) and have updated the revised mechanism used to resolve measurement-model $\Delta^{17}O(HNO_3)$ discrepancies in the base simulation (Fig. 7). The average model-measurement bias decreased from 6.7±3.3 ‰ in the base mechanism to 1.6 ±1.3 ‰ in the modified mechanism (summarized in Table 5).

**Comment:** Fig.11: some experiments correlated with USC_API mechanism, experiment 5 correlated with USC_API_KIE mechanism. need explaination.

**Response**: Thank you for the comment. In this revised manuscript, we have removed the $\delta^{15}N$ modeling results, including those shown in the previous version of Figure 11, to better focus on the $\Delta^{17}O(HNO_3)$ analysis and address key reviewer concerns. The $\delta^{15}N$ model-measurement comparison involved numerous parameters that require further evaluation, including uncertain $NO_y$ source values and nitrogen isotope fractionation factors, which are not yet well constrained. Interpreting the $\delta^{15}N$ model-measurement comparison requires evaluating numerous parameters, such as $NO_y$ source signatures and nitrogen isotope fractionation values that are currently uncertain or poorly constrained. We believe that the $\delta^{15}N$ modeling is better suited for a dedicated follow-up study focused on quantifying the various fractionation effects and supported by additional targeted experiments investigating specific $HNO_3$ formation pathways. In this manuscript, we instead focused on $\Delta^{17}O(HNO_3)$, where we conducted extensive sensitivity tests and developed a revised mechanism that improves agreement across all experiments (see Section 3.3.4).

**Reviewer #3**
**General Comments:** The goal of this manuscript is to investigate the interaction between nitrogen oxides (NOx) and α-pinene oxidation reactions by conducting chamber experiments and analyzing stable nitrogen (N) and oxygen (O) isotopes of nitrogen species, comparing the results with a model developed by the authors (using separate models for O and N). This research incorporates various innovative aspects, including the development of the model, methods for analyzing the isotopes of different nitrogen compounds, chamber experiments, and the analysis of organic nitrates. The originality of the authors' work is clearly reflected throughout the manuscript. Given the importance and novelty of this research, I believe the manuscript should be considered for publication with revisions addressing the following comments.

**Response:** We thank the reviewer for their thoughtful and supportive comments regarding the originality and importance of our work. We have carefully addressed their major comments and suggestions. In particular, we have restructured the abstract and introduction to place greater emphasis on the experimental work and its relevance for advancing the use of isotopes to track $NO_x$ chemistry. We have added additional details regarding the relationship between organic nitrate and particulate nitrate (p$NO_3$), clarified the potential for isotope exchange between $HNO_3$ and p$NO_3$, and expanded the presentation and interpretation of the $\delta^{15}N$ data. These were very helpful and critical points, and we believe the revisions have substantially strengthened the manuscript. Detailed responses to each comment are provided below in our point-by-point reply.

**Comment: 1. Abstract and Introduction**

The expressions such as "However, uncertainties remain regarding their coupling in NOx loss, renoxification, and oxidation chemistry" and "This study provides insights into Δ17O transfer dynamics, nitrogen isotope fractionation, and the role of NOx-BVOC chemistry in air quality, highlighting the potential of Δ17O and δ15N as tools for evaluating complex atmospheric processes" are quite general and lack specific details about the research. These phrases may be more appropriate for a broader overview, but for a scientific manuscript, I recommend specifying how the controlled laboratory experiments involving α-pinene, which are introduced for the first time, are crucial to understanding the issue. The introduction would benefit from a clearer statement about the significance of this understanding and how the absence of this knowledge has impacted the field. Specifically, I would like the authors to emphasize the necessity of considering organic nitrate in the context of previous observations of inorganic nitrate's Δ17O and δ15N, and how these experiments and models address gaps in previous work.

**Response:** Thank you for the thoughtful comment. We agree that our initial framing in the abstract and introduction was overly general and did not sufficiently highlight the novelty and importance of our controlled chamber experiments with $\alpha$-pinene. In response, we have substantially revised the abstract to explicitly describe how these experiments address critical gaps in understanding the fate of $RONO_2$ and its contributions to $HNO_3$ and $pNO_3$ budgets. We now emphasize how isotopic measurements ($\Delta^{17}O$, $\delta^{15}N$, $\delta^{18}O$) provide unique constraints on nitrate formation pathways, particularly distinguishing organic versus inorganic sources, and how our findings help to test and refine mechanistic assumptions in current models.

In the revised introduction, we now more clearly articulate the significance of resolving the fate of $RONO_2$ and its isotopic signature on Lines 72-77. We also discuss how uncertainties in previous $\Delta^{17}O$ and $\delta^{15}N$ interpretations for inorganic nitrate may stem from untested assumptions regarding $\Delta^{17}O$ transfer dynamics and nitrogen isotope fractionation patterns on Lines 105-109 and Lines 132-138. These revisions help to contextualize why the chamber experiments are essential and what specific knowledge gaps they address in the broader field of atmospheric nitrogen cycling and nitrate isotope chemistry.

**Comment:  2. Organic Nitrate and p-NO3 Relationship**
I understand that organic nitrate is collected after passing through a denuder system and hydrolyzed to nitrate for isotope analysis. However, is organic nitrate always expected to exist as a particle (nothing in gas-phase?)? For example, PAN is considered to exist in the gas phase. Thus, the equivalence of organic nitrate and p-NO3 is somewhat unclear. At the very least, I believe the authors should clarify why p-NO3 is considered to be organic nitrate and provide further discussion, potentially citing references, about the particle-phase composition of organic nitrate as they see it.

**Response:** Thank you for this thoughtful comment. We agree that organic nitrates can exist in both the gas and particle phases, including gas-phase $RONO_2$ and $RO_2NO_2$ species such as PAN. In our experiments, we had online measurements of peroxyacetyl nitrate (PAN) using a CIMS and qualitative detection of gas-phase organic nitrates using a FIGAERO-HR-ToF-CIMS, although these gas-phase organic nitrate data were not fully calibrated (thus the data was not shown). However, **we do not believe** that gas-phase organic nitrates significantly impacted our denuder–filter system measurements. Typically, gas-phase organic nitrates are collected using a specialized

XAD-coated denuder system (e.g., Rindelaub, McAvey, Shepson, 2015). Our denuder system, optimized for the collection of $HNO_3$ and $NO_2$, was designed to minimize such interferences. Supporting this, our $HNO_3$ denuder-based measurements showed strong correlation ($R^2 = 0.98$) with simultaneous online CIMS measurements of $HNO_3$ (Blum et al., 2020), and our $NO_2$ denuder-based measurements were strongly correlated ($R^2 = 0.97$) with online CAPS measurements (Blum et al., 2020). These results suggest that the denuders selectively collected the targeted compounds without significant cross-interference from gas-phase organic nitrates.

Additionally, while it is known that some denuder coatings can hydrolyze PAN under certain conditions, previous studies have shown that PAN is not significantly collected on the guaiacol/KOH-coated denuders used for $NO_2$ collection in our study (Buttini et al., 1987). This is consistent with preliminary laboratory testing we conducted to validate our denuder performance prior to the experiments. Regarding the particulate phase, the strong agreement between our filter-based $pNO_3$ measurements and AMS $pNO_3$ measurements indicates that the nitrate extracted from the filters primarily represents $pNO_3$ (as opposed to additional contributions from gas phase organic nitrates). This further supports the interpretation that the collected nitrate on the filters is derived from particulate organic nitrate species rather than from gas-phase nitrate or gas-phase organic nitrates. One of the strengths of our experimental setup was the integration of multiple online and offline instruments, allowing us to cross-validate collection efficiencies and target specificity. This comprehensive approach provides confidence that our isotope measurements reflect the intended nitrogen species with minimal artifact contribution. We have added clarification to the revised manuscript to address this important point and included relevant references where appropriate. We have added the following to the revised text on Lines 547-550: "Further, while gas-phase organic nitrates (e.g., $RONO_2$, $RO_2NO_2$) can be present and were detected by CIMS measurements during the experiments, the strong agreement between filter-based and AMS-based $pNO_3$ measurements supports that the nitrate extracted from aerosol filters was primarily derived from particle-phase organic nitrate rather than from gas-phase organic nitrate contributions."

Regarding the concern about why $pNO_3$ is considered to be predominantly organic nitrate in this study, we note that multiple independent lines of evidence support this interpretation:

**AMS Mass Spectral Analysis ($f$(pNO3, Org)):** We calculated the fraction of $pNO_3$ derived from organic nitrates ($f$(pNO_3, Org)) using $NO^+/NO^{2+}$ signals from the HR-ToF-AMS. Across all experiments, the f(pNO_3, Org) values averaged $1.25 \pm 0.04$, suggesting that the $pNO_3$ signal was dominated by organic nitrate. (Lines 469-474)

**Molar Ratio of NH4/SO4:** We further examined the $NH_4/SO_4$ molar ratios from the AMS data. Ratios remained consistently near 1.5 for the duration of all experiments. In contrast, significant $HNO_3$ uptake and neutralization would be expected to cause a sharp increase in the $NH_4/SO_4$ ratio (Takeuchi and Ng, 2019), which was not observed. (Lines 473-478)

**$\delta^{15}N$ Evidence:** We observed a large $\delta^{15}N$ difference (~47 ‰) between $HNO_3$ and $pNO_3$, indicating that these species originate from distinct sources. Since inorganic particulate nitrate would be expected to isotopically equilibrate with $HNO_3$, resulting in a much smaller offset (~1–

3 ‰) and slight enrichment in $pNO_3$ (Bekker et al., 2023), the substantial $\delta^{15}N$ difference supports an organic nitrate origin for the collected $pNO_3$. (Lines 462-466)

Together, these isotopic and chemical composition analyses provide strong support that the collected $pNO_3$ was predominantly derived from organic nitrate species rather than inorganic nitrate. We have expanded on our discussion of the $\delta^{15}N$-based evidence of organic nitrate dominating $pNO_3$ on Lines 462-466, "For example, the observed significant $\delta^{15}N$ difference between $HNO_3$ and $pNO_3$ of ~47 ‰ suggests that these species may originate from distinct sources. Given that inorganic nitrate would typically equilibrate isotopically between $HNO_3$ and $NO_3^-$ with an expected offset of only ~1–3 ‰, and often slightly enriched in $pNO_3$ (Bekker et al., 2023), the substantially lower $\delta^{15}N$ values observed in $pNO_3$ imply that the collected nitrate may originate from organic nitrate species or $NO_y$ formation pathways unique from $HNO_3$ production.

**Comment: 3. Isotopic Exchange Between HNO3 and p-NO3**
It seems that the authors assume no isotopic exchange between HNO3 and p-NO3 after production in the chamber experiments. However, existing research suggests that HNO3 and p-NO3 should be in equilibrium, with HNO3 showing a lower δ15N and p-NO3 a higher δ15N (e.g., Geng et al., 2014 PNAS). The authors' atmospheric observations also reproduce this δ15N pattern between HNO3 and p-NO3 (Bekker et al., 2023, https://doi.org/10.5194/acp-23-4185-2023). The authors seem to suggest that no such isotopic exchange occurs in their experiments, but it would be helpful to introduce this existing pattern and explain why it does not occur in their experiments. Specifically, what conditions in their chamber experiments prevent isotopic exchange between HNO3 and p-NO3.

**Response:** Thank you for this helpful comment. We agree that isotopic exchange between $HNO_3$ and $pNO_3$ has been observed in ambient field studies, typically resulting in a small $\delta^{15}N$ offset (~1–3 ‰) between simultaneously collected samples, with $pNO_3$ slightly enriched relative to $HNO_3$ (Bekker et al., 2023; Elliott et al., 2009). A much larger enrichment factor of up to ~21 ‰ has also been proposed to explain $\delta^{15}N$ variations observed in a Summit, Greenland ice core (Geng et al., 2014). However, an enrichment factor of this magnitude (still favoring $\delta^{15}N$ enrichment in $pNO_3$) has not been observed in recent simultaneous $HNO_3$ and $pNO_3$ ambient collections, where particulate nitrate is assumed to be dominated by inorganic nitrate (e.g., in the northeastern U.S., where biogenic VOC emissions are low), observed from laboratory fractionation of $NO_3^-(aq)$ from $HNO_3$ (g) (~2 ‰; Blum et al., 2020) or theoretically calculated (~3‰; Walters and Michalski, 2015), which all point towards a lower enrichment than needed to explain the ice core $\delta^{15}N$ shift. Thus, although there maybe uncertainty regarding the exact magnitude of the enrichment, all studies consistently suggest that $\delta^{15}N$ should be enriched in $pNO_3$ and depleted in $HNO_3$.

However, in our chamber experiments, we observed a different pattern: the average paired $\delta^{15}N$ difference between $HNO_3$ and $pNO_3$ was ~47 ‰, with $\delta^{15}N(HNO_3)$ higher than $\delta^{15}N(pNO_3)$, opposite to what would be expected if isotopic exchange had occurred. This suggests that isotopic exchange between $HNO_3$ and $pNO_3$ was not a dominant process in our system. We speculate that this difference arises due to the chemical speciation of the $pNO_3$. In field studies, the $pNO_3$ is presumably inorganic (derived from $HNO_3$ condensation), allowing for isotopic equilibrium with $HNO_3$, following the thermodynamic equilibrium involving the $HNO_3$ partitioning between the

gas and condensed phase. In contrast, multiple lines of evidence ($\delta^{15}$N, AMS $f$(pNO$_3$, Org), NH$_4$/SO$_4$ ratios) suggest that pNO$_3$ in our experiments was dominated by organic nitrate species (pRONO$_2$). Therefore, isotopic exchange between HNO$_3$ and organic nitrate may not occur or occur at rates that are negligible over the timescale of our experiments. We have clarified this point and added discussion of this issue in the revised manuscript. On Lines 462-466, we added: "For example, the observed significant $\delta^{15}$N difference between HNO$_3$ and pNO$_3$ of ~47‰ suggests that these species may originate from distinct sources. Given that inorganic nitrate would typically equilibrate isotopically between HNO$_3$ and NO$_3^-$ with an expected offset of only ~1–3 ‰, and often slightly enriched in pNO$_3$ (Bekker et al., 2023), the substantially lower $\delta^{15}$N values observed in pNO$_3$ imply that the collected nitrate may originate from organic nitrate species or NO$_y$ formation pathways unique from HNO$_3$ production."

**Comment: 4. Presentation of δ15N**
In Figures 2 and 9-11, it seems that δ15N is shown relative to atmospheric N2. Given that the initial δ15N of the N sources differs, the resulting δ15N values also vary. For clarity, I suggest normalizing the data by presenting δ15N vs initial NOy values, which would better illustrate the change relative to initial values. Or, it would also be helpful to specify on the y-axis of the figures not only the units in permil but also the reference point (e.g., vs. N2), and add d15N of initial precursor.
**Response:** Thank you for the helpful suggestion. After considering this comment and the feedback from all reviewers, we have revised our approach to presenting and interpreting the $\delta^{15}$N data. Specifically, we have shifted our focus from absolute $\delta^{15}$N values (referenced to Air) to the differences in $\delta^{15}$N between species, namely $\Delta\delta^{15}$N(HNO$_3$ - NO$_2$) and $\Delta\delta^{15}$N(pNO$_3$ – NO$_2$). This approach reduces uncertainties associated with assumptions about the starting $\delta^{15}$N values of the NO$_y$ reactants, which were not all directly measured in this study. Importantly, this change maintains the scientific impact of the work by still allowing robust investigation of isotope fractionation processes between the various NO$_y$ species.

In the revised Figure 2, we now clearly label Panel A as showing $\delta^{15}$N values referenced to air (‰ vs. Air). We have also added a new Panel B that presents the $\Delta\delta^{15}$N values (i.e., differences relative to NO$_2$). Similarly, we have updated all relevant model $\delta^{15}$N figures to focus on $\Delta\delta^{15}$N comparisons and have revised Section 3.1.2 to reflect this shift in interpretation, specifically on Lines 427-443. We believe these changes improve the clarity and consistency of the $\delta^{15}$N presentation and address the reviewer's helpful suggestion.

**Specific Comments**
**Comment: L146**: It is stated that particles collected on a Teflon filter using FIGAERO are analyzed using HR-ToF-I-CIMS. However, it is unclear how the particles collected on the Teflon filter are introduced into HR-ToF-I-CIMS for analysis. Specifically, the process by which the filter particles are treated and converted into a measurable form for analysis is not described in detail. Could you please clarify this step?
**Response:** Thank you for pointing this out. We understand how the original wording may have been confusing. The FIGAERO-HR-ToF-I-CIMS was used for the quantification of gaseous

organic nitrates. The Teflon filter in the FIGAERO setup served to remove particulate components prior to the gas-phase analysis. No particle-phase analysis was conducted using the HR-ToF-I-CIMS in this work. We have modified the text in the revised manuscript on Line 182 to clearly state that the FIGAERO-HR-ToF-I-CIMS measurements correspond to gaseous organic nitrate.

**Comment: L152**: Please provide more detail on "water-soluble aerosol components".
**Response:** Thank you for the comment. In our study, "water-soluble aerosol components" refer to the aerosol constituents that are soluble in water and thus measurable by the particle-into-liquid sampler (PILS) coupled to ion chromatography (IC). The PILS collects particles into a continuous flow of ultrapure water, meaning that only the water-soluble fraction of the aerosol is analyzed. This approach differs from the HR-ToF-AMS, which measures both water-soluble and water-insoluble aerosol components. For example, nitrate measured by the AMS represents total particulate nitrate ($pNO_3$, including both inorganic and organic forms), whereas nitrate measured by the PILS-IC system corresponds only to the water-soluble fraction of particulate nitrate (inorganic nitrate + potentially some organic nitrate). We have updated the text on Lines 188-191 to clarify the meaning of "water-soluble aerosol components" and to briefly explain the distinction between the PILS-IC and AMS measurements as follows: "This method differs from the HR-ToF-AMS measurements, which quantify total aerosol composition, including both water-soluble and water-insoluble components. For example, nitrate measured by the AMS represents total $pNO_3$, whereas nitrate measured by the PILS-IC system corresponds only to the water-soluble fraction of $pNO_3$."

**Comment: L167**: Why was a quartz filter used? It seems that Teflon might also have been considered, as noted in L146. Also, was it considered that organic nitrate primarily exists in the particle phase and may not have been captured in the HNO3 or NO2 collections? Did the authors explore the possibility of gas-phase organic nitrates, and if so, how did the filter pass through them?
**Response:** Thank you for the comment. We used quartz filters rather than Teflon filters primarily because the filters were extracted in water after collection. In our laboratory experience, Teflon filters can be challenging to extract efficiently in water due to their hydrophobic nature. In contrast, quartz filters are more hydrophilic, facilitating better extraction of water-soluble species. Additionally, because we were targeting organic components, quartz filters were advantageous as they could be pre-fired at high temperatures to remove background organic contaminants. Teflon filters cannot be heated to similarly high temperatures without degradation, making quartz filters more suitable for our experimental objectives. Teflon filter was used for aerosol removal before entry to the FIGAERO-HR-ToF-I-CIMS following the standard lab protocol used for that instrument, which is independent from our aerosol collection for offline $NO_3^-$ concentration and isotope analysis.

As noted in our response to Comment #2, we do not anticipate significant collection of gas-phase organic nitrates in our denuder–filter sampling setup. The denuder removed gas-phase species upstream of the filter, and our evaluation with online CIMS and AMS measurements suggests that the collected $pNO_3$ primarily represented particle-phase material. Thus, the quartz filters were intended to capture particle-phase species, including particulate organic nitrates, with minimal

interference from gas-phase organic nitrates. We added the following to the revised manuscript on Line 208-210, "Quartz filters were used to facilitate water extraction and enable high-temperature pre-cleaning for organic vapor removal."

**Comment: L206**: The 'O' should not be italicized.
**Response:** Thank you for catching this error. This has been appropriately corrected in the revised manuscript on Line 256.

**Comment: L207**: The term "mass-independent effect" is unclear. Does it refer to mass-independent isotope effects or mass-independent fractionation?
**Response:** Thanks for pointing this out. We have revised the terminology to "mass-independent fractionation". This correction has been made on Line 257.

**Comment: L296**: If discussing NOx oxidation, reactions such as NO2 photolysis are not included as an "oxidation process". If needed, please rephrase this.
**Response:** Thank you for the comment. However, since we have removed the $\delta^{15}N$ modeling from the revised manuscript, the sentence on Line 296 has also been removed and is no longer included in the current version.

**Comment: L327**: The difference between 11.8 ‰ and 23.2 ‰ is not a "slight difference". The authors suggest this difference arises due to kinetic isotope effects associated with RO2/HO2 reactions. Are there any relevant studies that predict such a difference between O2 and RO2/HO2, and is this comparison reasonable?
**Response:** Thank you for the comment, which Reviewer #1 also brought up. Our response to this as previously stated, "Thank you for noting that the difference between the $\delta^{18}O$ values is larger than originally conveyed. We agree that the use of the word "near" may have been misleading and have revised the sentence for clarity. While the derived $\delta^{18}O$ value of 11.1 ‰ is ~12 ‰ lower than that of atmospheric $O_2$ (~23 ‰), this difference is still relatively modest in the context of the broad $\delta^{18}O$ range predicted in atmospheric oxidants to range from ~-50 to 120 ‰. We have updated the text on Lines 377-385: "The derived low-end $\delta^{18}O$ oxidant value is consistent with a scenario in which these radicals are sourced from atmospheric $O_2$ ($\delta^{18}O \approx 23$ ‰; Craig, 1957), but undergo isotopic fractionation during their formation and subsequent oxygen atom transfer to $NO_y$. Although the exact $\delta^{18}O$ enrichment factors for $RO_2/HO_2$ or OH formation and reaction are not well-constrained, a net isotope enrichment factor ~-12 ‰ is plausible, particularly for unidirectional reactions involving $^{18}O$ fractionation (Walters and Michalski, 2016). Additionally, contributions from OH could further influence the low $\delta^{18}O$ endmember, especially if oxygen atom exchange with ambient water vapor occurs (Dubey et al., 1997). Altogether, the inferred $\delta^{18}O$ of 11.1 ±1.0 ‰ for the low-$\Delta^{17}O$ oxidant endmember likely represents a composite signal from multiple oxidants ($RO_2$, $HO_2$, OH) originating from $O_2$ and/or $H_2O$, modified by kinetic and equilibrium isotope effects."

**Comment: L337**: It is clear that a balance exists between NO branching ratios involving O3 and RO2/HO2. However, it would be helpful to add that O3 has a high Δ17O value and RO2/HO2 has a low Δ17O to clarify this point.
**Response:** Thank you for the suggestion. We have added the following text to the revised manuscript on Lines 370-372, "The observed $\Delta^{17}O$ and $\delta^{18}O$ values of $NO_y$ species are anticipated

to reflect a balance between oxygen atom transfer from $O_3$, which has high $\Delta^{17}O$ and $\delta^{18}O$ values, and oxygen atom transfer from $RO_2$, $HO_2$, OH, and $H_2O$ which have lower $\Delta^{17}O$ and $\delta^{18}O$ values."

**Comment: L371**: Could the authors explain why 1.040 was used for Fang et al., 2021?
**Response:** Yes, thank you for the question. We have added clarification to the revised manuscript on Lines 432–435: "…based on the assumption that $NO_2$ and the excited $HNO_3$ intermediate formed during the $NO_2$ + OH reaction achieve isotopic equilibrium prior to collisional deactivation"

**Comment: L377-379**: I agree with predicting from isotopic data, but it would be helpful to discuss why condensation from HNO3 to p-NO3 or thermal equilibrium between HNO3 and p-NO3 does not occur in the chamber. The discussion should address this point as it is linked to Major Comment 2.
**Response:** Thank you for the thoughtful comment. We agree that this is an important consideration. We have added a sentence to clarify why condensation of $HNO_3$ to $pNO_3$ was unlikely under our experimental conditions. Specifically, the aerosol was highly acidic and contained significant organic content, both of which would suppress $HNO_3$ partitioning into the particle phase. This supports our interpretation that $pNO_3$ is primarily derived from organic nitrate pathways. The revised sentence is now included in the manuscript on Lines 479-484, "Furthermore, the acidic nature of the particles and limited availability of $NH_4^+$ likely inhibited $HNO_3$ uptake, suppressing condensation pathways and reinforcing the interpretation that $pNO_3$ originated predominantly from organic nitrate formation.

**Comment: 3**: The y-axis is labeled "p-NO3", but I believe AMS is not directly measuring p-NO3. Other methods involve converting the particle form to NO3- before measurement. Could the authors clarify this discrepancy?
**Response:** We believe the reviewer is referring to Figure 3. We would like to clarify that the AMS does directly measure $pNO_3$. We recognize that different methods quantify $pNO_3$ through different techniques. For example, filter-based and PILS-based techniques involve collecting particles and extracting nitrate into aqueous solution prior to analysis, whereas AMS measures the total particulate nitrate (both inorganic and organic nitrate) directly in the particle phase. Discrepancies between the various $pNO_3$ quantification methods are discussed in detail in the revised text on Lines 491–512. We have also added uncertainty estimates (±20 % for filters, ±14 % for AMS, and ±20 % for PILS), which show that the filter- and AMS-based $pNO_3$ measurements are in close agreement when considering measurement uncertainties. We also note that the PILS tends to underestimate $pNO_3$ relative to AMS because it only captures the water-soluble fraction of particulate nitrate and may not fully capture less soluble organic nitrate species that are still detected by AMS. We have revised Figure 3 caption as follows, "The observed $pNO_3$ concentration data are shown for each of the conducted experiments. Concentrations were determined using a High-Resolution Time-of-Flight Aerosol Mass Spectrometer (HR-ToF-AMS), a particle-into-liquid sampler (PILS), and filter collection (Filter). The AMS measures total particulate nitrate, including both inorganic and organic nitrate species. The PILS method quantifies only the water-soluble fraction of particulate nitrate, while filter extractions represent water-soluble components as well as particulate organic nitrates that could hydrolyze over approximately one week. The start of chamber dilution is indicated by the dashed vertical lines,

corresponding to the abrupt decrease in pNO$_3$." Further, we have added uncertainty bands to the pNO$_3$ measurements.

**Comment: L491**: The relative production routes of organic nitrate (+OH/O2/NO vs +NO3) are unclear. I believe it would be useful to describe the relationship between Δ17O values and oxidants in this context.

**Response:** Thank you for the helpful comment. We have added **Table 1** to clarify the expected $\Delta^{17}O$ values from different organic nitrate formation pathways based on isotopic mass balance. We also revised the text to better explain the link between observed $\Delta^{17}O$ values and the dominant oxidants involved, on Lines 706-708, "The simulated $\Delta^{17}O(ONIT)$ values closely matched the $\Delta^{17}O(pNO_3)$ observations, with an average bias of -1.4 ± 2.4 ‰ ($n = 7$), suggesting that the model accurately captured the relative contributions of organic nitrate production routes (RO$_2$ + NO vs. BVOC + NO$_3$; see Table 1) under the various experimental conditions."

**References:**

Bekker, C., Walters, W. W., Murray, L. T., and Hastings, M. G.: Nitrate chemistry in the northeast US – Part 1: Nitrogen isotope seasonality tracks nitrate formation chemistry, Atmospheric Chemistry and Physics, 23, 4185–4201, https://doi.org/10.5194/acp-23-4185-2023, 2023.

Blum, D. E., Walters, W. W., and Hastings, M. G.: Speciated Collection of Nitric Acid and Fine Particulate Nitrate for Nitrogen and Oxygen Stable Isotope Determination, Analytical Chemistry, 92, 16079–16088, 2020.

Buttini, P., Di Palo, V., and Possanzini, M.: Coupling of denuder and ion chromatographic techniques for NO$_2$ trace level determination in air, Science of The Total Environment, 61, 59–72, https://doi.org/10.1016/0048-9697(87)90356-1, 1987.

Elliott, E. M., Kendall, C., Boyer, E. W., Burns, D. A., Lear, G. G., Golden, H. E., Harlin, K., Bytnerowicz, A., Butler, T. J., and Glatz, R.: Dual nitrate isotopes in dry deposition: Utility for partitioning NO$_x$ source contributions to landscape nitrogen deposition, Journal of Geophysical Research. Biogeosciences, 114, 2009.

Geng, L., Alexander, B., Cole-Dai, J., Steig, E. J., Savarino, J., Sofen, E. D., and Schauer, A. J.: Nitrogen isotopes in ice core nitrate linked to anthropogenic atmospheric acidity change, PNAS, 111, 5808–5812, https://doi.org/10.1073/pnas.1319441111, 2014.

Li, J., Zhang, X., Orlando, J., Tyndall, G., and Michalski, G.: Quantifying the nitrogen isotope effects during photochemical equilibrium between NO and NO$_2$: implications for $\delta^{15}N$ in tropospheric reactive nitrogen, Atmospheric Chemistry and Physics, 20, 9805–9819, https://doi.org/10.5194/acp-20-9805-2020, 2020.

Sharp, Z.: Principles of Stable Isotope Geochemistry, 2nd Edition, Open Textbooks, https://doi.org/https://doi.org/10.25844/h9q1-0p82, 2017.

Takeuchi, M. and Ng, N. L.: Chemical composition and hydrolysis of organic nitrate aerosol formed from hydroxyl and nitrate radical oxidation of $\alpha$-pinene and $\beta$-pinene, Atmospheric Chemistry and Physics, 19, 12749–12766, https://doi.org/10.5194/acp-19-12749-2019, 2019.

Walters, W. W. and Michalski, G.: Theoretical calculation of nitrogen isotope equilibrium exchange fractionation factors for various $NO_y$ molecules, Geochimica et Cosmochimica Acta, 164, 284–297, https://doi.org/10.1016/j.gca.2015.05.029, 2015.

Walters, W. W. and Michalski, G.: Ab initio study of nitrogen and position-specific oxygen kinetic isotope effects in the $NO + O_3$ reaction, The Journal of Chemical Physics, 145, 224311, https://doi.org/10.1063/1.4968562, 2016.

Walters, W. W., Simonini, D. S., and Michalski, G.: Nitrogen isotope exchange between NO and $NO_2$ and its implications for $\delta^{15}N$ variations in tropospheric $NO_x$ and atmospheric nitrate, Geophys. Res. Lett., 43, 2015GL066438, https://doi.org/10.1002/2015GL066438, 2016.

---

## Author Response (AR2)

**Response to Editor:**

**Comment**: Public justification (visible to the public if the article is accepted and published): Table 1: first entry should be NO + O3, not NO3 + O3.

**Response**: Thank you for catching this typo. Table 1 has been corrected accordingly.